# Bifurcation analysis of an influenza A (H1N1) model with treatment and vaccination

**Kazi Mehedi Mohammad** [ID]**, Asma Akter Akhi, Md. Kamrujjaman** [ID]*

Department of Mathematics, University of Dhaka, Dhaka, Bangladesh

* kamrujjaman@du.ac.bd

**Data Availability Statement:** All relevant data are within the article and its Supporting information files.

**Funding:** The author(s) received no specific funding for this work.

## Abstract

This research uses numerical simulations and mathematical theories to simulate and analyze the spread of the influenza virus. The existence, uniqueness, positivity, and boundedness of the solution are established. We investigate the fundamental reproduction number guaranteeing the asymptotic stability of equilibrium points that are endemic and disease-free. We also examine the qualitative behavior of the models. Using the Lyapunov method, Routh-Hurwitz, and other criteria, we explore the local and global stability of these states and present our findings graphically. Our research assesses control policies and proposes alternatives, performing bifurcation analyses to establish prevention strategies. We investigate transcritical, Hopf, and backward bifurcations analytically and numerically to demonstrate disease transmission dynamics, which is novel to our study. Contour plots, box plots, and phase portraits highlight key characteristics for controlling epidemics. The disease's persistence depends on its fundamental reproduction quantity. To validate our outcomes, we fit the model to clinical data from influenza cases in Mexico and Colombia (October 1, 2020, to March 31, 2023), aiming to analyze trends, identify critical factors, and forecast influenza trajectories at national levels. Additionally, we assess the efficacy of implemented control policies.

## Introduction

Based on historical records, the flu has existed for at least 1,500 years. Hippocrates (5th century BC) is credited with first describing the history of influenza when he wrote that a disease like the flu had traveled from northern Greece to the southern islands and beyond. When a flu outbreak struck Florence, Italy in the 1300s, the city's officials dubbed it influenza di freddo, or "cold influence", presumably a reference to their theory of the disease's cause.

Numerous flu epidemics have been documented throughout history, ranging from one that originated in Asia and moved to Europe and Africa in 1580 to others that have occurred throughout centuries in Europe and Britain. With a third of the world's population affected and an estimated 50 million deaths, the 1918 "Spanish flu" pandemic is referred to as the "mother of all pandemics" and was the deadliest pandemic in history [1, 2].

Throughout human history, pandemics and epidemics have decimated the population several times, frequently changing the course of history drastically and putting an end to entire

**Competing interests:** The authors have declared that no competing interests exist.

civilizations. Mathematical models that explain the dynamics of infectious diseases are crucial to public health because they shed light on adopting effective and feasible disease-control measures. Flu, or influenza, is a condition brought on by the Orthomyxoviridae virus that mostly affects the nose, bronchi, throat, and occasionally the lungs. Humans who have influenza frequently have fever over 38 degrees Celsius, coughing, headaches, sniffles, and anorexia. ILI, or Influenza-like Illness, is an unpleasant body condition. Virus transmission lasts 2–7 days and typically resolves on its own [3–5]. This flu is often referred to as a self-limiting disease by Indonesians. If there are no complications from other disorders, the illness will be gone in 4–7 days. The immune system of a person has a significant impact on the disease's severity. The creation of airborne particles and virus-containing aerosols is essential for respiratory transmission. When people speak and breathe normally, aerosols are created. Sneezing is a method of expulsion from the nasal cavity that is more efficient if the infection generates more snot [6–8].

The World Health Organization (WHO) proclaimed the influenza A (H1N1) virus, a new virus, to be a pandemic on June 11, 2009, after it was discovered in Mexico and the United States [2]. A (H1N1) virus strain that primarily affected children and young people without immunity to the new strain and started in North America but spread globally also became a pandemic [9, 10]. Since many elderly persons had previously been exposed to a similar H1N1 virus strain, they were shielded by their antibodies. However, it caused the deaths of more than 200,000 individuals worldwide [2].

Despite the availability of vaccinations for numerous infectious diseases, the world continues to experience significant suffering and mortality from these diseases. Given this context, additional research is essential to determine the control of the widespread transmission of the H1N1 influenza A pandemic virus. Sequencing the mathematical models, Nguyen Huu Khanh has considered the SEIR model, which describes the influenza virus's propagation while taking human illness resistance into account [11]. Consequently, in the simulation, an exposed or infected individual. Without therapy, a person could return to a vulnerable one. Data fitting by the least square method including analytical behavior are studied in [12]. Tian et al. have developed mathematical modeling of COVID-19 with vaccination using fractional derivatives with a case study in [13]. Recently, Asghar et al. have investigated the analysis of the SARS-Cov-2 disease with reinfection in [14].

In a series, many authors looked to the well-known SEIR model or its adaptations to explain how people move through various compartments, which stand in for the phases of disease across the entire population over time [6, 15, 16]. By receiving an annual influenza vaccination, people can avoid influenza. Due to the virus's fast mutation, a vaccine created for one year might not be effective the following year [17]. Additionally, the virus's antigenic drift may occur after the year's vaccine has been developed, making it less protective. As a result, outbreaks are more likely to happen, especially among high-risk populations. Other precautions include avoiding ill people, hiding coughs and sneezes, and often washing your hands [1, 2, 6].

In a recent study, stability analysis and optimal control of avian influenza virus A with time delays is analyzed in [18]. Samanta et al. carried out mathematical and computer modeling along with permanence and extinction for a nonautonomous avian-human influenza epidemic model with distributed time delay [19]. Furthermore, the global dynamics of a nonautonomous SIRC model for influenza A with distributed time delay based on differential equations and dynamical Systems is studied effectively [20]. Meanwhile, in [21] Saha et al. have investigated the dynamical behavior of the SIRS model incorporating government action and public response in the presence of deterministic and fluctuating environments.

Most pandemics present an exponential curve followed by a gradual flattening, reducing the epidemic peak [22]. Without established therapies or vaccinations, understanding the

transmission dynamics of a new infectious disease is imperative for flattening the curve. Mathematical models are essential resources for public health authorities, as their decisions on optimizing control measures rely on the model's short- and long-term predictions [23]. Amongst the various models used for describing such epidemic evolution, classic compartmental models such as SIR and SEIR are of immense value to decision-makers and even non-expert operators for their simplicity, reliability, and usage of multiple data sources [1, 2, 24]. During seasonal influenza outbreaks, epidemiological models have played a crucial role for countries like Italy and Mexico, providing insights into current situations, evaluating the effectiveness of outbreak control measures, exploring alternative interventions, and offering guidance for similar settings [2]. Moreover, in disadvantaged settings, prediction models with multiple features will be of great value to healthcare workers for monitoring patients within limited resources [1, 22, 25, 26].

In the case of the modeling of infectious disease, the traditional SIR model permits the determination of critical conditions of disease occurrence in the population with total population size [22]. Influenza is classified as a person-to-person transmissible disease. In several cases, infected people have no apparent symptoms, and, in those cases, an SEIR model is principally used since the exposed class (E) individuals spread the infection rapidly. In contrast to the SIR epidemic model, SEIR is a more updated and sophisticated model that is biologically plausible regarding numerous pandemics and infectious disorders [23, 27]. Hence, a multi-compartment model is an effective tool for forecasting the dynamics of present influenza disease patterns.

Considering the existing literature and historical notes, the main objectives of this article are:

- Vaccination and treatment strategy to control the disease spreading and outbreaks.

- Perform theoretical observation of the model by examining the solution's existence, positivity, and boundedness.

- Observe the basic reproduction number with vaccination and without vaccination.

- Perform Hopf, forward and backward bifurcation analysis of disease-free and endemic equilibrium and analyze their local and global stability.

- Contour plots, box plots, and phase plane analyses were conducted to scrutinize single and multi-compartment interactions.

- To validate the model, real data from influenza cases in Mexico and Colombia are used.

The findings of this article regarding the goal are:

- Disease-free equilibrium (DFE) and endemic equilibrium (EE) points persist in the system.

- Persistence theorem considering the eigenvalue and their corresponding eigenfunction analysis.

- After numerical simulation, it is evident that transcritical bifurcation occurs at DFE when the basic reproduction number ($\mathcal{R}_0$) is equal to one. That means the DFE becomes stable and unstable.

- Backward bifurcation property arises in the model because of the reinfection of the susceptible population. Moreover, the disease transmission rate, population density, interventions, and contact patterns influence the relationship between the basic reproduction number and the force of infection.

- When a Hopf bifurcation occurs, the disease-free equilibrium becomes unstable, and a stable limit cycle appears. This cyclic pattern shows the periodic oscillations in the dynamics of the disease, with the number of individuals in each compartment varying over time.

- We conducted contour plots, box plots, and relative influence analyses to illustrate various scenarios and their effects on the basic reproduction number $\mathcal{R}_0$.

- The model is validated with clinical data from Mexico and Colombia.

This paper is organized as follows: mathematical model is discussed elaborately at the begining of this article. Existence, positivity, and boundedness of solutions are described in the supporting information section (Appendix 1). Further, the determination of fixed points such as disease-free equilibrium (DFE) and endemic equilibrium (EE) and calculation of basic reproduction number ($\mathcal{R}_0$) (with control and without control) are also presented in the supporting information sections (Appendices 2 and 3), respectively. Then, the existence of the endemic equilibrium point is calculated, and forward-backward and Hopf bifurcation analysis are carried out. The local and global stability of the Disease-Free Equilibrium (DFE) and Endemic Equilibrium (EE) are presented. Additionally, the stability and persistence of solutions are discussed. Fothermore, we presented a variety of numerical examples, elucidating the results of contour plot and box plot analyses, phase plane analysis focusing on the relative influence on $\mathcal{R}_0$, and computational biological findings. Sections titled 'A Case Study of Influenza in Mexico', and 'A Case Study of Influenza in Colombia' conclude by comparing the model solution with the real data analysis utilizing case studies from Mexico, and Colombia. Finally, the outcomes are summarized and briefly discussed.

## Mathematical model formulation of influenza

In this study, we consider a modified version of the typical SEIR model, we propose the following six compartments' potential SVEIRT (Susceptible-Vaccinated-Exposed-Infectious-Treatment-Removal) mathematical model. The SVEIRT model is based on several key assumptions. First, the total population is dynamic, with individuals entering the susceptible class through recruitment at a constant rate $\Lambda$ and exiting all compartments due to natural death at rate $\mu$. The disease is transmitted through contact between susceptible individuals and either exposed $E(t)$ or infectious $I(t)$ individuals, with transmission rates $\beta_1$ and $\beta_2$, respectively. Vaccination occurs at a constant rate $\phi$, and the vaccine provides partial protection, reducing the susceptibility of vaccinated individuals by a factor of $1 - \varepsilon$. Exposed individuals progress to the infectious class at rate $\alpha$, and infected individuals either recover at rate $\gamma$, receive treatment at rate $\gamma_1$, or die due to disease at rate $\delta$. Recovered individuals are assumed to gain permanent immunity, while treated individuals are considered part of the infectious population but receive medical intervention. Finally, all compartments are subject to the same natural death rate $\mu$, and demographic changes are not influenced by disease-related factors other than mortality in the infectious class.

Even though a person is vaccinated against Influenza, they can still acquire the infection due to several factors. The flu vaccine targets specific strains of the virus, but the virus mutates rapidly, leading to variations not covered by the vaccine. This phenomenon, known as "antigenic drift", can result in partial immunity or reduced vaccine effectiveness. Additionally, it takes about two weeks after vaccination for immunity to develop, leaving a window for infection. Lastly, individual immune responses to vaccines can vary, and some people may not build strong enough protection despite being vaccinated [1, 8, 10].

In the SVEIRT model, $S(t)$ represents the susceptible population, which is subject to recruitment at rate $\Lambda$ and can contract the disease through contact with exposed $E(t)$ and infectious $I(t)$ individuals at rates $\beta_1$ and $\beta_2$, respectively. Susceptibles may also be vaccinated at rate $\phi$, or die naturally at rate $\mu$. The vaccinated population $V(t)$ increases as individuals are vaccinated from $S(t)$, but they may still become exposed or infected at reduced rates due to vaccine efficacy $\varepsilon$, while also dying at rate $\mu$. Exposed individuals $E(t)$ become infected at rate $\alpha$ or die at rate $\mu$, having been infected by interaction with $S$ or $V$. Infectious individuals $I(t)$ arise from the exposed class at rate $\alpha$ or through reduced infection from the vaccinated group, and they may either recover at rate $\gamma$, receive treatment at rate $\gamma_1$, or die due to natural and disease-related causes at rates $\mu$ and $\delta$. Recovered individuals $R(t)$ gain immunity, leaving the infectious class at rate $\gamma$, while the treated population $T(t)$ increases due to treatment of infectious individuals at rate $\gamma_1$, with both classes experiencing natural mortality at rate $\mu$.

The waning of vaccine immunity can be reflected in the terms involving $\beta_1$ and $\beta_2$, which represent the transmission rates from exposed ($E$) and infectious ($I$) individuals. Specifically, the factor $(1 - \varepsilon)$ already accounts for the reduced susceptibility of vaccinated individuals. To model waning immunity, $\varepsilon$ could be time-dependent, decreasing over time as immunity wanes. This would result in a gradual increase in susceptibility, thereby increasing the effective transmission rates $\beta_1$ and $\beta_2$ for the vaccinated population as immunity weakens.

Base on the above formulation as a system of ordinary differential equations is as follows:

$$\begin{cases} \dfrac{dS}{dt} = \Lambda - (\beta_1 E + \beta_2 I)S - (\mu + \phi)S. \\[2mm] \dfrac{dV}{dt} = \phi S - (1 - \varepsilon)(\beta_1 E + \beta_2 I)V - \mu V. \\[2mm] \dfrac{dE}{dt} = (\beta_1 E + \beta_2 I)S - (\alpha + \mu)E. \\[2mm] \dfrac{dI}{dt} = \alpha E + (1 - \varepsilon)(\beta_1 E + \beta_2 I)V - (\mu + \delta + \gamma + \gamma_1)I. \\[2mm] \dfrac{dR}{dt} = \gamma I - \mu R. \\[2mm] \dfrac{dT}{dt} = \gamma_1 I - \mu T. \end{cases} \tag{1}$$

for $t \in (0, \infty)$ with initial conditions,

$$S(0) = S_0 \geq 0, V(0) = V_0 \geq 0, E(0) = E_0 \geq 0, I(0) = I_0 \geq 0, R(0) = R_0 \geq 0, \text{ and } T(0) = T_0 \geq 0, \tag{2}$$

and the total population for the SVEIRT model is found by,

$$N(t) \equiv S(t) + V(t) + E(t) + I(t) + R(t) + T(t). \tag{3}$$

The definitions of all state variables and model parameters with brief descriptions are presented in Table 1, and they are non-negative because of the dynamics of a population. The whole cycle and the flow diagram for the suggested model (1) are illustrated in Figs 1 and 2. Assume that the following guidelines control the transmission of disease:

(i). The total population stays fixed at a level $N$ over the interval.

**Table 1. The model parameters values with descriptions.**

| Notation | Definition | Value | Source |
|---|---|---|---|
| $N$ | Total number of human population | | |
| $S$ | Total number of susceptible population | | |
| $V$ | Total number of vaccinated population | | |
| $E$ | Total number of exposed population | | |
| $I$ | Total number of infected population | | |
| $R$ | Total number of recovered population | | |
| $T$ | Total number of treated population | | |
| $\alpha$ | The transition rate from $E$ to $I$ | 0.75 week$^{-1}$ | [3, 11] |
| $\Lambda$ | The recruitment rate in $S$ class | $5 \times 10^2$ week$^{-1}$ | [6, 7] |
| $\beta_1$ | The transmission rate from contact with $E$ to class $S$ | [0.0045, 0.0055] week$^{-1}$ | [8, 10] |
| $\beta_2$ | The transmission rate from contact with $I$ to class $S$ | [0.0045, 0.0055] week$^{-1}$ | [16, 17] |
| $\gamma$ | The recovery rate of $I$ | 0.65 week$^{-1}$ | [9, 28] |
| $\gamma_1$ | The treatment progression rate of $I$ | 0.25 week$^{-1}$ | [4, 5] |
| $\mu$ | The natural death rate | $5 \times 10^{-2}$ week$^{-1}$ | [3, 4] |
| $\delta$ | The disease induced death rate | 0.3 week$^{-1}$ | [9, 11] |
| $\lambda = (1 - \varepsilon)$ | The vaccine inefficiency rate | 0.55 | [6, 9] |
| $\phi$ | The rate of progression to the vaccinated class $V$ | 0.3 week$^{-1}$ | [1, 2] |

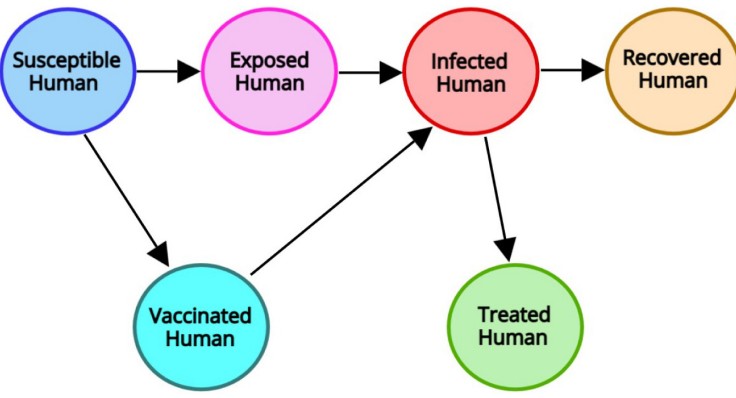

**Fig 1. The transmission cycle of disease in the SVEIRT model.**

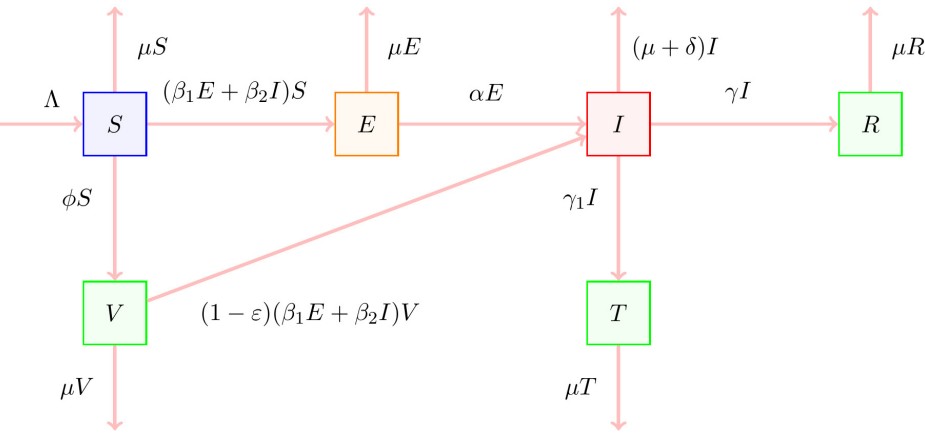

**Fig 2. The compartmental diagram for SVEIRT model.**

(ii). The susceptible population becomes infected by contracting the disease. The rate of change of the susceptible population is proportional to the number of contracts between the $S(t)$ and $E(t)$ also between the $S(t)$ and $I(t)$. This number is proportional to the number of susceptible, exposed, and infected persons.

We have stated a few auxiliary results, and the supporting section contains the proofs of the existence, positivity, and boundedness of the solution.

## Determination of fixed points

To identify the equilibrium points $(\tilde{S}, \tilde{V}, \tilde{E}, \tilde{I}, \tilde{R}, \tilde{T})$ of the system (1), we set the value of each derivative to zero. Thus, in situations of equilibrium, we obtain,

$$
\begin{cases}
\Lambda - (\beta_1 \tilde{E} + \beta_2 \tilde{I})\tilde{S} - (\mu + \phi)\tilde{S} = 0. \\
\phi\tilde{S} - (1-\varepsilon)(\beta_1 \tilde{E} + \beta_2 \tilde{I})\tilde{V} - \mu\tilde{V} = 0. \\
(\beta_1 \tilde{E} + \beta_2 \tilde{S})\tilde{S} - (\alpha + \mu)\tilde{E} = 0. \\
\alpha\tilde{E} + (1-\varepsilon)(\beta_1 \tilde{E} + \beta_2 \tilde{I})\tilde{V} - (\mu + \delta + \gamma + \gamma_1)\tilde{I} = 0. \\
\gamma\tilde{I} - \mu\tilde{R} = 0. \\
\gamma_1\tilde{I} - \mu\tilde{T} = 0.
\end{cases}
\tag{4}
$$

Now, we have to solve the right side of the equations (4) in the derivative terms to find steady states, and by solving the consequent equations for compartments $S$, $V$, $E$, $I$, $R$, and $T$, we discover that there are only two biologically significant equilibria in total. These events can be categorized into two scenarios: either the influenza virus becomes extinct in the area $E = I = 0$ or persisting within the region ($E \neq 0, I \neq 0$).

### The disease-free equilibrium (DFE) point

The disease-free equilibrium (DFE) point of (1) is, $\mathcal{E}^0 \equiv \left(\frac{\Lambda}{\mu+\phi}, \frac{\phi\Lambda}{\mu(\mu+\phi)}, 0, 0, 0, 0\right)$.

At disease-free equilibrium point $S = \frac{\Lambda}{\mu+\phi}$. Here, $\Lambda$ represents the recruitment rate, assumed proportional to the total population, hence $\Lambda = \mu N$. After simplification DFE becomes $\mathcal{E}^0 \equiv \left(\frac{\mu N}{\mu+\phi}, \frac{\phi N}{\mu+\phi}, 0, 0, 0, 0\right)$.

### The endemic equilibrium (EE) point

The endemic equilibrium (EE) point for the system (1) is, $\mathcal{E}^* = (S^*, V^*, E^*, I^*, R^*, T^*)$. Here,
$S^* = \frac{\Lambda - (\alpha + \mu)E^*}{(\mu + \phi)}$, $V^* = \frac{\phi(\Lambda - a_1 E^*)}{a_2(\mu + \lambda\lambda_1)}$, $R^* = \frac{\lambda I^*}{\mu}$, $T^* = \frac{\lambda_1 I^*}{\mu}$,

$E^* = \frac{(\Lambda\beta_1 - a_1 a_2 - a_1\beta_2 I^*) \pm \sqrt{(\Lambda\beta_1 - a_1 a_2 - a_1\beta_2 I^*)^2 + 4\Lambda\beta_2 I^* a_1\beta_1}}{2a_1\beta_1}$.

Where, $a_1 = \alpha + \mu$, $a_2 = \mu + \phi$, $\lambda_1 = \beta_1 E^* + \beta_2 I^*$, and $\lambda = 1 - \epsilon$.

A concise analysis of the endemic equilibrium point can be found in the supporting section.

## Basic reproduction number and the presence of endemic equilibrium

The fundamental reproduction number, a crucial threshold quantity for examining infectious disease modeling, has been calculated in this section. To mathematically quantify the volatility of an infectious disease, it was developed for the study of epidemiology. It establishes whether

the disease will disappear over time or remain prevalent in the community. This threshold quantity, usually denoted by $\mathcal{R}_0$, is defined as the expected number of secondary infections resulting from a single primary infection in a population where everyone is susceptible. When $\mathcal{R}_0 > 1$, referring that one primary infection can lead to several subsequent infections, the DFE becomes unstable, causing an epidemic in the population. Conversely, if $\mathcal{R}_0 < 1$, the DFE is locally asymptotically stable, preventing the disease from persisting in the community [22]. That reflects the scenario that the situation is under control. Therefore, an efficient plan should be created as soon as a pandemic emerges to ensure that the reproduction number falls to less than zero. Since, the considered model (1) has DFE, $\mathcal{E}^0 \equiv \left( \frac{\Lambda}{\mu+\phi}, \frac{\phi\Lambda}{\mu(\mu+\phi)}, 0, 0, 0, 0 \right)$; hence, $\mathcal{R}_0$ can be calculated analytically. In this section, we have employed the next-generation matrix method to determine the basic reproduction number for the Influenza model presented in (1). The calculation is based on the formula $\mathcal{R}_0 = \rho(FV^{-1})$, that represents the spectral radius of $FV^{-1}$.

Succinct computation is shown in the supporting section.

## Basic reproduction number with control

The threshold value for the system (1) associated with controlling strategies can be presented as follows,

$$\mathcal{R}_{0V} = \frac{\Lambda[\alpha\beta_2 + \beta_1(\mu + \delta + \gamma + \gamma_1)]}{(\mu+\phi)(\alpha+\mu)(\mu+\delta+\gamma+\gamma_1)} + \frac{\Lambda\phi\beta_2\lambda}{\mu(\mu+\phi)(\mu+\delta+\gamma+\gamma_1)}. \tag{5}$$

## Basic reproduction number without control

The fundamental reproduction number for the system (1) in the absence of control strategies can be expressed as follows,

$$\mathcal{R}_0 = \frac{\Lambda[\alpha\beta_2 + \beta_1(\gamma + \gamma_1 + \mu + \delta)]}{(\mu+\phi)(\alpha+\mu)(\gamma+\gamma_1+\delta+\mu)}. \tag{6}$$

## Presence of endemic equilibrium

The existence of the EE point and its uniqueness is dependent upon the corresponding threshold number $\mathcal{R}_0 > 1$. There endures a solitary EE $\mathcal{E}^* \equiv (S^*, V^*, E^*, I^*, R^*, T^*)$ for the model (1). From system (25) we have,

$$S^* = \frac{\Lambda}{\lambda_1 + \mu + \phi}, \ R^* = \frac{\lambda I^*}{\mu}, \ \text{and} \ \ T^* = \frac{\lambda_1 I^*}{\mu}.$$

Now, from second equation of (25),

$$\phi S^* = (\lambda\lambda_1 + \mu)V^*.$$

$$V^* = \frac{\phi S}{\lambda\lambda_1 + \mu} = \frac{\Lambda\phi}{(\lambda\lambda_1 + \mu)(\lambda_1 + \mu + \phi)}.$$

$$V^* = \frac{\Lambda\phi}{\lambda\lambda_1^2 + \mu\lambda_1 + \mu\lambda\lambda_1 + \mu^2 + \phi\lambda\lambda_1 + \phi\mu}.$$

Let, $p = \lambda\lambda_1^2 + \mu\lambda_1 + \mu\lambda\lambda_1 + \mu^2 + \phi\lambda\lambda_1 + \phi\mu$.

Now, adding the second and fourth equation of (25) we have,

$$\phi S^* - \mu V^* + \alpha E^* - (\mu + \delta + \gamma + \gamma_1)I^* = 0$$

$$\Rightarrow \frac{\phi \Lambda}{\lambda_1 + \mu + \phi} - \frac{\mu \phi \Lambda}{p} + \alpha E^* - (\mu + \delta + \gamma + \gamma_1)I^* = 0$$

$$\Rightarrow I^* = \frac{E^* \alpha}{(\mu + \delta + \gamma + \gamma_1)} - \frac{\alpha \mu \phi \Lambda}{\alpha p(\mu + \delta + \gamma + \gamma_1)} + \frac{\phi \Lambda \alpha}{\alpha(\lambda_1 + \mu + \phi)(\mu + \delta + \gamma + \gamma_1)}.$$

Now, from third equation of (25) we have get, $E^* = \frac{\Lambda \lambda_1}{(\lambda_1 + \mu + \phi)(\alpha + \mu)}$. Substituting this in above expression we have,

$$I^* = \frac{\Lambda \lambda_1 \alpha}{a_1(\alpha + \mu)(\lambda_1 + \mu + \phi)} - \frac{\alpha \mu \phi \Lambda}{a_1 \alpha p} + \frac{\phi \Lambda \alpha}{\alpha(\lambda_1 + \mu + \phi)a_1}.$$

Let, $a_1 = (\mu + \delta + \gamma + \gamma_1)$. From (33) we have, $R_0 = \frac{\Lambda(\alpha \beta_2 + \beta_1 a_1)}{(\mu + \phi)(\alpha + \mu)a_1}$. Now,

$$I^* = \frac{\Lambda \lambda_1 \alpha^2 p - (\alpha + \mu)(\lambda_1 + \mu + \phi) + p(\alpha + \mu)\phi \alpha \Lambda}{a_1(\alpha + \mu)(\lambda_1 + \mu + \phi)\alpha p}$$

$$= \frac{\Lambda \lambda_1 \alpha^2 p}{a_1(\alpha + \mu)(\lambda_1 + \mu + \phi)\alpha p} + \frac{\left(\left(\frac{\Lambda(\alpha \beta_2 + \beta_1 a_1)}{(\mu + \phi)(\alpha + \mu)a_1}\right) - 1\right)p\phi\{1 - (\mu + \phi + \lambda_1)\}(\mu + \phi)}{(\mu + \phi + \lambda_1)}$$

$$= \frac{\Lambda \lambda_1 \alpha^2 p}{a_1(\alpha + \mu)(\lambda_1 + \mu + \phi)\alpha p} + \frac{(\mathcal{R}_0 - 1)p\phi\{1 - (\mu + \phi + \lambda_1)\}(\mu + \phi)}{(\mu + \phi + \lambda_1)}.$$

Let, the infectious force at the endemic steady state,

$$\lambda_1^* = (\beta_1 E^* + \beta_2 I^*)$$

$$\Rightarrow \lambda_1^* = \frac{\beta_1 \Lambda \lambda_1}{(\lambda_1 + \mu + \phi)(\alpha + \mu)} + \frac{\beta_2(\Lambda \alpha^2 p \lambda_1 - (\alpha + \mu)(\lambda_1 + \mu + \phi)\alpha \mu \phi \Lambda + p(\alpha + \mu)\phi \Lambda \alpha)}{a_1 \alpha p(\alpha + \mu)(\lambda_1 + \mu + \phi)}$$

$$\Rightarrow \lambda_1^* = \frac{a_1 \alpha p \beta_1 \Lambda \lambda_1 + \Lambda \alpha^2 p \lambda_1 - (\alpha + \mu)(\lambda_1 + \mu + \phi)\alpha \mu \phi \Lambda + p(\alpha + \mu)\phi \Lambda \alpha}{a_1 \alpha p(\alpha + \mu)(\lambda_1 + \mu + \phi)}$$

$\Rightarrow \quad \lambda_1^* a_1 \alpha p \{\lambda_1^*(\alpha + \mu) + (\mu + \phi)(\alpha + \mu)\} = p[a_1 \alpha \beta_1 \Lambda \lambda_1^* + \lambda_1^* \Lambda \alpha^2 + (\alpha + \mu)\phi \Lambda \alpha] - \alpha \mu \phi \Lambda \{\lambda_1^*(\alpha + \mu) + (\mu + \phi)(\alpha + \mu)\}$

$\Rightarrow \quad \lambda_1^* a_1 \alpha \{\lambda_1^{*2} \lambda + \lambda_1^*(\mu + \lambda(\mu + \phi)) + \mu(\mu + \phi)\}[\lambda_1^*(\alpha + \mu) + (\mu + \phi)(\alpha + \mu)] = \{\lambda_1^{*2} \lambda + \lambda_1^*(\mu + \lambda(\mu + \phi)) + \mu(\mu + \phi)\}[a_1 \alpha \beta_1 \Lambda \lambda_1^* + \lambda_1^* \Lambda \alpha^2 + (\alpha + \mu)\phi \Lambda \alpha] - \lambda_1^*(\alpha + \mu)\alpha \mu \phi \Lambda - (\alpha + \mu)(\mu + \phi)\alpha \mu \phi \Lambda$

$\Rightarrow \quad [\lambda_1^{*3} a_1 \alpha \lambda + \lambda_1^{*2} a_1 \alpha(\mu + \lambda(\mu + \phi)) + \lambda_1^* a_1 \alpha \mu(\mu + \phi)][\lambda_1^*(\alpha + \mu) + (\mu + \phi)(\alpha + \mu)] = \lambda_1^{*3} \lambda a_1 \alpha \beta_1 \Lambda + \lambda_1^{*3} \lambda \Lambda \alpha^2 + \lambda_1^{*2} \lambda \alpha \Lambda \phi(\alpha + \mu) + \lambda_1^{*2} \Lambda \alpha a_1 \beta_1(\mu + \lambda(\mu + \phi)) + \lambda_1^{*2}(\mu + \lambda(\mu + \phi))\Lambda \alpha^2 + \lambda_1^*(\mu + \lambda(\mu + \phi))(\alpha + \mu)\phi \Lambda \alpha + \mu(\mu + \phi)a_1 \alpha \beta_1 \Lambda \lambda_1^* + \mu(\mu + \phi)\lambda_1^* \Lambda \alpha^2 + (\mu + \phi)\mu(\alpha + \mu)\phi \Lambda \alpha - \lambda_1^*(\alpha + \mu)\alpha \mu \phi \Lambda - (\alpha + \mu)(\mu + \phi)\alpha \mu \phi \Lambda$

$\Rightarrow \quad \lambda_1^{*4}(\alpha + \mu)a_1 \alpha \lambda + \lambda_1^{*3} a_1 \alpha(\alpha + \mu)(\mu + \lambda(\mu + \phi)) + \lambda_1^*(\alpha + \mu)a_1 \alpha \mu(\mu + \phi) + \lambda_1^{*3} a_1 \alpha \lambda(\mu + \phi)(\alpha + \mu) + \lambda_1^{*2} a_1 \alpha(\mu + \lambda(\mu + \phi))(\mu + \phi)(\alpha + \mu) + \lambda_1^* a_1 \alpha \mu(\mu + \phi)^2(\alpha + \mu) = \lambda_1^{*3}[\lambda a_1 \alpha \beta_1 \Lambda + \lambda \Lambda \alpha^2] + \lambda_1^{*2}[\lambda \alpha \Lambda \phi(\alpha + \mu) + \Lambda \alpha a_1 \beta_1(\mu + \lambda(\mu + \phi)) + (\mu + \lambda(\mu + \phi))\Lambda \alpha^2] + \lambda_1^*[(\mu + \lambda(\mu + \phi))(\alpha + \mu)\phi \Lambda \alpha + \mu(\mu + \phi)a_1 \alpha \beta_1 \Lambda + \mu(\mu + \phi)\Lambda \alpha^2 + (\alpha + \mu)\alpha \mu \phi \Lambda] + (\mu + \phi)\mu(\alpha + \mu)\phi \Lambda \alpha - (\mu + \phi)(\alpha + \mu)\alpha \mu \phi \Lambda$

$\Rightarrow \quad \lambda_1^{*4}(\alpha + \mu)a_1 \alpha \lambda + \lambda_1^{*3}[a_1 \alpha(\alpha + \mu)(\mu + \lambda(\mu + \phi)) + a_1 \alpha \lambda(\mu + \phi)(\mu + \alpha) - \lambda a_1 \alpha \beta_1 \Lambda + \lambda \Lambda \alpha^2] + \lambda_1^{*2}[a_1 \alpha(\mu + \lambda(\mu + \phi))(\mu + \phi)(\alpha + \mu) - \lambda \alpha \Lambda \phi(\alpha + \mu) - \Lambda \alpha a_1 \beta_1(\mu + \lambda(\mu + \phi)) - (\mu + \lambda(\mu + \phi))\Lambda \alpha^2] + \lambda_1^*[(\alpha + \mu)(\mu + \phi)a_1 \alpha \mu + a_1 \alpha \mu(\mu + \phi)^2(\alpha + \mu) - (\mu + \lambda(\mu + \phi))(\alpha + \mu)\phi \Lambda \alpha - \mu(\mu + \phi)a_1 \alpha \beta_1 \Lambda - \mu(\mu + \phi)\Lambda \alpha^2 - (\alpha + \mu)\alpha \mu \phi \Lambda] - (\mu + \phi)(\alpha + \mu)\mu \phi \Lambda \alpha - (\mu + \phi)(\alpha + \mu)\alpha \mu \phi \Lambda = 0.$

Here, the basic reproduction number,

$$\mathcal{R}_0 = \frac{\Lambda[\alpha \beta_2 + \beta_1(\gamma + \gamma_1 + \mu + \delta)]}{(\alpha + \mu)(\mu + \phi)(\mu + \delta + \gamma + \gamma_1)}.$$

$$1 - \mathcal{R}_0 = \frac{(\alpha + \mu)(\mu + \phi)a_1 - \Lambda[\alpha \beta_2 + \beta_1 a_1]}{(\alpha + \mu)(\mu + \phi)a_1}.$$

Thus, substituting the above formulas into the infectious force, at a steady state, we have the polynomial in the form,

$$b_4 \lambda_1^4 + b_3 \lambda_1^3 + b_2 \lambda_2 + b_1 \lambda_1 + b_0 = 0.$$

Where,

$b_0 = (\mu + \phi)(\alpha + \mu)\mu \phi \Lambda \alpha - (\mu + \phi)(\alpha + \mu)\alpha \mu \phi \Lambda = (1 - \mathcal{R}_0)(\alpha + \mu)(\mu + \phi)a_1 \alpha \mu \phi.$

$b_1 = \left[\dfrac{(\alpha + \mu)(\mu + \phi)a_1 - \Lambda[\alpha \beta_2 + \beta_1 a_1]}{(\alpha + \mu)(\mu + \phi)a_1}\right]\mu \alpha(\alpha + \mu)(\mu + \phi)a_1 + (\alpha + \mu)(\mu + \phi)a_1 \alpha \mu(1 - \mathcal{R}_{0V}) - \mu(\mu + \phi)\Lambda \alpha^2 - (\alpha + \mu)\alpha \mu \phi \Lambda$

$\quad = (1 - \mathcal{R}_0)\mu \phi + (1 - \mathcal{R}_{0V})\alpha \mu - \mu(\mu + \phi)\Lambda \alpha^2 - (\alpha + \mu)\alpha \mu \phi \Lambda.$

$b_2 = \alpha(\mu + \lambda(\mu + \phi))(\alpha + \mu)(\mu + \phi)a_1(1 - \mathcal{R}_0) - \lambda \alpha \Lambda \phi(\alpha + \mu) - (\mu + \lambda(\mu + \phi))\Lambda \alpha^2.$

$b_3 = a_1 \alpha(\alpha + \mu)(\mu + \lambda(\mu + \phi)) + \alpha \lambda a_1(\alpha + \mu)(\mu + \phi)(1 - \mathcal{R}_0) + \lambda \Lambda \alpha^2.$

$b_4 = (\alpha + \mu)a_1 \alpha \lambda.$

After calculating $\lambda_1^*$ from the polynomial and inserting positive values of $\lambda_1^*$ in the formulas of $S^*$, $V^*$, $E^*$, $I^*$, $R^*$, and $T^*$, the components of the endemic equilibrium point (EEP) are determined. Moreover, it follows from the expression of polynomial coefficients that $b_4$ is always positive, and $b_0$, $b_1$, $b_2$, $b_3$ are positive (negative) if $\mathcal{R}_0$ and $\mathcal{R}_{0V}$ is less (greater) than one. Hence, the following outcomes can be deduced,

- Four or two endemic equilibria if $b_2 > 0$, $b_3 < 0$, $b_4 > 0$, and $\mathcal{R}_0 < 1$, $\mathcal{R}_{0V} < 1$.

- Two endemic equilibria if $b_2 > 0$, $b_3 > 0$, $b_4 < 0$, and $\mathcal{R}_0 < 1$, $\mathcal{R}_{0V} < 1$.

- No endemic equilibria otherwise if $\mathcal{R}_0 < 1$ and $\mathcal{R}_{0V} < 1$.

Hence, by applying the Descartes rule of signs [29, 30], the EE point of the model (1), $\mathcal{E}^*$ exists iff $\mathcal{R}_0 > 1$.

## Forward and backward bifurcation analysis

We proceed to establish the bifurcation condition of the equilibrium point that exists. In Section, we have obtained the DFE point. We will now define the characteristics of the DFE point, $\mathcal{E}^0$. We have observed that the DFE state is locally asymptotically stable if the $\mathcal{R}_0 < 1$ and unstable if the $\mathcal{R}_0 > 1$ [31, 32]. It reveals that when $\mathcal{R}_0 = 1$, the preceding analysis becomes ineffective. The crucial factor $\mathcal{R}_0 = 1$ is equivalent to,

$$\beta_2 = \beta_2^{[TC]} = \frac{(\mu + \phi)(\alpha + \mu)(\mu + \delta + \gamma + \gamma_1) - \mu N \beta_1(\mu + \delta + \gamma + \gamma_1)}{\phi N \lambda(\alpha + \mu) + \mu N \alpha}.$$

In the upcoming theorem, we will depict that the model system (1) undergoes the transcritical bifurcation (TC) at the DFE point $\mathcal{E}^0$ when the critical parameter $\beta_2$ reaches its critical value $\beta_2 = \beta_2^{[TC]}$.

**Theorem 1**. *Transcritical bifurcation of the system (1) obtains at the point of no disease $(\mathcal{E}_0)$ when model parameter $\beta_2$ goes through the critical value $\beta_2 = \beta_2^{[TC]}$.*

*Proof.* When $\beta_2 = \beta_2^{[TC]}$, one of the eigenvalues becomes zero, causing the collapse of standard eigen-method analysis. In such cases, we employ Somtomayor's Theorem [27, 33] to examine the characteristics of the DFE point. Let $V$ and $W$ represent the eigenvectors with respect to the zero eigenvalue of $J(\mathcal{E}^0)$ and $J[(\mathcal{E}^0)]^T$, respectively. So,

$$V = \begin{pmatrix} -\dfrac{(\mu + \delta + \gamma + \gamma_1)(\alpha + \mu)}{\alpha(\mu + \phi)} \\ \dfrac{\mu + \delta + \gamma + \gamma_1}{\alpha} \\ 1 \\ \dfrac{\delta}{\mu} \end{pmatrix}, \text{ and } W = \begin{pmatrix} 0 \\ 2 \\ \dfrac{(\mu + \phi)(\mu + \alpha) - \beta_1 \Lambda}{\alpha(\mu + \phi)} \\ 0 \end{pmatrix}.$$

Considering the sub-model of the above model (1), by taking the SEIR compartment we have,

$$F = \begin{pmatrix} \Lambda - (\beta_1 E + \beta_2 I)S - (\mu + \phi)S \\ (\beta_1 E + \beta_2 I)S - (\alpha + \mu)E \\ \alpha E - (\mu + \delta + \gamma + \gamma_1)I \\ \gamma I - \mu R \end{pmatrix}.$$

Next, we examine the system's dynamic behavior by methodically adjusting the parameters close to each equilibrium point. We adapt Sotomayor's theorem for local bifurcation analysis [33]. The modified theorem states that the Jacobian matrix of the modified SEIR system at the DFE point $\mathcal{E}^0$ appears transcritical bifurcation.

It is demonstrated that the Jacobian matrix of the system $(\mathcal{E}^0, \beta^* = \beta_2)$ can be evaluated as $J = Df(\mathcal{E}^0, \beta^*)$. Here,

$$J = \begin{pmatrix} -(\mu + \phi) & -\beta_1 S & -\beta_2^* S & 0 \\ 0 & \beta_1 S - (\alpha + \mu) & \beta_2 S & 0 \\ 0 & \alpha & -(\mu + \delta + \gamma + \gamma_1) & 0 \\ 0 & 0 & \gamma & -\mu \end{pmatrix}.$$

Where,

$$\beta_2^* = \frac{(\mu + \phi)(\alpha + \mu)(\mu + \delta + \gamma + \gamma_1) - \mu N \beta_1 (\mu + \delta + \gamma + \gamma_1)}{\phi N \lambda (\alpha + \mu) + \mu N \alpha}.$$

From the Jacobian matrix, the third $\lambda_R$ eigenvalue $\lambda_I$ in the direction of $I$ is $-(\mu + \delta + \gamma + \gamma_1)$ while the $\lambda_S$ and $\lambda_R$ are negative. Further, the eigenvector $V = (v_1, v_2, v_3, v_4)^T$ corresponding to the $\lambda_I$ satisfying the condition $Jz = \lambda z$ then $Jz = 0$ gives,

$$\begin{pmatrix} -(\mu + \phi) & -\beta_1 S & -\beta_2^* S & 0 \\ 0 & \beta_1 S - (\alpha + \mu) & \beta_2 S & 0 \\ 0 & \alpha & -(\mu + \delta + \gamma + \gamma_1) & 0 \\ 0 & 0 & \gamma & -\mu \end{pmatrix} \begin{pmatrix} v_1 \\ v_2 \\ v_3 \\ v_4 \end{pmatrix} = \begin{pmatrix} 0 \\ 0 \\ 0 \\ 0 \end{pmatrix}.$$

From which we get,

$$-(\mu + \phi)v_1 - \beta_1 S v_2 - \beta_2^* S v_3 = 0$$
$$(\beta_1 E + \beta_2 I)v_1 + \{\beta_1 S - (\alpha + \mu)\}v_2 + \beta_2 S v_3 = 0$$
$$0 + \alpha v_2 - (\mu + \delta + \gamma + \gamma_1)v_3 = 0$$
$$\gamma v_3 - \mu v_4 = 0.$$

The finding of the above system of equations is,

$$V = (v_1, v_2, v_3, v_4)^T$$
$$= \left( \frac{\beta_1 S(\mu + \delta + \gamma + \gamma_1)\mu}{-(\alpha + \mu)\alpha\gamma}v_4 - \frac{\beta_2^* S \mu}{\gamma(\alpha + \mu)}v_4, \; \frac{(\mu + \delta + \gamma + \gamma_1)\mu}{\alpha\gamma}v_4, \; \frac{\mu v_4}{\gamma}, \; v_4 \right).$$

Similarly, the eigenvector $W = (w_1, w_2, w_3, w_4)^T$ can be written as,

$$J^T w = \begin{pmatrix} -(\mu + \phi) & \beta_1 E + \beta_2 I & 0 & 0 \\ -\beta_1 S & \beta_1 S - (\alpha + \mu) & \alpha & 0 \\ -\beta_2^* S & \beta_2 S & -(\mu + \delta + \gamma + \gamma_1) & \gamma \\ 0 & 0 & 0 & -\mu \end{pmatrix} \begin{pmatrix} w_1 \\ w_2 \\ w_3 \\ w_4 \end{pmatrix} = 0.$$

We have the solutions,

$$w_4 = 0, \quad w_1 = \frac{(\beta_1 E + \beta_2 I)}{(\mu + \phi)} w_2, \text{ and}$$

$$w_3 = \frac{\beta_1 S(\beta_1 E + \beta_2 I) + (\alpha + \mu)(\mu + \phi)}{\alpha(\mu + \phi)} w_2.$$

Where $w_2$ is a free variable. Now, it is possible to write the simplified SEIR system in vector form,

$$\frac{dX}{dt} = f(X).$$

Here, $X = (S, E, I, R)^T$ and $F = (F_1, F_2, F_3, F_4)^T$ with $F_i (i = 1, 2, 3, 4)$, then calculate the $\frac{dF}{d\beta_2} = F_{\beta_2}$. From which we can get that,

$$F_\beta^* = \begin{pmatrix} -SI \\ SI \\ 0 \\ 0 \end{pmatrix}.$$

Then,

$$F_\beta^*(\mathcal{E}^0, \beta^*) = \begin{pmatrix} 0 \\ 0 \\ 0 \\ 0 \end{pmatrix}, \quad w^T \cdot F_\beta(\mathcal{E}^0, \beta^*) = 0.$$

$$DF_\beta(\mathcal{E}^0, \beta^*) = \begin{pmatrix} 0 & -S_1 & 0 & 0 \\ 0 & S_1 & 0 & 0 \\ 0 & 0 & 0 & 0 \\ 0 & 0 & 0 & 0 \end{pmatrix}, \quad \text{where } S_1 = \frac{\pi}{\mu}.$$

$$w^T \cdot [DF_\beta(\mathcal{E}^0, \beta^*) \cdot z] = w_2 v_2 S_1 \neq 0.$$

Based on Sotomayor's theorem, when the parameter $\beta_2$ gets over the bifurcation value $\beta_2^{[TC]}$, the transcritical bifurcation arises at the DFE. Then according to the [17] we have obtained,

$$W^T F_{\beta_2}\big|_{\mathcal{E}^0, \beta_2 = \beta_2^{[TC]}} = 0$$

$$W^T DF_{\beta_2}\big|_{\mathcal{E}^0, \beta_2 = \beta_2^{[TC]}} V = -\frac{\Lambda(\mu + \delta + \gamma + \gamma_1)}{\alpha(\mu + \phi)} \neq 0$$

$$W^T D^2 F_{\beta_2}\big|_{\mathcal{E}^0, \beta_2 = \beta_2^{[TC]}} (V, V) = \frac{-2(\mu + \delta + \gamma + \gamma_1)^2 (\mu + \alpha)\beta_1}{\alpha^2(\mu + \phi)} \neq 0$$

Consequently, the framework undergoes transcritical bifurcation as the rate of infection $I$ class $(\beta_2)$ exceeds the critical value $\beta_2 = \beta_2^{[TC]}$. There endures a critical infection rate for the $I$ class, beyond which endemic diseases spread throughout the population. Below this threshold, the disease is easily manageable.

The force of infection, $\lambda_1 = \beta_1 E + \beta_2 I$, is proportional to the density of $E$ and $I$ compartments as well as transmission rates $\beta_1, \beta_2$. It indicates a new infection term based on the interaction of the susceptible class with exposed and infected populations in a rate $\beta_1$ and $\beta_2$, respectively. This term has a positive influence on the measurement of the disease density. In a transcritical bifurcation of the basic reproduction number ($\mathcal{R}_0$), the dynamics of the susceptible population versus the force of infection ($\lambda_1$) are as follows:

Before the bifurcation point, $\mathcal{R}_0$ is below a critical threshold, and the force of infection is low, leading to a gradual decrease in the susceptible population $S(t)$. Fig 3(a) shows that when $\lambda_1$ ranges from 0 to 0.75, $S(t)$ remains relatively stable. After $\lambda_1$ exceeds 0.75, $S(t)$ starts to decline. At the bifurcation point, $\mathcal{R}_0$ reaches its critical threshold, causing an abrupt increase in $\lambda_1$ and a rapid rise in infection rates. This can lead to an accelerated spread of the disease. After the bifurcation, the high force of infection causes the susceptible population to decrease more quickly, with a significant increase in the number of infected individuals, potentially

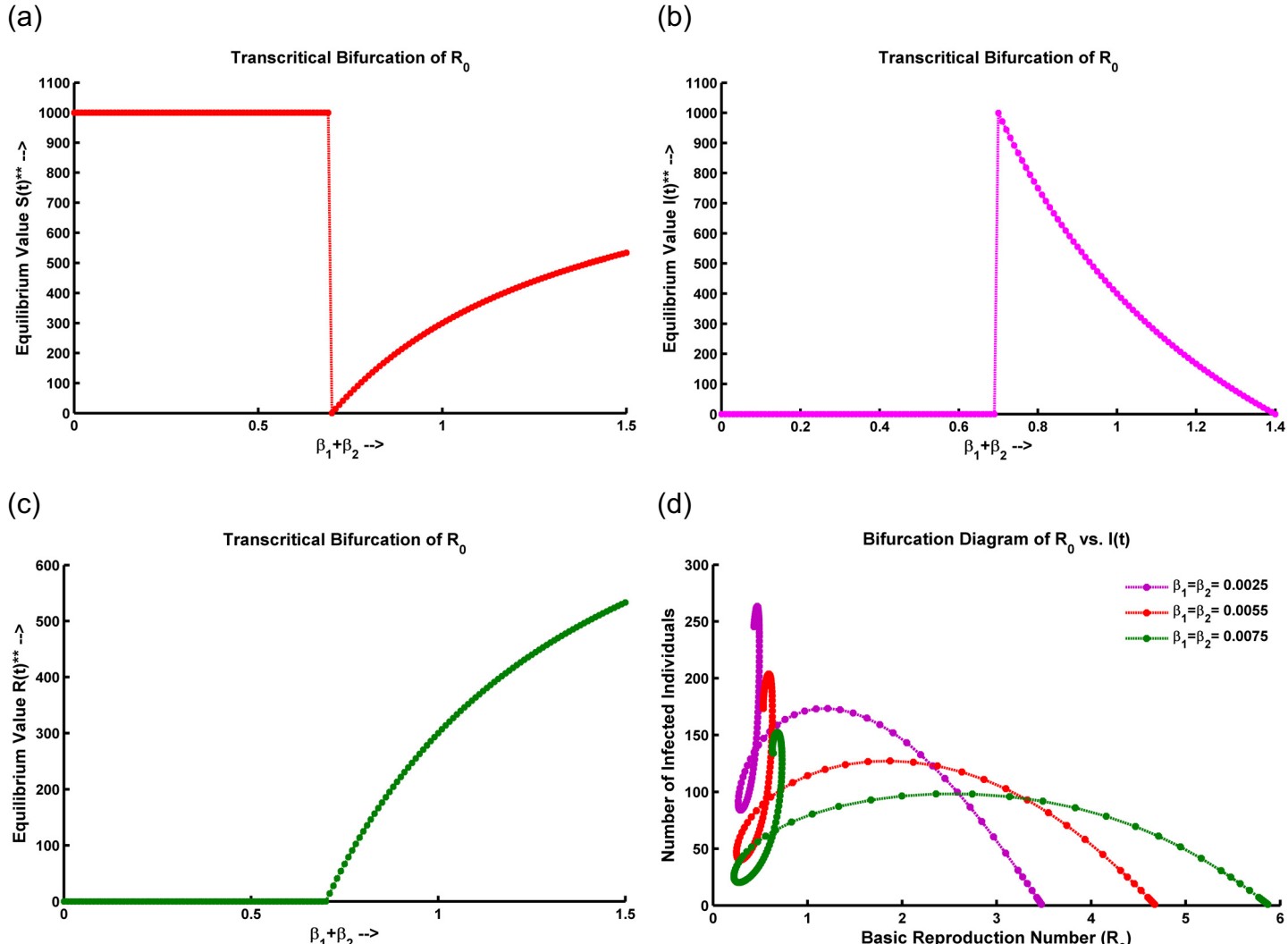

**Fig 3. Transcritical bifurcation occurs in the (a) $S(t)$ compartment, (b) $I(t)$ compartment, (c) $R(t)$ compartment, and (d) bifurcation of $I(t)$ concerning $\mathcal{R}_0$ at the DFE point, with all parameters sourced from Table 1.**

resulting in an epidemic. The system then stabilizes at a new equilibrium with a higher number of infected individuals compared to the pre-bifurcation phase.

From Fig 3(b), the transcritical bifurcation of the fundamental reproduction number ($\mathcal{R}_0$) is an essential point in a disease model where the behavior of the infected population changes. At this point, the rate at which susceptible individuals get infected, known as the force of infection, is crucial. Before this bifurcation, if the force of infection is low (between 0 and 0.65), the infected population stays small or may disappear entirely, meaning the disease can't sustain itself. At the bifurcation point, the force of infection reaches critical levels. Beyond this level, the infection rate is high enough to maintain disease transmission, causing the infected population to grow. After the bifurcation, when the infection rate is above the critical threshold (0.65 to 1.4), the infected population either grows rapidly or stabilizes at a higher level. That makes the disease endemic with ongoing transmission and a large portion of the population remaining infected.

During a transcritical bifurcation of the threshold value ($\mathcal{R}_0$), the behavior of the recovered population versus the force of infection ($\lambda_1$) is described in Fig 3(c). When the force of infection is low (0 to 0.65), the recovered population $R(t)$ remains low, indicating that few people have contracted and recovered from the disease, suggesting inefficient disease spread. At a critical threshold, denoted as $\mathcal{R}_0^*$, the disease moves from a non-endemic state to an endemic state, becoming self-sustaining. As $\lambda_1$ exceeds 0.65, the recovered population rises exponentially, showing efficient disease spread and an increasing number of recoveries over time.

Fig 3(d) depicts the relationship between the infected population and the basic reproduction number ($\mathcal{R}_0$). When $\mathcal{R}_0 > 1$, each infected person spreads the disease to more than one individual, potentially leading to a large-scale outbreak. When $\mathcal{R}_0 = 1$, the infected population remains stable, indicating an endemic state where the disease persists without significant growth or decline. When $\mathcal{R}_0 < 1$, the infected population declines as each person infects fewer than one individual, leading to the epidemic subsiding and eventually dying out. Thus, $I(t)$ increases with rising $\mathcal{R}_0$, highlighting the basic reproduction number as a major determinant of infectious disease behavior, indicating outbreak potential, transmission sustainability, and the effectiveness of control measures.

From Fig 3, it is noticeable that when $\mathcal{R}_0 < 1$, the system described by (1) shows only a stable disease-free equilibrium point. Conversely, when $\mathcal{R}_0 > 1$, a stable endemic equilibrium occurs, leading to the instability of the DFE. This instability, denoting a transition from stable to unstable, happens precisely at the critical point $\mathcal{R}_0 = 1$, leading to a transcritical bifurcation at DFE points. Therefore, when the model parameter $\beta_2$ exceeds its critical value $\beta_2^{[TC]}$, the stability of the disease-free equilibrium shifts from a stable state to an unstable one.

In the next part, we explore the phenomenon of backward bifurcation of the modified system of (1) as follows,

$$
\begin{aligned}
f_1 = x_1' &= \Lambda - (\beta_1 x_3 + \beta_2 x_4)x_1 - (\mu + \phi)x_1 + \phi_1 x_2. \\
f_2 = x_2' &= \phi x_1 - (1 - \varepsilon)(\beta_1 x_3 + \beta_2 x_4)x_2 - (\mu + \phi_1)x_2. \\
f_3 = x_3' &= (\beta_1 x_3 + \beta_2 x_4)x_1 - (\alpha + \mu)x_3. \\
f_4 = x_4' &= \alpha x_3 + (1 - \varepsilon)(\beta_1 x_3 + \beta_2 x_4)x_2 - (\mu + \delta + \gamma + \gamma_1)x_4. \\
f_5 = x_5' &= \gamma x_4 - \mu x_5. \\
f_6 = x_6' &= \gamma_1 x_4 - \mu x_6.
\end{aligned}
\tag{7}
$$

Where, $(S, V, E, I, R, T) = (x_1, x_2, x_3, x_4, x_5, x_6)$. Here, we have considered that, after vaccination, a portion of the vaccinated population progresses to susceptible compartments by losing immunity at a constant rate $\phi_1$. Firstly, we examine bifurcation analysis with the help of the center manifold theorem [31, 32]. We investigate the properties of the equilibrium solutions in

the vicinity of the bifurcation point $x = x_0$ where $\mathcal{R}_0 = 1$. As $\mathcal{R}_0$ can be difficult as a direct bifurcation parameter, we introduce a new parameter $\mu_1$ for this purpose. We define $\mu_1$ as a bifurcation parameter such that $\mathcal{R}_0 < 1$ for $\mu_1 < 0$ and $\mathcal{R}_0 > 1$ for $\mu_1 > 0$. Moreover, we ensure that $x_0$ persists in a disease-free equilibrium (DFE) for all values of $\mu_1$. Consider the structure,

$$\dot{x} = f(x, \mu_1).$$

Here, the restriction is that $f$ possesses at least two continuous derivatives concerning both $x$ and $\mu_1$. The line $(x_0, \mu_1)$ represents the disease-free equilibrium (DFE), and the local stability of the DFE undergoes a shift at the point $(x_0, 0)$. Applying center manifold theory, we demonstrate the existence of non-trivial (endemic) equilibria close to the bifurcation point $(x_0, 0)$. Before exploring these findings, we introduce some notation and gather relevant facts.

We represent the partial derivative of $f$ with respect to $x$ at $x = x_0$, $\mu_1 = 0$ as $D_x f(x_0, 0)$. Assuming that $D_x f(x_0, 0)$ has simple zero eigenvalues, we define $v$ and $w$ as the corresponding left and right null vectors such that $vw = 1$. We also ensure that the remaining eigenvalues of $D_x f(x_0, 0)$ possess negative real parts. Let's,

$$a = \frac{v}{2} D_{xx} f(x_0, 0) w^2 = \frac{1}{2} \sum_{i,j,k=1}^{n} v_i w_j w_k \frac{\partial^2 f_i}{\partial x_j \partial x_k}(x_0, 0).$$

$$b = v D_{x\mu_1} f(x_0, 0) w = \sum_{i,j=1}^{n} v_i w_j \frac{\partial^2 f_i}{\partial x_j \partial \mu_1}(x_0, 0).$$

We will demonstrate that the sign of $a$ dictates the characteristics of the EE near the bifurcation point. Before delving into this analysis, it is essential to mention that the expression for $a$ can be revised using the outcomes established in the preceding sections. We utilize center manifold theory to theoretically divine the existence of the backward bifurcation phenomenon in the model (7), as outlined below.

**Theorem 2**. *Consider the following system of ordinary differential equations, incorporating a parameter $\phi$,*

$$\frac{dx}{dt} = f(x, \phi), f : \mathbb{R}^n \times \mathbb{R} \to \mathbb{R}, f \in \mathbb{C}^2(\mathbb{R}^n \times \mathbb{R}). \tag{8}$$

*Without departing generality, we assume that $x = 0$ serves as an equilibrium for the system (8) across all parameter values of $\phi$. Suppose that*

*(a) The matrix $A = D_x f(0, 0)$ represents the linearized matrix of system (8) at the equilibrium $x = 0$, where $\phi$ evaluated at 0. In this case, 0 is a simple eigenvalue of A, and all other eigenvalues of A reflect negative real parts.*

*(b) The matrix A possesses a non-negative right eigenvector $w$ and a left eigenvector $v$ associated with the zero eigenvalue. Let $f_k$ denote the $k^{th}$ component of $f$, and*

$$a = \sum_{k,i,j=1}^{n} v_k w_i w_j \frac{\partial^2 f_k}{\partial x_i \partial x_j}(0, 0), \quad b = \sum_{k,i=1}^{n} v_k w_i \frac{\partial^2 f_k}{\partial x_i \partial \beta}(0, 0).$$

*Subsequently, the signs of a and b have a complete influence on the local dynamics of system (8) around zero.*

*(i) Case-1: For $\phi < 0$ with $|\phi| \ll 1$, the equilibrium at $x = 0$ is locally asymptotically stable and a positive unstable equilibrium exists if $a > 0$ and $b > 0$. When $0 < \phi \ll 1$, a negative and locally asymptotically stable equilibrium is present, and the equilibrium at $x = 0$ is unstable.*

*(ii) Case-2: In the scenario where $a < 0$ and $b < 0$, for $|\phi| \ll 1$, the equilibrium at $x = 0$ is unstable. There is a negative unstable equilibrium and the equilibrium at $x = 0$ becomes locally asymptotically stable when $0 < \phi \ll 1$.*

*(iii) Case-3: With $a > 0$ and $b < 0$, the equilibrium at $x = 0$ is unstable and a locally asymptotically stable negative equilibrium emerges when $\phi < 0$ with $|\phi| \ll 1$. When $0 < \phi \ll 1$, the equilibrium at $x = 0$ becomes stable, and a positive unstable equilibrium emerges.*

*(iv) Case-4: $\phi$ changes from negative to positive, $x = 0$ changes its stability from stable to unstable when $a < 0$, $b > 0$. A negative unstable equilibrium correspondingly becomes positive and locally asymptotically stable.*

*In particular, if $a > 0$ and $b > 0$, a backward bifurcation occurs at $\phi = 0$. These conditions, delineating the bifurcation locally at $\mathcal{R}_0 = 1$, align with the scenarios illustrated in Fig 4. Specifically, conditions (ii) and (iv) denote a forward bifurcation scenario, while conditions (i) and (iii) indicate the occurrence of a backward bifurcation [34, 35].*

Let, $x = (x_1, x_2, x_3, x_4, x_5, x_6)^T = (S, V, E, I, R, T)^T$. Thus, the model (7) is modified with the previous model (1) by taking reinfection term $\phi_1$ which goes backward from $V$ compartment to $S$. Thus, model (7) is in the form $\frac{dx}{dt} = f(x)$, with $f(x) = (f_1(x), f_2(x), \cdots f_6(x))$. The Jacobian matrix of the system (7) at DFE $\mathcal{E}^0$ is given as,

$$
J^*(E_{0V})|_{\beta_2 = \beta_2^*} = \begin{pmatrix}
a_{11} & \phi_1 & -\beta_1 x_1 & -\beta_2 x_1 & 0 & 0 \\
\phi & a_{22} & -(1-\varepsilon)\beta_1 x_1 & -(1-\varepsilon)\beta_2 x_2 & 0 & 0 \\
\beta_1 x_3 + \beta_2 x_4 & 0 & a_{33} & \beta_2 x_1 & 0 & 0 \\
0 & a_{42} & \alpha + \lambda\beta_1 x_2 & a_{44} & 0 & 0 \\
0 & 0 & 0 & \gamma & -\mu & 0 \\
0 & 0 & 0 & \gamma_1 & 0 & -\mu
\end{pmatrix}
$$

where, $a_{11} = -(\beta_1 x_3 + \beta_2 x_4) - (\mu + \phi)$, $a_{22} = -\lambda(\beta_1 x_3 + \beta_2 x_4) - (\mu + \phi_1)$, $a_{33} = \beta_1 x_1 - (\alpha + \mu)$, $a_{42} = \lambda(\beta_1 x_3 + \beta_2 x_4)$, and $a_{44} = \lambda\beta_2 x_2 - (\mu + \delta + \gamma + \gamma_1)$.

We have considered contact rate $\beta_2 = \beta_2^*$ as the bifurcation parameter, setting $\mathcal{R}_0 = 1$ gives,

$$
S_0\alpha\beta_2^* + V_0\beta_2\lambda(\alpha + \mu) + S_0\beta_1(\mu + \delta + \gamma + \gamma_1) = (\alpha + \mu)(\mu + \delta + \gamma + \gamma_1)
$$

$$
\beta_2^* = \frac{(\alpha + \mu)(\mu + \delta + \gamma + \gamma_1) - S_0\beta_1(\mu + \delta + \gamma + \gamma_1)}{V_0\lambda(\alpha + \mu) + S_0\alpha}
$$

$$
\beta_2 = \beta^* = \frac{(\mu + \phi)(\alpha + \mu)(\mu + \delta + \gamma + \gamma_1) - \mu N\beta_1(\mu + \delta + \gamma + \gamma_1)}{\phi N\lambda(\alpha + \mu) + \mu N\alpha}
$$

[By putting $S_0$ and $V_0$ at the DFE value.]

At the DFE point, we have $x_1 = \frac{\Lambda}{\mu + \phi} = \frac{\mu N}{\mu + \phi} = \frac{k_1}{k_2}$ , $x_2 = \frac{N\phi}{\mu + \phi} = \frac{k_3}{k_2}$.

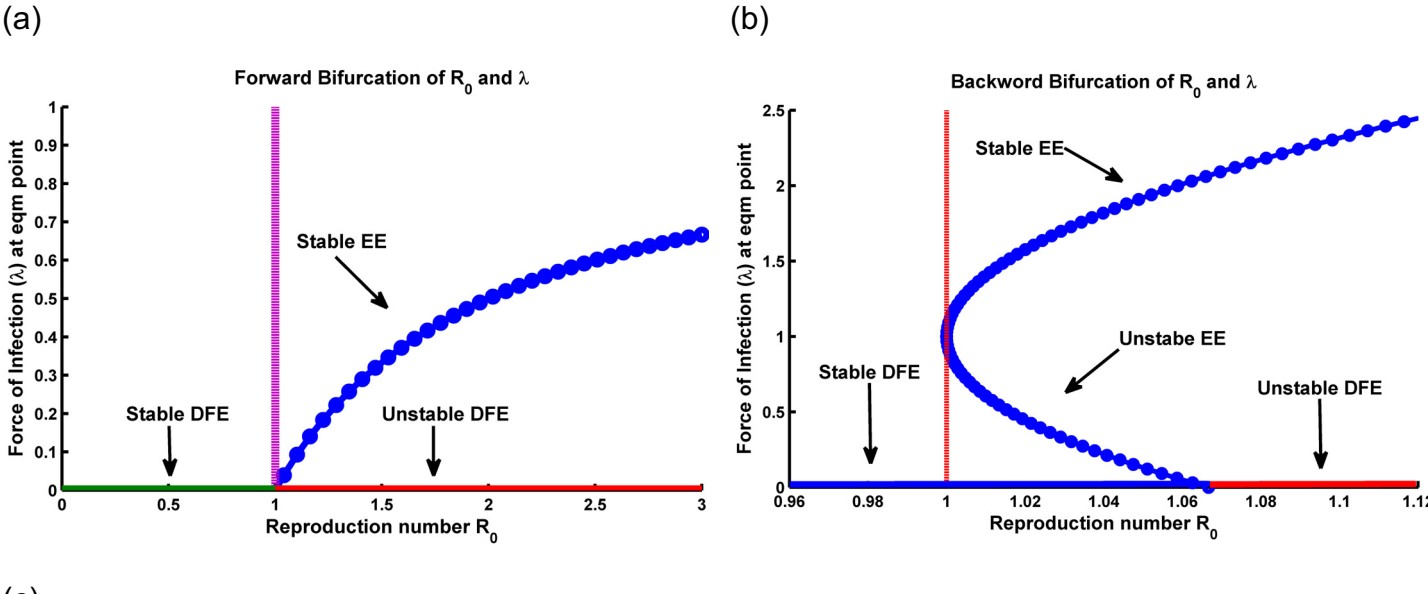

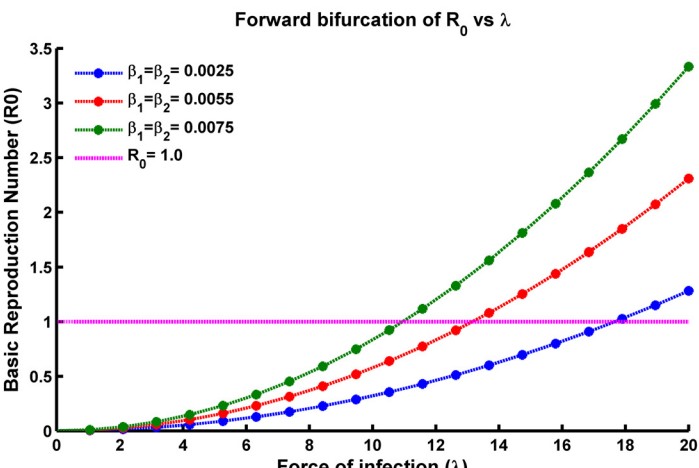

**Fig 4. Simmulation of model (1) showing (a) forward bifurcation vs $\mathcal{R}_0$, (b) backward bifurcation vs $\mathcal{R}_0$, and (c) bifurcation of $\mathcal{R}_0$ vs the force of infection λ at the equilibrium point respectively, where all the parameters are taken from Table 1.**

Here, we let $k_1 = \mu N$, $k_3 = N\phi$, and $k_2 = \mu + \phi$. Hence,

$$
J^*(E_{0V})\big|_{\beta_2 = \beta_2^*} =
\begin{pmatrix}
-(\mu + \phi) & \phi_1 & \dfrac{-\beta_1 k_1}{k_2} & \dfrac{-\beta_2 k_1}{k_2} & 0 & 0 \\
\phi & -(\mu + \phi_1) & -\lambda\beta_1 \dfrac{k_3}{k_2} & -\lambda\beta_2 \dfrac{k_3}{k_2} & 0 & 0 \\
0 & 0 & \beta_1 \dfrac{k_1}{k_2} - (\alpha + \mu) & \beta_2 \dfrac{k_1}{k_2} & 0 & 0 \\
0 & 0 & \alpha + \lambda\beta_1 \dfrac{k_3}{k_2} & a_{44} & 0 & 0 \\
0 & 0 & 0 & \gamma & -\mu & 0 \\
0 & 0 & 0 & \gamma_1 & 0 & -\mu
\end{pmatrix}
$$

where, $a_{44} = \lambda\beta_2 \frac{k_3}{k_2} - (\mu + \delta + \gamma + \gamma_1)$.

The transformed system (7) at the DFE $\mathcal{E}^0$ calculated for $\beta_2 = \beta_2^*$ has a hyperbolic equilibrium point i.e. a simple eigenvalue with 0 and all other eigenvalues have a negative real part. We therefore apply the Centre Manifold theorem in order to analyze the dynamics of (7) near $\beta_2 = \beta_2^*$.

The Jacobian of (7) at $\beta_2 = \beta_2^*$ denoted by $J(\mathcal{E}^0)|_{\beta_2=\beta^*}$. Now, the right ($w$) and left ($v$) eigenvector are computed from $J(\mathcal{E}^0)|_{\beta_2=\beta^*}$ (associated with zero eigenvalues) are given as

**Right Eigenvector**: $J^*(E_{0V})\underline{w} = 0$. Where,

$$\underline{w} = \begin{pmatrix} w_1 \\ w_2 \\ w_3 \\ w_4 \\ w_5 \\ w_6 \end{pmatrix}, \text{ and } \underline{0} = \begin{pmatrix} 0 \\ 0 \\ 0 \\ 0 \\ 0 \\ 0 \end{pmatrix}.$$

For zero eigenvalue we have obtained,

$$-k_2 w_1 + \phi_1 w_2 - \frac{\beta_1 k_1}{k_2} w_3 - \frac{\beta_2 k_1}{k_2} w_4 = 0.$$

$$\phi w_1 - (\mu + \phi_1) w_2 - \lambda \beta_1 \frac{k_3}{k_2} w_3 - \lambda \beta_2 \frac{k_3}{k_2} w_4 = 0.$$

$$\left( \beta_1 \frac{k_1}{k_2} - (\alpha + \mu) \right) w_3 + \beta_2 \frac{k_1}{k_2} w_4 = 0. \tag{9}$$

$$\left( \alpha + \lambda \beta_1 \frac{k_3}{k_2} \right) w_3 + \left( \lambda \beta_2 \frac{k_3}{k_2} - (\mu + \delta + \gamma + \gamma_1) \right) w_4 = 0.$$

$$\gamma w_4 - \mu w_5 = 0.$$

$$\gamma_1 w_4 - \mu w_6 = 0.$$

Let, $w_4 > 0$ be the free variable. Here,

$$w_5 = \frac{\gamma w_4}{\mu}, \text{ and } w_6 = \frac{\gamma_1 w_4}{\mu}. \tag{10}$$

Also, from the third equation of (9) we have,

$$w_3 = \frac{\beta_2 k_1 w_4}{k_2(\alpha + \mu) - \beta_1 k_1}. \tag{11}$$

Multiply first equation of (7) by $\phi$, and second equation of (7) by $k_2$,

$$-k_2^2 \phi w_1 + \phi \phi_1 k_2 w_2 - \beta_1 k_1 \phi w_3 - \beta_2 k_1 \phi w_4 = 0.$$

$$\phi k_2^2 w_1 - k_2^2(\mu + \phi_1) w_2 - \lambda \beta_1 k_3 k_2 w_3 - \lambda \beta_2 k_2 k_3 w_4 = 0.$$

By adding these two equations we obtain,

$$w_2\{k_2 \phi \phi_1 - k_2^2(\mu + \phi_1)\} - w_3(\beta_1 k_1 \phi + \lambda \beta_1 k_3 k_2) - w_4(\beta_2 k_1 \phi + \beta_2 k_3 k_2 \lambda) = 0$$

$$\Rightarrow \quad w_2\{k_2 \phi \phi_1 - k_2^2(\mu + \phi_1)\} - \frac{\beta_2 k_1(\beta_1 k_1 \phi + \lambda \beta_1 k_3 k_2)}{k_2(\alpha + \mu) - \beta_1 k_1} w_4 - w_4(\beta_2 k_1 \phi + \beta_2 k_3 k_2 \lambda) = 0 \tag{12}$$

$$\Rightarrow \quad w_2 = \frac{\beta_2 k_1(\beta_1 k_1 \phi + \lambda \beta_1 k_3 k_2) + (\beta_2 k_1 \phi + \beta_2 k_3 k_2 \lambda)(k_2(\alpha + \mu) - \beta_1 k_1)}{(k_2 \phi \phi_1 - k_2^2(\mu + \phi_1))(k_2(\alpha + \mu) - \beta_1 k_1)} w_4 = \frac{C_{11}}{C_{22}} w_4.$$

Substituting these expressions of $w_2$ and $w_3$ in the first equation of (9) we get,

$$w_1 = \frac{1}{k_2}\left[\frac{\phi C_{11}}{C_{22}} - \frac{\beta_1 k_1^2 \beta_2}{k_2^2(\alpha + \mu) - \beta_1 k_1 k_2} - \frac{\beta_2 k_1}{k_2}\right]w_4. \tag{13}$$

Thus, all parameters of right eigenvectors $w_1$, $w_2$, $w_3$, $w_5$, and $w_6$ can be expressed in terms of $w_4$.

**Left Eigenvector**: Similarly, from $J(E_{0V})|_{\beta_2=\beta*}$ we obtain,

$$[v1, v2, v3, v4, v5, v6]J^*(E_{0V}) = [0, 0, 0, 0, 0, 0].$$

Then,

$$-k_2 v_1 + \phi v_2 = 0,$$
$$\phi_1 v_1 - (\mu + \phi_1)v_2 = 0,$$
$$\frac{-\beta_1 k_1}{k_2}v_1 - \frac{\lambda \beta_1 k_3}{k_2}v_2 + \left(\frac{\beta_1 k_1}{k_2} - (\alpha + \mu)\right)v_3 + \left(\alpha + \lambda \beta_1 \frac{k_3}{k_2}\right)v_4 = 0, \tag{14}$$
$$-\frac{\beta_2 k_1}{k_2}v_1 - \frac{\lambda \beta_2 k_3}{k_2}v_2 + \frac{\beta_2 k_1}{k_2}v_3 + \left[\lambda \beta_2 \frac{k_3}{k_2} - (\mu + \delta + \gamma + \gamma_1)\right]v_4 + \gamma v_5 + \gamma_1 v_6 = 0,$$
$$-\mu v_5 = 0, \quad \text{and} \quad -\mu v_6 = 0.$$

Therefore, $v_1 = \frac{\phi v_2}{k_2}$, $v_5 = 0$, and $v_6 = 0$. Putting $v_5 = v_6 = 0$ on the third and fourth equation of (14),

$$\beta_1 k_1 v_1 + \lambda \beta_1 k_3 v_2 - (\beta_1 k_1 - k_2(\alpha + \mu))v_3 - (\alpha k_2 + \lambda \beta_1 k_3)v_4 = 0.$$
$$-\beta_2 k_1 v_1 - \lambda \beta_2 k_3 v_2 + \beta_2 k_1 v_3 + [\lambda \beta_2 k_3 - k_2(\mu + \delta + \gamma + \gamma_1)]v_4 = 0.$$

Now, multiplying third equation by $\beta_2$ and fourth equation by $\beta_1$ of (14),

$$\beta_2 \beta_1 k_1 v_1 + \lambda \beta_1 \beta_2 k_3 v_2 - (\beta_1 \beta_2 k_1 - \beta_2 k_2(\alpha + \mu))v_3 - \beta_2(\alpha k_2 + \lambda \beta_1 k_3)v_4 = 0.$$
$$-\beta_1 \beta_2 k_1 v_1 - \lambda \beta_1 \beta_2 k_3 v_2 + \beta_2 \beta_1 k_1 v_3 + \beta_1[\lambda \beta_2 k_3 - k_2(\mu + \delta + \gamma + \gamma_1)]v_4 = 0.$$

By performing addition,

$$[\beta_2 k_2(\alpha + \mu)]v_3 = [\beta_2(\alpha k_2 + \lambda \beta_1 k_3) - \beta_1(\lambda \beta_2 k_3 - k_2(\mu + \delta + \gamma + \gamma_1))]v_4.$$

That implies,

$$v_3 = \frac{C_{33}}{C_{44}}v_4. \tag{15}$$

Where $C_{33} = [\beta_2(\alpha k_2 + \lambda \beta_1 k_3) - \beta_1(\lambda \beta_2 k_3 - k_2(\mu + \delta + \gamma + \gamma_1))]$, and $C_{44} = [\beta_2 k_2(\alpha + \mu)]$. In the third equation of (14) we have got,

$$\left[\frac{\beta_1 k_1 \phi}{k_2^2} + \frac{\lambda \beta_1 k_3}{k_2}\right]v_2 = \left[\left(\frac{\beta_1 k_1}{k_2} - (\alpha + \mu)\right)\frac{C_{33}}{C_{44}} + \left(\alpha + \lambda \beta_1 \frac{k_3}{k_2}\right)\right]v_4.$$

Thus,

$$v_2 = \frac{C_{32}}{C_{42}}v_4. \tag{16}$$

Where, $C_{32} = \left[\left(\frac{\beta_1 k_1}{k_2} - (\alpha + \mu)\right)\frac{C_{33}}{C_{44}} + \left(\alpha + \lambda \beta_1 \frac{k_3}{k_2}\right)\right]$, and $C_{42} = \left[\frac{\beta_1 k_1 \phi}{k_2^2} + \frac{\lambda \beta_1 k_3}{k_2}\right]$.

Hence,

$$v_1 = \frac{\phi}{k_2} v_2 = \frac{\phi C_{32}}{k_2 C_{42}} v_4. \tag{17}$$

Thus, $v_1$, $v_2$, and $v_3$ can be expressed in terms of $v_4$. Hence, in the left eigenvector, we can assume $v_4$ as a free variable. Since $v_4$ is a free variable, we evaluate the second-order partial derivatives $f_i$ at the DFE point $\mathcal{E}^0$ to exhibit the existence of backward bifurcation. Furthermore, in $f_4$ there are $\beta_1$ and $\beta_2$ terms (which are contact rates related to disease transmission). So, the associate non-zero second partial derivative of the model (7) evaluated at $(E_{0V}, \beta^*)$. Now,

$$f_4 = \alpha x_3 + (1 - \varepsilon)(\beta_1 x_3 + \beta_2 x_4)x_2 - (\mu + \delta + \gamma + \gamma_1)x_4.$$

Taking derivative with respect to $x_3$ and $x_4$ with possible other combinations, as $x_3$ and $x_4$ indicates $E$ and $I$ terms.

$$\frac{\partial f_4}{\partial x_3} = \alpha + (1 - \varepsilon)\beta_1 x_2^0 = \alpha + \frac{(1 - \varepsilon)(x_1 + x_2 + x_3 + x_4 + x_5 + x_6)\phi\beta_1}{\mu + \phi},$$

$$\frac{\partial^2 f_4}{\partial x_3 \partial x_1} = \frac{\partial^2 f_4}{\partial x_1 \partial x_3} = \frac{(1 - \varepsilon)\phi\beta_1}{\mu + \phi}, \quad \frac{\partial^2 f_4}{\partial x_3 \partial x_2} = \frac{\partial^2 f_4}{\partial x_2 \partial x_3} = \frac{(1 - \varepsilon)\phi\beta_1}{\mu + \phi}, \quad \frac{\partial^2 f_4}{\partial x_3 \partial x_3} = \frac{\partial^2 f_4}{\partial x_3 \partial x_3} = \frac{(1 - \varepsilon)\phi\beta_1}{\mu + \phi},$$

$$\frac{\partial^2 f_4}{\partial x_3 \partial x_4} = \frac{\partial^2 f_4}{\partial x_4 \partial x_3} = \frac{(1 - \varepsilon)\phi\beta_1}{\mu + \phi}, \quad \frac{\partial^2 f_4}{\partial x_3 \partial x_5} = \frac{\partial^2 f_4}{\partial x_5 \partial x_3} = \frac{(1 - \varepsilon)\phi\beta_1}{\mu + \phi}, \quad \frac{\partial^2 f_4}{\partial x_3 \partial x_6} = \frac{\partial^2 f_4}{\partial x_6 \partial x_3} = \frac{(1 - \varepsilon)\phi\beta_1}{\mu + \phi}.$$

Now, $\frac{\partial f_4}{\partial x_4} = (1 - \varepsilon)\beta_2 x_2^0 - (\mu + \delta + \gamma + \gamma_1)$. Thus, putting the above $E_{0V}$ value,

$$\frac{\partial f_4}{\partial x_4} = \frac{(1 - \varepsilon)\beta_2(x_1 + x_2 + x_3 + x_4 + x_5 + x_6)\phi}{\mu + \phi} - (\mu + \delta + \gamma + \gamma_1),$$

$$\frac{\partial^2 f_4}{\partial x_4 \partial x_1} = \frac{\partial^2 f_4}{\partial x_1 \partial x_4} = \frac{(1 - \varepsilon)\phi\beta_2}{\mu + \phi}, \quad \frac{\partial^2 f_4}{\partial x_4 \partial x_2} = \frac{\partial^2 f_4}{\partial x_2 \partial x_4} = \frac{(1 - \varepsilon)\phi\beta_2}{\mu + \phi}, \quad \frac{\partial^2 f_4}{\partial x_4 \partial x_3} = \frac{\partial^2 f_4}{\partial x_3 \partial x_4} = \frac{(1 - \varepsilon)\phi\beta_2}{\mu + \phi},$$

$$\frac{\partial^2 f_4}{\partial x_4 \partial x_4} = \frac{\partial^2 f_4}{\partial x_4 \partial x_4} = \frac{(1 - \varepsilon)\phi\beta_2}{\mu + \phi}, \quad \frac{\partial^2 f_4}{\partial x_4 \partial x_5} = \frac{\partial^2 f_4}{\partial x_5 \partial x_4} = \frac{(1 - \varepsilon)\phi\beta_2}{\mu + \phi}, \quad \frac{\partial^2 f_4}{\partial x_4 \partial x_6} = \frac{\partial^2 f_4}{\partial x_6 \partial x_4} = \frac{(1 - \varepsilon)\phi\beta_2}{\mu + \phi}.$$

Now, we determine the bifurcation coefficients of $\bar{a}$ and $\bar{b}$ defined in Theorem 2, stated by Castillo-Chavez and Song which is given as follows,

$$\begin{aligned}
\bar{b} &= \sum_{k,i=1}^{6} v_k w_i \frac{\partial^2 f_k(E_{0V}, \beta_2^* = 0)}{\partial x_i \partial \beta_2^*} \\
&= v_1 w_4 \frac{\partial^2 f_1}{\partial x_4 \partial \beta_2^*} + v_2 w_4 \frac{\partial^2 f_2}{\partial x_4 \partial \beta_2^*} + v_3 w_4 \frac{\partial^2 f_3}{\partial x_4 \partial \beta_2^*} + v_4 w_4 \frac{\partial^2 f_4}{\partial x_4 \partial \beta_2^*} \\
&\quad + v_1 w_3 \frac{\partial^2 f_1}{\partial x_3 \partial \beta_2^*} + v_2 w_3 \frac{\partial^2 f_2}{\partial x_3 \partial \beta_2^*} + v_3 w_3 \frac{\partial^2 f_3}{\partial x_3 \partial \beta_2^*} + v_4 w_3 \frac{\partial^2 f_4}{\partial x_3 \partial \beta_2^*}.
\end{aligned}$$

Here, we take combination of $v_1$, $v_2$, $v_3$, $v_4$ in expression of $\bar{b}$ as $\beta_1$, $\beta_2$ are present, also take combination of $w_3$, $w_4$ as they related to $E$ and $I$ compartments [36, 37]. Now, putting the DFE

value $(\mathcal{E}^0)$ we have,

$$\frac{\partial f_1}{\partial x_4} = -\beta_2^* x_1 \Rightarrow \frac{\partial^2 f_1}{\partial x_4 \partial \beta_2^*} = -x_1, \quad \frac{\partial f_2}{\partial x_4} = -(1-\varepsilon)\beta_2^* x_2 \Rightarrow \frac{\partial^2 f_2}{\partial x_4 \partial \beta_2^*} = -(1-\varepsilon)x_2,$$

$$\frac{\partial f_3}{\partial x_4} = \beta_2^* x_1 \Rightarrow \frac{\partial^2 f_3}{\partial x_4 \partial \beta_2^*} = x_1, \quad \frac{\partial f_4}{\partial x_4} = (1-\varepsilon)\beta_2^* x_2 - (\mu + \delta + \gamma + \gamma_1) \Rightarrow \frac{\partial^2 f_3}{\partial x_4 \partial \beta_2^*} = (1-\varepsilon)x_2.$$

$$\frac{\partial f_1}{\partial x_3} = -\beta_1 x_1 \Rightarrow \frac{\partial^2 f_1}{\partial x_3 \partial \beta_2^*} = 0, \quad \frac{\partial f_2}{\partial x_3} = -(1-\varepsilon)\beta_1 x_2 \Rightarrow \frac{\partial^2 f_2}{\partial x_3 \partial \beta_2^*} = 0,$$

$$\frac{\partial f_3}{\partial x_3} = -\beta_1 x_1 - (\alpha + \mu) \Rightarrow \frac{\partial^2 f_3}{\partial x_3 \partial \beta_2^*} = 0, \quad \frac{\partial f_4}{\partial x_3} = \alpha + (1-\varepsilon)\beta_1 x_2 \Rightarrow \frac{\partial^2 f_4}{\partial x_3 \partial \beta_2^*} = 0.$$

Putting these values in the above expression of $\bar{b}$,

$$\bar{b} = -v_1 x_1^* - v_2(1-\varepsilon)x_2^* + v_3 x_1^* + (1-\varepsilon)x_2^*.$$

As $v_4$ and $w_4$ are free variables we put $v_4 = w_4 = 1$. Now, substituting the DFE point we have,

$$\bar{b} = \frac{-v_1 \mu N - v_2(1-\varepsilon)N\phi + v_3 \mu N + (1-\varepsilon)N\phi}{\mu + \phi}.$$

Putting the expressions of $v_1$, $v_2$, and $v_3$ from (17), (16), and (15) we get explicit expression of $\bar{b}$ in terms of model parameters and it reflects that $\bar{b} > 0$ automatically.

In this study, for Influenza disease models, vaccine efficacy ($\epsilon$) is typically less than 1 to reflect real-world conditions where vaccines are not perfectly effective. This imperfection accounts for factors such as individual immune response variability, vaccine strain mismatches, or waning immunity over time. If $\epsilon = 1$, assuming perfect efficacy, the model implies complete protection conferred by vaccination, potentially leading to unrealistic scenarios. These could include an overestimation of vaccine impact, an underestimation of disease transmission, or an unrealistic portrayal of herd immunity dynamics. In reality, achieving perfect vaccine efficacy is rare due to breakthrough infections and other variables. That necessitates the parameter $\epsilon < 1$ to accurately account for the complexities of influenza transmission and vaccination dynamics. As vaccine efficiency ($\epsilon$) increases, the disease can be effectively controlled in a significantly shorter timeframe.

Here, vaccine efficiency is $0 < \varepsilon < 1$, and the reinfection rate from the vaccination compartment is $0 < \phi_1 < 1$. Now,

$$\bar{a} = \sum_{k,i,j=1}^{6} v_k w_i w_j \frac{\partial^2 f_k}{\partial x_i \partial x_j}(E_{0V}, \beta_2^*).$$

Here, we take $v_k = v_4$ as $v_4$ is a free variable, and $w_j = \{w_3, w_4\}$ as they correspond to the contact rates $\beta_1$ and $\beta_2$. Then,

$$\begin{aligned}
\bar{a} &= v_4 w_1 \left[ w_3 \frac{\partial^2 f_4}{\partial x_1 \partial x_3} + w_4 \frac{\partial^2 f_4}{\partial x_1 \partial x_4} \right] + v_4 w_2 \left[ w_3 \frac{\partial^2 f_4}{\partial x_2 \partial x_3} + w_4 \frac{\partial^2 f_4}{\partial x_2 \partial x_4} \right] + \\
&\quad v_4 w_3 \left[ w_3 \frac{\partial^2 f_4}{\partial x_3 \partial x_3} + w_4 \frac{\partial^2 f_4}{\partial x_3 \partial x_4} \right] + v_4 w_4 \left[ w_3 \frac{\partial^2 f_4}{\partial x_4 \partial x_3} + w_4 \frac{\partial^2 f_4}{\partial x_4 \partial x_4} \right] + \\
&\quad v_4 w_5 \left[ w_3 \frac{\partial^2 f_4}{\partial x_5 \partial x_3} + w_4 \frac{\partial^2 f_4}{\partial x_5 \partial x_4} \right] + v_4 w_6 \left[ w_3 \frac{\partial^2 f_4}{\partial x_6 \partial x_3} + w_4 \frac{\partial^2 f_4}{\partial x_6 \partial x_4} \right].
\end{aligned}$$

As $v_4$ and $w_4$ are free variables, so putting $v_4 = 1$ and $w_4 = 1$ in the expression of $\bar{a}$ we have,

$$\bar{a} = \frac{(1-\varepsilon)\phi\beta_1(w_1w_3 + w_2w_3 + w_3w_3 + w_3 + w_5w_3 + w_6w_3)}{(\mu + \phi)} +$$
$$\frac{(1-\varepsilon)\phi\beta_2^*(w_1 + w_2 + w_3 + 1 + w_5 + w_6)}{\mu + \phi}.$$

Substituting the expression of $w_1$, $w_2$, $w_3$, $w_4$, $w_5$, and $w_6$ from (13), (12) and (11) we have the explicit expression of $\bar{a}$ in terms of model parameters. Hence, $\bar{a} > 0$ automatically, as all parameters are non-negative, reinfection rate is $0 < \phi_1 < 1$, and vaccine efficiency rate is $0 < \varepsilon < 1$.

Since $\bar{a} > 0$ and $\bar{b} > 0$, by Theorem 2 the modified system (7) undergoes backward bifurcation at $R_0 = 1$. So, the condition to occur backward bifurcation in mode (7) is for $\bar{a} > 0$. This follows that the bifurcation parameter $\bar{a} > 0$ whenever,

$$\beta_2^* > \frac{(\varepsilon - 1)\beta_1(w_1w_3 + w_2w_3 + w_3w_3 + w_3 + w_5w_3 + w_6w_3)}{(1-\varepsilon)(w_1 + w_2 + w_3 + 1 + w_5 + w_6)}.$$

as required. Hence, the condition or crucial parameter for bifurcation ($\beta^*$) is obtained. Additionally, it is important to emphasize that in situations where susceptible individuals under lock-down do not become infected during the lock-down period (i.e., $\beta_2^* = 0$), the bifurcation coefficient $\bar{a}$ takes on a negative value. Thus, $\bar{a} < 0$ is the case according to the Theorem 2, for which no backward bifurcation occurs, hence forward bifurcation occurs. In other terms, this research demonstrates that the occurrence of backward bifurcation in the model (7) is a result of susceptible individuals getting reinfected during the lock-down period. This result is consistent when the DFE of the model is globally asymptotically stable with $\beta_2^* = 0$. Since $\bar{a} < 0$ and $\bar{b} > 0$ at $\beta^* = \beta_2^*$, a transcritical bifurcation occurs according to to the theorems in [31, 32].

In a forward bifurcation plot (Fig 4(a)), we typically have $\mathcal{R}_0$ on the x-axis and the force of infection ($\lambda_1$) on the y-axis. From Fig 4(a), we see that up to $\mathcal{R}_0 \in [0, 1]$, the force of infection lies in zero level. The plot shows different branches or curves that represent the possible force of infection for different values of $\mathcal{R}_0$. A forward bifurcation occurs when there is a critical value of $\mathcal{R}_0$ at which a qualitative change in the behavior of the force of infection occurs. This means that as $\mathcal{R}_0$ increases past this critical value ($\mathcal{R}_0 = 1$), the force of infection exhibits a sudden change in its behavior, such as transitioning from a low to a high level of infection or from a stable to an unstable state [38].

Fig 4(b) shows the backward bifurcation of the force of infection against $\mathcal{R}_0$. This plot highlights the transition from high to low infection levels as $\mathcal{R}_0$ exceeds 1, indicating that reducing $\mathcal{R}_0$ below a certain threshold may not eradicate the infection. It underscores the need to consider additional factors in public health strategies to address the persistence or resurgence of infections.

From Fig 4(c), when plotting the basic reproduction number ($\mathcal{R}_0$) for the force of infection ($\lambda_1$), the scenario of $\mathcal{R}_0$ depends on how $\mathcal{R}_0$ changes as the force of infection increases or decreases. Fig 4(c) shows that an increase in the force of infection corresponds to a higher rate at which susceptible individuals become infected. Meanwhile, $\mathcal{R}_0$ increases with increasing the force of infection, which indicates that the disease becomes more transmissible. A higher force of infection leads to a greater number of new infections caused by each infected individual. On the other hand, $\mathcal{R}_0$ decreases with the decreasing of the force of infection, which means that the disease becomes less transmissible. A lower force of infection results in a reduced number of new infections caused by each infected individual [33]. Transmission with contact rate $\beta_1$

and $\beta_2$, and factors such as population density, contact patterns, and interventions can also influence the relationship between $\mathcal{R}_0$ and the force of infection ($\lambda_1$).

Fig 4 illustrates that the bifurcation coefficient $\bar{b}$ is positive. Consequently, based on theorem [35, 38], the model (7) represents a backward bifurcation phenomenon when the backward coefficient, $\bar{a}$, is positive. Fig 4 depicts the corresponding forward and backward bifurcation diagram. Importantly, when setting the Influenza reinfection term $\phi_1$ to 0 as the modification parameter for enhanced susceptibility from the vaccination compartment, it was perceived that the bifurcation coefficient $\bar{a}$ is less than 0. In the Influenza co-infection scenario, if there is no reinfection after recovery and vaccinated individuals are protected from getting Influenza, backward bifurcation does not occur. This outcome aligns with the earlier result. From an epidemiological perspective, this implies that controlling Influenza becomes more challenging, even with continuous vaccination efforts and a reproduction number $\mathcal{R}_0 < 1$.

## Hopf-bifurcation analysis of the model

Hopf bifurcation occurs in an epidemic model when there is a transition from a stable equilibrium point to a stable limit cycle. In other words, the system exhibits periodic oscillations instead of converging to a steady state [39]. This is indicated in the SVEIRT model when the basic reproduction number $\mathcal{R}_0$ is greater than 1 and the stability of the DFE equilibrium point changes. This can happen when the reproduction number crosses a certain threshold value. The occurrence of Hopf bifurcation in an epidemic model is significant because it can lead to the emergence of sustained, periodic disease outbreaks [40]. This can happen even if the disease would have otherwise died out in the absence of intervention or natural immunity. Sometimes this occurs when the model includes time delays and the steady-state equilibrium becomes unstable, leading to oscillatory behavior or limit cycles in the system. The periodicity of the outbreaks can also make it more difficult to control the disease using traditional methods such as vaccination or quarantine.

Specifically, the Hopf bifurcation occurs when a pair of complex conjugate eigenvalues of the Jacobian matrix cross the imaginary axis as the value of the threshold quantity $\mathcal{R}_0$ increases. This causes the model to display limit cycles or periodic oscillations. Hopf bifurcation is crucial to understanding infectious disease dynamics because it can generate complex and unpredictable patterns in disease spread, with profound implications for public health [41].

To determine the conditions for a Hopf bifurcation in the SVEIRT model, one can use the Routh-Hurwitz criterion, which involves computing the coefficients of the characteristic polynomial of the linearized system evaluated at the DFE. We need to compute the Jacobian matrix evaluated at each equilibrium point. By setting the determinant of the Jacobian matrix evaluated at this point to zero,

$$|J(S^*, V^*, E^*, I^*, R^*, T^*) - \lambda I| = 0.$$

The characteristic polynomial can be written as

$$p(\lambda) = \lambda^6 + a_1\lambda^5 + a_2\lambda^4 + a_3\lambda^3 + a_4\lambda^2 + a_5\lambda + a_6.$$

Where the coefficients $a_1$, $a_2$, $a_3$, $a_4$, $a_5$, and $a_6$ depend on the model parameters and $\lambda$ is a complex eigenvalue with a positive real part that determines the stability of the limit cycle that arises from the bifurcation. To find the condition for a Hopf bifurcation, we need to calculate the sign of the coefficient of the linear term in the normal form of the system near the

equilibrium point. The normal form is given by,

$$\dot{z} = (\alpha + i\omega)z - \mu|z|^2 z.$$

Where $z$ is a complex variable representing the deviation from the equilibrium point, $\alpha$ and $\omega$ are real constants, and $\mu$ is a small parameter. To obtain the normal form, we need to examine the eigenvalues of the Jacobian matrix evaluated at the equilibrium point. If the real part of one of the eigenvalues changes sign as a parameter is varied, a Hopf bifurcation occurs. The characteristic polynomial gives us the eigenvalues of the Jacobian matrix. Therefore, we can use the coefficients of the characteristic polynomial to find the normal form coefficients. The coefficients are related to the normal form coefficients as follows,

$$\alpha = \frac{1}{2}(a_5 - a_1), \quad \omega = \frac{1}{2}(a_4 - a_2), \text{ and}$$

$$\mu = \frac{1}{4}(a_1 a_5 - a_2 a_4 + a_3 a_3) - \frac{1}{2}(a_1 a_4 + a_2 a_5) + a_3 a_6.$$

The condition for a Hopf bifurcation is that $\alpha = 0$ and $\omega \neq 0$. Therefore, we need to set $a_5 - a_1 = 0$ and $a_4 - a_2 \neq 0$. This gives us the conditions,

$$a_5 = a_1, \quad a_4 \neq a_2.$$

If the Routh-Hurwitz criterion yields that all the coefficients of the polynomial are positive, then the DFE is stable, and there is no Hopf bifurcation. However, if one or more of the coefficients are negative, then the disease-free equilibrium is unstable, and a Hopf bifurcation can occur [39–41]. In summary, the condition for a Hopf bifurcation is that the characteristic polynomial has a repeated eigenvalue, i.e, $a_5 = a_1$, and the coefficient of the quartic term is different from the coefficient of the quadratic term, i.e, $a_4 \neq a_2$.

To find the conditions for a Hopf bifurcation in our model (1), we need to analyze the Jacobian matrix evaluated at the equilibrium point and determine when a pair of complex conjugate eigenvalues crosses the imaginary axis. First, we need to determine the disease-free equilibrium point $\mathcal{E}^0$. The disease-free equilibrium corresponds to $E = I = 0$. Assuming $(S, V, E, I, R, T) = (S_0, V_0, 0, 0, R_0, T_0)$, we can find $S_0$ and $V_0$ as follows. From the equations,

$$\frac{dS}{dt} = \Lambda - (\mu + \phi)S = 0 \Rightarrow S_0 = \frac{\Lambda}{\mu + \phi},$$

$$\frac{dV}{dt} = \phi S - \mu V = 0 \Rightarrow V_0 = \frac{\phi S_0}{\mu} = \frac{\phi \Lambda}{\mu(\mu + \phi)}.$$

The other variables at the DFE are $E_0 = I_0 = R_0 = T_0 = 0$.

Next, we form the Jacobian matrix $J$ evaluated at $\mathcal{E}^0$. The Jacobian matrix for the system is,

$$J = \begin{pmatrix} a_{11} & 0 & -\beta_1 S & -\beta_2 S & 0 & 0 \\ \phi & a_{22} & -(1-\varepsilon)\beta_1 V & -(1-\varepsilon)\beta_2 V & 0 & 0 \\ 0 & 0 & \beta_1 S - (\alpha + \mu) & \beta_2 S & 0 & 0 \\ 0 & 0 & \alpha & a_{44} & 0 & 0 \\ 0 & 0 & 0 & \gamma & -\mu & 0 \\ 0 & 0 & 0 & \gamma_1 & 0 & -\mu \end{pmatrix}.$$

where, $a_{11} = -(\beta_1 E + \beta_2 I + \mu + \phi)$, $a_{22} = -(\mu + (1-\varepsilon)(\beta_1 E + \beta_2 I))$, $a_{44} = (1 - \varepsilon)(\beta_1 V + \beta_2 V) -$

$(\mu + \delta + \gamma + \gamma_1)$. At the DFE $\mathcal{E}^0$, the Jacobian matrix simplifies to,

$$J(\mathcal{E}^0) = \begin{pmatrix} -(\mu + \phi) & 0 & -\beta_1 S_0 & -\beta_2 S_0 & 0 & 0 \\ \phi & -\mu & 0 & 0 & 0 & 0 \\ 0 & 0 & \beta_1 S_0 - (\alpha + \mu) & \beta_2 S_0 & 0 & 0 \\ 0 & 0 & \alpha & b_{44} & 0 & 0 \\ 0 & 0 & 0 & \gamma & -\mu & 0 \\ 0 & 0 & 0 & \gamma_1 & 0 & -\mu \end{pmatrix}.$$

where, $b_{44} = -(1 - \varepsilon)\beta_2 V_0 - (\mu + \delta + \gamma + \gamma_1)$. To check for Hopf bifurcation, we need to find the eigenvalues of $J(\mathcal{E}^0)$ and analyze the conditions under which a pair of complex conjugate eigenvalues crosses the imaginary axis.

The characteristic equation of $J(\mathcal{E}^0)$ is given by,

$$\det(J(\mathcal{E}^0) - \lambda I) = 0.$$

To identify the conditions for Hopf bifurcation, we need to find when the real part of a pair of complex conjugate eigenvalues becomes zero. This typically requires setting up the characteristic polynomial and finding the parameter values for which the polynomial has purely imaginary roots $\lambda = \pm i\omega$.

For brevity, let's focus on the $2 \times 2$ submatrix that could potentially give rise to complex conjugate eigenvalues. Consider the subsystem involving $E$ and $I$,

$$\begin{pmatrix} \beta_1 S_0 - (\alpha + \mu) & \beta_2 S_0 \\ \alpha & -(1 - \varepsilon)\beta_2 V_0 - (\mu + \delta + \gamma + \gamma_1) \end{pmatrix}.$$

Let $a = \beta_1 S_0 - (\alpha + \mu)$ and $b = \beta_2 S_0$, and $c = -(1 - \varepsilon)\beta_2 V_0 - (\mu + \delta + \gamma + \gamma_1)$.
The characteristic equation for this subsystem is,

$$\lambda^2 - (a + c)\lambda + (ac - b\alpha) = 0.$$

The eigenvalues are,

$$\lambda = \frac{a + c \pm \sqrt{(a + c)^2 - 4(ac - b\alpha)}}{2}.$$

To occur Hopf bifurcation, the eigenvalues must be purely imaginary, which means,

$$(a + c) = 0 \quad \text{and} \quad (a + c)^2 - 4(ac - b\alpha) < 0.$$

Setting $a + c = 0$, we get,

$$\beta_1 S_0 - (\alpha + \mu) - (1 - \varepsilon)\beta_2 V_0 - (\mu + \delta + \gamma + \gamma_1) = 0.$$

Solving for $\beta_1$, $\beta_2$, or any parameter of interest, we get the Hopf bifurcation condition. After simplifying and rearranging terms,

$$\beta_1 \frac{\Lambda}{\mu + \phi} = (\alpha + \mu) + (1 - \varepsilon)\left(\beta_2 \frac{\phi\Lambda}{\mu(\mu + \phi)} + \mu + \delta + \gamma + \gamma_1\right).$$

Hence, the condition for Hopf bifurcation is,

$$\beta_1 \frac{\Lambda}{\mu + \phi} - (1 - \varepsilon)\beta_2 \frac{\phi\Lambda}{\mu(\mu + \phi)} = (\alpha + \mu) + (1 - \varepsilon)(\mu + \delta + \gamma + \gamma_1).$$

## Graphical analysis of hopf-bifurcation

Hopf bifurcation is represented in the susceptible population in Fig 5(a), oscillating within 5 to 100 weeks, and describes a complex dynamic pattern. The population's number of susceptible individuals ($S(t)$) exhibits periodic fluctuations over time. This suggests a qualitative change in the transmission dynamics of the infectious disease, leading to sustained oscillations in population susceptibility.

When Hopf bifurcation occurs in the vaccinated population presented in Fig 5(b), it suggests that the disease transmission dynamics are changing due to the vaccination efforts. Meanwhile, the temporal pattern of the vaccinated population's susceptibility to the disease. The vaccinated population initially increases after the introduction of vaccination, leading to a decrease in disease transmission. After a certain period (20 weeks), the vaccinated population's susceptibility starts oscillating, leading to periodic fluctuations in disease transmission (Fig 5 (b)). These oscillations can occur due to various factors, such as waning immunity, contact patterns changing, or the emergence of new pathogen variants. The specific characteristics of the oscillations (e.g., amplitude, frequency) can provide insights into the dynamics of the disease and the effectiveness of vaccination strategies.

The specific interpretation of Fig 5(c) describing the oscillation after 20 weeks in the exposed population $E(t)$. In the case of a Hopf bifurcation, the oscillations in the exposed population could suggest periodic fluctuations in the number of individuals transitioning from the susceptible to the exposed state and vice versa. The precise implications of such oscillations would depend on various factors, including the specific disease being modeled, the population dynamics, and the model parameters. It reflects seasonal patterns, cyclic changes in human behavior or interventions, or other factors that influence the transmission dynamics of the disease.

In Fig 5(d), Hopf bifurcation is represented in the infected population in epidemiology. The number of infected individuals exhibits sustained oscillations after a certain period, such as 20 weeks. This means that the population of infected individuals cyclically fluctuates over time. The observed oscillations could stem from seasonal influences on disease transmission, variations in human behavior or interventions over time, and self-regulating feedback mechanisms within the population and disease dynamics.

In Fig 5(e) and 5(f), the representation of a Hopf bifurcation in the recovered and treatment population with oscillation after 15 weeks suggests a specific dynamic behavior in the disease system. A Hopf bifurcation occurs when a system undergoes a qualitative change in its behavior as a parameter (in this case, possibly an infection rate or treatment effectiveness) crosses a critical threshold. In this scenario, Fig 5(e) and 5(f) describe the fluctuation or oscillation observed in the population of individuals who have recovered from the disease and those undergoing treatment over 15 weeks. The oscillation indicates that the population sizes

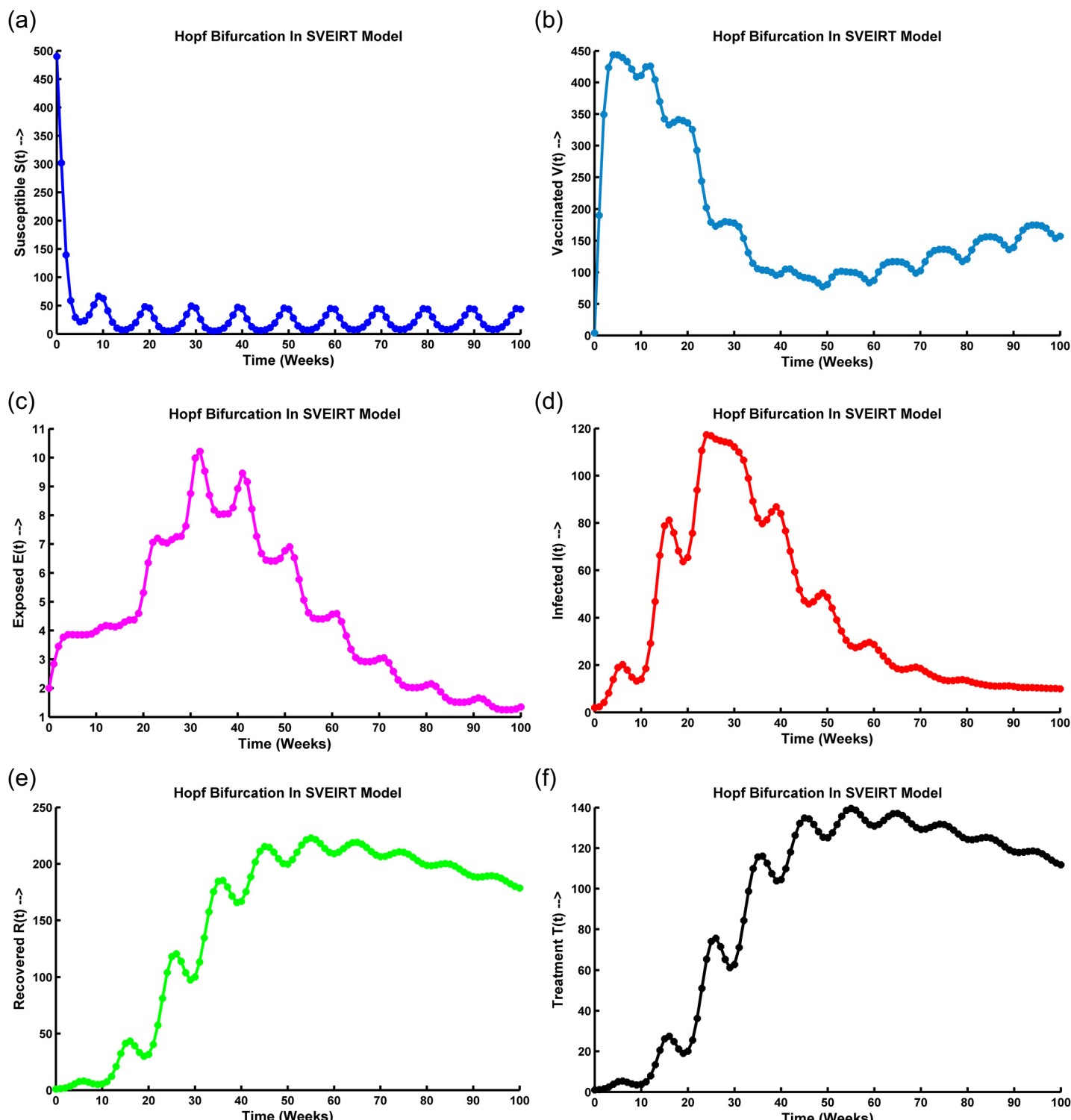

**Fig 5. Hopf bifurcation occurs at (a)** $S(t)$ **compartment, (b)** $V(t)$ **compartment, (c)** $E(t)$ **compartment, (d)** $I(t)$ **compartment, (e)** $R(t)$ **compartment, and (f)** $T(t)$ **compartment when the periodic instability occurs, the parameter values are taken from Table 1.**

of these two groups are changing periodically, possibly with a recurring pattern. Some observations are: the Hopf bifurcation and subsequent oscillation suggest that the disease dynamics have transitioned from a stable state to cyclical or periodic behavior. The patterns indicate that factors such as seasonal variations, behavioral changes, or the dynamics of immunity and treatment influence the population sizes of the recovered and treatment groups.

Once a Hopf bifurcation occurs, the disease-free equilibrium becomes unstable, and a stable limit cycle emerges. The limit cycle represents the oscillatory behavior of the disease dynamics, where the number of individuals in each compartment (susceptible, vaccinated, exposed, infected, recovered, and treated) varies periodically over time. The exact conditions for the emergence of the limit cycle depend on the specific values of the model parameters and cannot be determined analytically. However, numerical simulations can be used to explore the system's behavior near the Hopf bifurcation and to estimate the parameters that lead to the emergence of the limit cycle.

Moreover, in epidemic models, Hope bifurcation oscillates with time due to dynamic interactions between the host population and the pathogen. Biologically, this oscillation can be justified by varying transmission rates, seasonal fluctuations in host susceptibility or behavior, and pathogen evolution. These dynamic changes create feedback loops within the system, leading to periodic oscillations in disease prevalence. Additionally, interventions like vaccination campaigns or behavioral changes may also introduce periodic fluctuations in transmission dynamics.

## Stability analysis and persistence of DFE and EE

We know that the stability characteristics of linear ODEs only depend on the system's eigenvalues. Since our suggested model (1) is non-linear, we must use Hartman and Grobman's theorem, and linearization to combine the local behavior of linear and non-linear systems. We now analyze the local stability of the equilibria at points $\mathcal{E}^0$ and $\mathcal{E}^*$ by approximating the nonlinear system of differential equations with a linear system. The system is then locally perturbed from equilibrium, and the long-term behavior that results is then examined. This is performed by linearizing the system about each equilibrium, using the Jacobian approach for (1). Analyzing the linearized system,

$$\dot{z} = J(\mathbb{E})z(t).$$

Here, $\mathbb{E}$ represents the equilibrium space. We can look into the stability of each equilibrium point $\mathbb{E} = \mathcal{E}^0$ and $\mathbb{E} = \mathcal{E}^*$. We will obtain that the property depends on a crucial factor, referred to as basic reproduction number $\mathcal{R}_0$ which is estimated above. As a result, the nature of $\mathcal{R}_0$ can be examined to determine whether persistence or extinction of disease occurs as $t \to \infty$ [8, 42].

In this section, we shall proceed to analyze the stability properties of the DFE and EE. Firstly, we analyze the following results regarding to local and global stability of DFE.

### Local stability of disease-free equilibrium state $(\mathcal{E}^0)$

**Theorem 3**. *The disease-free equilibrium point is locally asymptotically stable if $\mathcal{R}_0 < 1$ and unstable if $\mathcal{R}_0 > 1$.*

*Proof.* The next step is to linearize the system and use the Routh-Hurwitz criterion to identify the circumstances in which the linear system has only negative eigenvalues. Because a point is deemed an attractor if its Jacobian matrix's eigenvalues at that location have negative

real portions, even slight disturbances from the equilibrium induce the system to gradually return there. Alternatively, if any eigenvalues represent positive real parts, slight deviations from the equilibrium lead to amplification, causing the system to diverge. This results in a "repeller" point, where the local behavior of the linearized system aligns with the non-linear system, following the Hartman-Grobman theorem. For the system, the Jacobian at the DFE point $\mathcal{E}^0 = (S_0, V_0, E_0, I_0, R_0, T_0)$ results in,

$$J(\mathcal{E}^0) = \begin{pmatrix} -(\mu + \phi) & 0 & -\beta_1 S_0 & -\beta_2 S_0 & 0 & 0 \\ \phi & -\mu & -\lambda\beta_1 V_0 & -\lambda\beta_2 V_0 & 0 & 0 \\ 0 & 0 & a_{33} & \beta_2 S_0 & 0 & 0 \\ 0 & 0 & \alpha + \lambda\beta_1 V_0 & a_{44} & 0 & 0 \\ 0 & 0 & 0 & \gamma & -\mu & 0 \\ 0 & 0 & 0 & \gamma_1 & 0 & -\mu \end{pmatrix} \quad (18)$$

Here, we let, $\lambda = (1 - \varepsilon)$, $a_{33} = \beta_1 S_0 - (\alpha + \mu)$, and $a_{44} = \lambda\beta_2 V_0 - (\mu + \delta + \gamma + \gamma_1)$.

At the DFE point we have substituted $S_0 = \frac{\mu N}{\mu + \phi}$, $V_0 = \frac{\phi N}{\mu + \phi}$, and rest of all the state variables 0. The characteristic polynomial of $J(\mathcal{E}^0)$ is got as,

$$|J(\mathcal{E}^0) - x\mathbb{I}| = 0.$$

Expanding the terms and ordering by powers of $x$ we obtain the required characteristic polynomial. This equation ultimately simplifies to,

$(x + \mu)^3 (x^2 + (\alpha + \mu)(\gamma + \gamma_1 + \delta + \mu - \beta_2 V_0 \lambda) + x(\alpha - \beta_1 S_0 + \gamma + \gamma_1 + \delta + 2\mu - V_0\beta_2\lambda) -$

$S_0(\alpha\beta_2 + \beta_1(\mu + \delta + \gamma + \gamma_1)))(x + \mu + \phi) = 0.$

$\Rightarrow \quad (x + \mu)^3 \{x^3 + x(\alpha + \mu)(\gamma + \gamma_1 + \delta + \mu - \beta_2 V_0 \lambda) + x^2(\alpha + 2\mu + \delta + \gamma + \gamma_1 - S_0\beta_1 - V_0\beta_2\lambda)$

$- S_0 x(\alpha\beta_2 + \beta_1(\mu + \delta + \gamma + \gamma_1)) + x^2(\mu + \phi) + (\alpha + \mu)(\mu + \phi)(\mu + \gamma + \gamma_1 + \delta - \beta_2 V_0 \lambda)$

$+ x(\mu + \phi)(2\mu + \alpha + \delta + \gamma + \gamma_1 - \beta_1 S_0 - V_0\beta_2\lambda) - S_0(\mu + \phi)(\alpha\beta_2 + \beta_1(\mu + \delta + \gamma + \gamma_1))\} = 0$

$\Rightarrow \quad (x + \mu^3)(x^3 + A_1 x^2 + A_2 x + A_3) = 0.$

Where

$$\begin{aligned} A_1 &= (\alpha + 2\mu + \delta + \gamma + \gamma_1 - S_0\beta_1 - V_0\beta_2\lambda) + (\mu + \phi). \\ A_2 &= (\alpha + \mu)(\gamma + \gamma_1 + \mu + \delta - \beta_2 V_0 \lambda) - S_0(\alpha\beta_2 + \beta_1(\mu + \delta + \gamma + \gamma_1)) + \\ &\quad (\mu + \phi)(2\mu + \alpha + \delta + \gamma + \gamma_1 - \beta_1 S_0 - V_0\beta_2\lambda). \\ A_3 &= (\alpha + \mu)(\mu + \phi)(\mu + \gamma + \gamma_1 + \delta - \beta_2 V_0 \lambda) - S_0(\mu + \phi)(\alpha\beta_2 + \beta_1(\mu + \delta + \gamma + \gamma_1)). \end{aligned}$$

According to the Routh-Hurwitz criterion [27, 42], all roots of the cubic equation (second factor of the polynomial expression), possess negative real part if and only if $A_1, A_2, A_3 > 0$ and $A_1 A_2 - A_3 > 0$. It reflects $A_1 > 0$, and after expressing $A_2$ in terms of $\mathcal{R}_0$ we have

obtained,

$$A_2 = (\alpha + \mu)(\mu + \phi)(\mu + \delta + \gamma + \gamma_1)(1 - \mathcal{R}_0) + (\mu + \phi)(\alpha + \mu)(\mu + \delta + \gamma + \gamma_1)(1 - \mathcal{R}_0) + 2\mu(\mu + \phi).$$

Thus, for $A_2 > 0$, it is necessary that $\mathcal{R}_0 < 1$. Similarly, we write $A_1$ as in terms of $\mathcal{R}_0$ as follows,

$$A_1 = (\alpha + \mu) + (\mu + \phi) + (\alpha + \mu)(\mu + \phi)(\mu + \delta + \gamma + \gamma_1)(1 - \mathcal{R}_0),$$

and using the previous condition if $\mathcal{R}_0 < 1$ we find that $A_1 > 0$. Moreover, we write $A_3$ as in terms of $\mathcal{R}_0$,

$$A_3 = (\alpha + \mu)(\mu + \phi)(\mu + \delta + \gamma + \gamma_1)(1 - \mathcal{R}_0).$$

Hence, we find that $A_3 > 0$ if $\mathcal{R}_0 < 1$. Clearly, for $\mathcal{R}_0 < 1$ we obtain that $A_1 A_2 - A_3 > 0$, and the Routh-Hurwitz criteria are satisfied. The characteristic roots (eigenvalues) of the Jacobian matrix at $J(\mathcal{E}^0)$ are $-\mu, -\mu, -\mu$, and the rest of them are determined by the nature of the coefficients of the cubic polynomials described above. Thus, $\mathcal{R}_0 < 1$ implies that all the eigenvalues $\lambda_i (i = 1, 2, \cdots, 6)$ of the linearized system are negative. Conversely, for the case $\mathcal{R}_0 > 1$, then there is at least one positive eigenvalue in the linearized system, and the equilibrium becomes unstable [6]. Hence, the DFE state $(\mathcal{E}^0)$ is locally asymptotically stable if $\mathcal{R}_0 < 1$ and unstable if $\mathcal{R}_0 > 1$.

The above analysis reveals that when $\mathcal{R}_0 = 1$ then the above analysis fails. The scenario $\mathcal{R}_0 = 1$ is equivalent to,

$$\beta_2 = \beta_2^{[TC]} = \frac{(\mu + \phi)(\alpha + \mu)(\mu + \delta + \gamma + \gamma_1) - \mu N \beta_1 (\mu + \delta + \gamma + \gamma_1)}{\phi N \lambda (\alpha + \mu) + \mu N \alpha}.$$

In the previous section, we have shown that model system (1) passes through transcritical bifurcation at disease-free state $(\mathcal{E}^0)$ when model parameter $\beta_2$ undergoes its critical value $\beta_2 = \beta_2^{[TC]}$.

## Global stability of DFE state $(\mathcal{E}^0)$

**Theorem 4**. *When $\mathcal{R}_0 < 1$, the DFE $\mathcal{E}^0$ is globally asymptotically stable. When $\mathcal{R}_0 > 1$, $\mathcal{E}^0$ is unstable.*

To prove this theorem we use the following lemma.

**Lemma 1**. [4]. *(Global stability of DFE) Consider the model written in the form,*

$$\frac{dX_1}{dt} = F(X_1, X_2),$$

$$\frac{dX_2}{dt} = G(X_1, X_2), \quad G(X_1, 0) = 0.$$

*Let $X_1 \in \mathbb{R}^m$ represent the number of uninfected individuals. Similarly, $X_2 \in \mathbb{R}^m$ signifies the count of infected people, encompassing latent, infectious, and other categories. $X_0 = (X_1^*)$ is the DFE for the system (1). Furthermore, the following conditions (H1) and (H2) are assumed:*

*(H1) $X_1^*$ is globally asymptotically stable when $\frac{dX_1}{dt} = F(X_1, 0)$.*

*(H2) For $G(X_1, X_2) = AX_2 - \hat{G}(X_1, X_2)$, $\hat{G}(X_1, X_2) \geq 0$ for $(X_1, X_2) \in \Omega$. Where the Jacobian $A = \frac{\partial G}{\partial X_2}(X_1^*, 0)$ is an M-matrix (the off-diagonal elements of A are non-negative), and $\Omega$ is*

*the region where the model makes biological sense. Then the DFE $X_0 = (X_1^*, 0)$ is globally asymptotically stable provided that $\mathcal{R}_0 < 1$.*

**Theorem 5**. *The disease-free steady state $(\mathcal{E}^0)$ of the model* (1) *is globally asymptotically stable when $\mathcal{R}_0 < 1$ and the disease dies out.*

The proof for Theorem 4, and 5 is displayed in the supporting section.

## Local stability of EE state $(\mathcal{E}^*)$

**Theorem 6**. *The endemic equilibrium point $\mathcal{E}^*(S^*, V^*, E^*, I^*, R^*, T^*)$ is locally asymptotically stable if $\mathcal{R}_0 > 1$ and unstable if $\mathcal{R}_0 < 1$.*

*Proof.* The analysis for $\mathcal{E}^*$ is similar to that of $\mathcal{E}^0$. The Jacobian of the system at the EE point (Linearizing (1) about $\mathcal{E}^*$) $\mathcal{E}^*(S^*, V^*, E^*, I^*, R^*, T^*)$ is as follows,

$$J(\mathcal{E}^*) = \begin{pmatrix} a_{11} & 0 & -\beta_1 S^* & -\beta_2 S^* & 0 & 0 \\ \phi & a_{22} & -\lambda\beta_1 V^* & -\lambda\beta_2 V^* & 0 & 0 \\ (\beta_1 E^* + \beta_2 I^*) & 0 & a_{33} & \beta_2 S^* & 0 & 0 \\ 0 & a_{42} & \alpha + \lambda\beta_1 V^* & a_{44} & 0 & 0 \\ 0 & 0 & 0 & \gamma & -\mu & 0 \\ 0 & 0 & 0 & \gamma_1 & 0 & -\mu \end{pmatrix} \tag{19}$$

where $a_{11} = -(\beta_1 E^* + \beta_2 I^*) - (\mu + \phi)$, $a_{22} = -\lambda(\beta_1 E^* + \beta_2 I^*) - \mu$, $a_{33} = \beta_1 S^* - (\alpha + \mu)$, $a_{42} = \lambda(\beta_1 E^* + \beta_2 I^*)$, and $a_{44} = \lambda\beta_2 V^* - (\mu + \delta + \gamma + \gamma_1)$.

At the EE state $\mathcal{E}^*$, by calculating the Jacobian matrix *J*, after that by solving $|(J - x\mathbb{I})| = 0$, the two eigenvalues are $-\mu$, $-\mu$ which are negative. Rest are the roots of the characteristic polynomial of the matrix,

$$J_1(\mathcal{E}^*) = \begin{pmatrix} a_{11} & 0 & -\beta_1 S^* & -\beta_2 S^* \\ \phi & a_{22} & -\lambda\beta_1 V^* & -\lambda\beta_2 V^* \\ (\beta_1 E^* + \beta_2 I^*) & 0 & \beta_1 S^* - (\alpha + \mu) & \beta_2 S^* \\ 0 & \lambda(\beta_1 E^* + \beta_2 I^*) & \alpha + \lambda\beta_1 V^* & a_{44} \end{pmatrix} \tag{20}$$

where $a_{11} = -(\beta_1 E^* + \beta_2 I^*) - (\mu + \phi)$, $a_{22} = -\lambda(\beta_1 E^* + \beta_2 I^*) - \mu$, and $a_{44} = \lambda\beta_2 V^* - (\mu + \delta + \gamma + \gamma_1)$.

The characteristic equation is obtained as follows,

$$x^4 + A_1 x^3 + A_2 x^2 + A_3 x + A_4 = 0.$$

Expanding the expressions and arranging them in terms of powers of $x$, these coefficients of the equation ultimately simplify to,

$$A_1 = \alpha - S^*\beta_1 + I^*\beta_2 + \delta + \gamma + \gamma_1 + I^*\beta_2\lambda - V^*\beta_2\lambda + E^*\beta_1(1+\lambda) + 4(\mu+\phi).$$

$$A_2 = -S^*\alpha\beta_2 + \alpha\gamma - S^*\beta_1\gamma + \alpha\gamma_1 - S^*\beta_1\gamma_1 + \alpha\delta - S^*\beta_1\delta + E^{*2}\beta_1^2\lambda - V^*\alpha\beta_2\lambda + I^{*2}\beta_2^2\lambda +$$
$$3\alpha\mu - 3S^*\beta_1\mu + 3\gamma\mu + 3\gamma_1\mu + 3\delta\mu + 6\mu^2 + \alpha\phi - S^*\beta_1\phi + \gamma\phi + \gamma_1\phi + \delta\phi -$$
$$V^*\beta_2\lambda\phi + 3\mu\phi + I^*\beta_2(\alpha + \gamma + \gamma_1 + \delta + \alpha\lambda - S^*\beta_1\lambda - V^*\beta_2\lambda + \gamma\lambda + \gamma_1\lambda + \delta\lambda + 3\mu + 3\lambda\mu + \lambda\phi)$$
$$+E^*\beta_1(\alpha + \gamma + \gamma_1 + \delta + \alpha\lambda - S^*\beta_1\lambda + 2I^*\beta_2\lambda - V^*\beta_2\lambda + \gamma\lambda + \gamma_1\lambda + \delta\lambda + 3\mu + 3\lambda\mu + \lambda\phi).$$

$$A_3 = -2S^*\alpha\beta_2\mu + 2\alpha\gamma\mu - 2S^*\beta_1\gamma\mu + 2\alpha\gamma_1\mu - 2S^*\beta_1\gamma_1\mu + 2\alpha\delta\mu - 2S^*\beta_1\delta\mu - 2V^*\alpha\beta_2\lambda\mu + 3\alpha\mu^2 -$$
$$3S^*\beta_1\mu^2 + 3\gamma\mu^2 + 3\gamma_1\mu^2 + 3\delta\mu^2 - 3V^*\beta_2\lambda\mu^2 + 4\mu^3 + E^{*2}\beta_1^2\lambda(\alpha + \gamma + \gamma_1 + \delta + 2\mu) +$$
$$I^{*2}\beta_2^2\lambda(2\mu + \alpha + \gamma + \gamma_1 + \delta) - S^*\alpha\beta_2\phi + \alpha\gamma\phi - S^*\beta_1\gamma\phi + \alpha\gamma_1\phi - S^*\beta_1\gamma_1\phi + \alpha\delta\phi -$$
$$S^*\beta_1\delta\phi - V^*\alpha\beta_2\lambda\phi + 2\alpha\mu\phi - 2S^*\beta_1\mu\phi + 2\gamma\mu\phi + 2\gamma_1\mu\phi + 2\delta\mu\phi - 2V^*\beta_2\lambda\mu\phi + 3\mu^2\phi +$$
$$I^*\beta_2(2\gamma\mu + 2\gamma_1\mu + 2\delta\mu - 2V^*\beta_2\lambda\mu + 2\gamma\lambda\mu + 2\gamma_1\lambda\mu + 2\delta\lambda\mu + 3\mu^2 + 3\lambda\mu^2 + \gamma\lambda\phi + \gamma_1\lambda\phi + \delta\lambda\phi +$$
$$2\lambda\mu\phi + \alpha(\gamma + \gamma_1 + \delta - S^*\beta_2\lambda - V^*\beta_2\lambda + \gamma\lambda + \gamma_1\lambda + \delta\lambda + 2\mu + 2\lambda\mu + \lambda\phi) -$$
$$S^*\lambda(-\beta_2\phi + \beta_1(\gamma + \gamma_1 + \delta + 2\mu + \phi))) + E^*\beta_1(2I^*\beta_2\gamma\lambda + 2I^*\beta_2\gamma_1\lambda + 2I^*\beta_2\delta\lambda + 2\gamma\mu + 2\gamma_1\mu +$$
$$2\delta\mu + 4I^*\beta_2\lambda\mu - 2V^*\beta_2\lambda\mu + 2\gamma\lambda\mu + 2\gamma_1\lambda\mu + 2\delta\lambda\mu + 3\mu^2 + 3\lambda\mu^2 + \gamma\lambda\phi + \gamma_1\lambda\phi + \delta\lambda\phi +$$
$$2\lambda\mu\phi + \alpha(\gamma + \gamma_1 + \delta + 2I^*\beta_2\lambda - S^*\beta_2\lambda - V^*\beta_2\lambda + \gamma\lambda + \gamma_1\lambda + \delta\lambda + 2\mu + 2\lambda\mu + \lambda\phi) -$$
$$S^*\lambda(-\beta_2\phi + \beta_1(2\mu + \gamma + \gamma_1 + \delta + \phi))).$$

$$A_4 = I^{*2}\alpha\beta_2^2\gamma\lambda + I^{*2}\alpha\beta_2^2\gamma_1\lambda + I^{*2}\alpha\beta_2^2\delta\lambda + I^*\alpha\beta_2\gamma\mu + I^*\alpha\beta_2\gamma_1\mu + I^*\alpha\beta_2\delta\mu + I^{*2}\alpha\beta_2^2\lambda\mu - I^*S^*\alpha\beta_2^2\lambda\mu$$
$$-I^*V^*\alpha\beta_2^2\lambda\mu + I^*\alpha\beta_2\gamma\lambda\mu - I^*S^*\beta_1\beta_2\gamma\lambda\mu + I^{*2}\beta_2^2\gamma\lambda\mu + I^*\alpha\beta_2\gamma_1\lambda\mu - I^*S^*\beta_1\beta_2\gamma_1\lambda\mu +$$
$$I^{*2}\beta_2^2\gamma_1\lambda\mu + I^*\alpha\beta_2\delta\gamma\mu - I^*S^*\beta_1\beta_2\delta\lambda\mu + I^{*2}\beta_2^2\delta\lambda\mu + I^*\alpha\beta_2\mu^2 - S^*\alpha\beta_2\mu^2 + \alpha\gamma\mu^2 - S^*\beta_1\gamma\mu^2 - I^*\beta_2\gamma\mu^2$$
$$+\alpha\gamma_1\mu^2 - S^*\beta_1\gamma_1\mu^2 + I^*\beta_2\gamma_1\mu^2 + \alpha\delta\mu^2 - S^*\beta_1\delta\mu^2 + I^*\beta_2\delta\mu^2 + I^*\alpha\beta_2\lambda\mu^2 - V^*\alpha\beta_2\lambda\mu^2 -$$
$$I^*S^*\beta_1\beta_2\lambda\mu^2 + I^{*2}\beta_2^2\lambda\mu^2 - I^*V^*\beta_2^2\lambda\mu^2 + I^*\beta_2\gamma\lambda\mu^2 + I^*\beta_2\gamma_1\lambda\mu^2 + I^*\beta_2\delta\lambda\mu^2 + \alpha\mu^3 - S^*\beta_1\mu^3 +$$
$$I^*\beta_2\mu^3 + \gamma\mu^3 + \gamma_1\mu^3 + \delta\mu^3 + I^*\beta_2\lambda\mu^3 - V^*\beta_2\lambda\mu^3 + \mu^4 + E^{*2}\beta_1^2\lambda(\alpha + \mu)(\mu + \delta + \gamma + \gamma_1) +$$
$$I^*\alpha\beta_2\gamma\lambda\phi - I^*S^*\beta_1\beta_2\gamma\lambda\phi + I^*\alpha\beta_2\gamma_1\lambda\phi - I^*S^*\beta_1\beta_2\gamma_1\lambda\phi + I^*\alpha\beta_2\delta\lambda\phi - I^*S^*\beta_1\beta_2\delta\lambda\phi -$$
$$S^*\alpha\beta_2\mu\phi + \alpha\gamma\mu\phi - S^*\beta_1\gamma\mu\phi + \alpha\gamma_1\mu\phi - S^*\beta_1\gamma_1\mu\phi + \alpha\delta\mu\phi - S^*\beta_1\delta\mu\phi + I^*\alpha\beta_2\lambda\mu\phi -$$
$$V^*\alpha\beta_2\lambda\mu\phi - I^*S^*\beta_1\beta_2\lambda\mu\phi + I^*S^*\beta_2^2\lambda\mu\phi + I^*\beta_2\gamma\lambda\mu\phi + I^*\beta_2\gamma_1\lambda\mu\phi + I^*\beta_2\delta\lambda\mu\phi + \alpha\mu^2\phi - S^*\beta_1\mu^2\phi +$$
$$\gamma\mu^2\phi + \gamma_1\mu^2\phi + \delta\mu^2\phi + I^*\beta_2\lambda\mu^2\phi - V^*\beta_2\lambda\mu^2\phi + \mu^3\phi + E^*\beta_1(-S^*\beta_1\gamma\lambda\mu - S^*\beta_1\gamma_1\lambda\mu -$$
$$S^*\beta_1\delta\lambda\mu + \gamma\mu^2 + \gamma_1\mu^2 + \delta\mu^2 - S^*\beta_1\lambda\mu^2 - V^*\beta_2\lambda\mu^2 + \gamma\lambda\mu^2 + \gamma_1\lambda\mu^2 + \delta\lambda\mu^2 + \mu^3 + \lambda\mu^3 +$$
$$2I^*\beta_2\lambda(\alpha + \mu)(\gamma + \gamma_1 + \delta + \mu) - S^*\beta_1\gamma\lambda\phi - S^*\beta_1\gamma_1\lambda\phi - S^*\beta_1\delta\lambda\phi - S^*\beta_1\lambda\mu\phi + S^*\beta_2\lambda\mu\phi +$$
$$\gamma\lambda\mu\phi + \gamma_1\lambda\mu\phi + \delta\lambda\mu\phi + \lambda\mu^2\phi + \alpha(\delta\mu - S^*\beta_2\lambda\mu - V^*\beta_2\lambda\mu + \delta\lambda\mu + \mu^2 + \lambda\mu^2 + \delta\lambda\phi + \lambda\mu\phi +$$
$$\gamma(\mu + \lambda\mu + \lambda\phi) + \gamma_1(\mu + \lambda\mu + \lambda\phi))).$$

As a consequence of the Routh-Hurwitz criteria [4, 42], every root of this bi-quadratic equation possesses a negative real part if and only if $A_1, A_2, A_3, A_4 > 0$, $A_1A_2 > A_3$ and $A_1A_2A_3 > A_3^2 + A_1^2A_4$. For $\mathcal{R}_0 > 1$,

$$S_0\alpha\beta_2 + S_0\beta_1(\mu + \delta + \gamma + \gamma_1) + V_0\beta_2\lambda(\alpha + \mu) > (\alpha + \mu)(\mu + \delta + \gamma + \gamma_1).$$

Now, rewriting $A_1$ in terms of $\mathcal{R}_0$,

$$I^*\beta_2\lambda + E^*\beta_1(1 + \lambda) + 4(\mu + \phi) - \mu + (\alpha + \mu)(\mu + \delta + \gamma + \gamma_1)(\mathcal{R}_0 - 1).$$

Hence, the nature $\mathcal{R}_0 > 1$ is mandatory, in order to satisfy $A_1 > 0$ and $A_1^2 > 0$. Again, from the expression of $A_2$,

$$(\alpha + \mu)(\mu + \delta + \gamma + \gamma_1)(\mathcal{R}_0 - 1) + \lambda(\gamma + \gamma_1 + 3\mu + \phi + \delta + \alpha) + 2\mu.$$

Further, from the coefficient of $E^*\beta_1$ in $A_2$,

$$\alpha + \gamma + \gamma_1 + \delta + \alpha\lambda - S^*\beta_1\lambda + 2I^*\beta_2\lambda - V^*\beta_2\lambda + \gamma\lambda + \gamma_1\lambda + \delta\lambda + 3\mu + 3\lambda\mu + \lambda\phi$$
$$= \lambda(\gamma + \gamma_1 + 3\mu + \phi + \delta + \alpha) + 2\mu + (\alpha + \mu)(\mu + \delta + \gamma + \gamma_1)(\mathcal{R}_0 - 1).$$

Therefore, from the above expression we find that it requires $\mathcal{R}_0 > 1$ to satisfy $A_2 > 0$. Now, from the expression of $A_3$, the coefficient of $I^*\beta_2$ can be expressed in terms of $\mathcal{R}_0$ as follows,

$$\alpha((\alpha + \mu)(\mu + \delta + \gamma + \gamma_1)(\mathcal{R}_0 - 1) + \mu + \lambda(\gamma + \gamma_1 + 2\mu + \delta + \phi)) - \lambda(\alpha + \mu)(\mu + \delta + \gamma + \gamma_1)$$
$$(\mathcal{R}_0 - 1) + \lambda\mu\beta_1.$$

Thus, we find that it is necessary $\mathcal{R}_0 > 1$ in order to satisfy $A_3 > 0$ and $A_3^2 > 0$. Hence, it shows that $A_1A_2 > A_3$ for $\mathcal{R}_0 > 1$. Finally, from the expression of $A_4$ taking a portion as coefficient of $E^*\beta_1$,

$$\alpha(\delta\mu - S^*\beta_2\lambda\mu - V^*\beta_2\lambda\mu + \delta\lambda\mu + \mu^2 + \lambda\mu^2 + \delta\lambda\phi + \lambda\mu\phi + \gamma(\mu + \lambda\mu + \lambda\phi) + \gamma_1(\mu + \lambda\mu + \lambda\phi))$$
$$= \alpha(\lambda\mu(\alpha + \mu)(\mu + \delta + \gamma + \gamma_1)(\mathcal{R}_0 - 1) + \mu^2 + \delta\lambda\phi + (\gamma + \gamma_1)(\mu + \lambda\mu + \lambda\phi)).$$

Clearly, $A_4 > 0$ for $\mathcal{R}_0 > 1$. Therefore, from the Routh-Hurwitz Criterion [7, 16] for the Jacobian matrix with characteristic polynomial of degree $n = 4$, combining all the above expressions $A_1A_2A_3 > A_3^2 + A_1^2A_4$ is satisfied for $\mathcal{R}_0 > 1$.

Therefore, all the roots of the characteristic equation will possess a negative real part. Consequently, when the $\mathcal{R}_0 > 1$, the EE point will be locally asymptotically stable. In contrast, if the $\mathcal{R}_0 < 1$, suggests that the infected state is unstable as the Jacobian includes at least one positive eigenvalue. This establishes the validity of the conclusion.

**Theorem 7**. *The endemic equilibrium state* $(\mathcal{E}^*)$ *of the system* (4) *is globally asymptotically stable for* $\mathcal{R}_0 > 1$ *and the disease persists.*

*Proof.* The proof is excluded as a comparable result can be found in [27].

**Theorem 8**. *The EE state* $(\mathcal{E}^*)$ *of the model* (1) *is globally asymptotically stable provided that* $\mathcal{R}_0 > 1$.

*Proof.* Considering the model (1) and $\mathcal{R}_0 > 1$, so that the associated unique EE $\mathcal{E}^*$ of the model exists (which proved already). To examine the global stability of $\mathcal{E}^*$ we have considered the following non-linear Lyapunov function of the Goh-Voltera type,

$$V = \left(S - S^{**} - S^{**}\ln\frac{S}{S^{**}}\right) + \left(E - E^{**} - E^{**}\ln\frac{E}{E^{**}}\right) + K\left(I - I^{**} - I^{**}\ln\frac{I}{I^{**}}\right)$$

$$\text{where} \quad K = \frac{(\beta_1 + \beta_2)S^*I^*}{\alpha E^*}.$$

Notice that, $V$ is non-negative, and becomes identically zero if and only if it is evaluated at the non-negative EE state $\mathcal{E}^*$. By performing the derivative of V along the solution curves of (1) yields,

$$V' = S'\left(1 - \frac{S^*}{S}\right) + E'\left(1 - \frac{E^*}{E}\right) + KI'\left(1 - \frac{I^*}{I}\right).$$

Here, prime ($'$) denotes the derivatives. Substituting the derivatives ($S', E', I'$) into this equation from (1) we have,

$$V' = [\Lambda - (\beta_1 E + \beta_2 I)S - (\mu + \phi)S]\left(1 - \frac{S^*}{S}\right) + [(\beta_1 E + \beta_2 I)S - (\alpha + \mu)E]\left(1 - \frac{E^*}{E}\right) +$$
$$K[\alpha E - (\mu + \delta + \gamma + \gamma_1)I]\left(1 - \frac{I^*}{I}\right).$$

Here, we have considered the simplified case of our model by assuming $\lambda = (1 - \varepsilon) = 0$ i.e., the vaccination rate is 100 effective.

At the steady state from equation (4) we have,

$$\Lambda = (\beta_1 E^* + \beta_2 I^*)S^* + (\mu + \phi)S^*, \text{ and } (\beta_1 E^* + \beta_2 I^*)S^* = (\alpha + \mu)E^*.$$

Substituting these relations in the expression of $V'$ we have obtained that,

$$V' = \left[(\beta_1 E^* + \beta_2 I^*)S^* + (\mu + \phi)S^* - (\beta_1 E + \beta_2 I)S - (\mu + \phi)S - (\beta_1 E^* + \beta_2 I^*)\frac{S^{*2}}{S} -\right.$$
$$\left.\frac{(\mu + \phi)S^{*2}}{S} + (\beta_1 E + \beta_2 I)S^* + (\mu + \phi)S^*\right] + \left[\beta_1 S(E - E^*) + \beta_2 IS\left(1 - \frac{E^*}{E}\right) -\right.$$
$$A_1(E - E^*)\Big] + \left[K\alpha E\left(1 - \frac{I^*}{I}\right) - KA_2(I - I^*)\right].$$

Here, $(\alpha + \mu) = A_1$, and $(\mu + \delta + \gamma + \gamma_1) = A_2$. By collecting all the infected classes without the star (*) and equating to zero we obtain,

$$-(\beta_1 E + \beta_2 I)S + (\beta_1 E + \beta_2 I)S^* + \beta_1 SE + \beta_2 IS - A_1 E + K\alpha E - KA_2 I = 0.$$

A little perturbation of steady state results in,

$$K = \frac{S^*(\beta_1 + \beta_2)}{A_2}, \ A_1 = \frac{(\beta_1 + \beta_2)S^*I^*}{E^*}, \ \alpha = \frac{A_2 I^*}{E^*}. \tag{21}$$

Substituting the expression from (21) into the expression of $V'$,

$$
\begin{aligned}
V' &= [(\beta_1 E^* + \beta_2 I^*)S^* + 2(\mu + \phi)S^* - (\beta_1 E + \beta_2 I)S] + \\
&\quad \left[ -(\mu + \phi)S - (\beta_1 E^* + \beta_2 I^*)\frac{S^{*2}}{S} - \frac{(\mu + \phi)S^{*2}}{S} + (\beta_1 E + \beta_2 I)S^* \right] + \\
&\quad \left[ \beta_1 S(E - E^*) + \beta_2 IS\left(1 - \frac{E^*}{E}\right) - \frac{(\beta_1 + \beta_2)S^* I^*}{E^*}(E - E^*) \right] + \\
&\quad \left[ \frac{S^*(\beta_1 + \beta_2)I^*}{E^*}\left(1 - \frac{I^*}{I}\right) - S^*(\beta_1 + \beta_2)(I - I^*) \right]. \\
&= (\mu + \phi)S^*\left[ 2 - \frac{S}{S^*} - \frac{S^*}{S} \right] + \\
&\quad \beta_1 E^* S^*\left[ 1 - \frac{S^*}{S} + \frac{E}{E^*} - \frac{S}{S^*} - \frac{E}{E^{*2}} + \frac{1}{E^*} + \frac{I^*}{E^{*2}} - \frac{I^{*2}}{E^{*2}I} - \frac{I}{E^*} + \frac{I^*}{E^*} \right] + \\
&\quad \beta_2 I^* S^*\left[ 3 - \frac{S^*}{S} + \frac{I}{I^*} - \frac{I}{I^*}\frac{S}{S^*}\frac{E^*}{E} - \frac{E}{E^*} + \frac{1}{E^*} - \frac{I^*}{E^* I} - \frac{I}{I^*} \right].
\end{aligned}
$$

Hence, we have got the new form of $V'$. Now, the coefficient of $\beta_1 E^* S^*$ gives,

$$
\left(1 - \frac{S^*}{S} - \frac{S}{S^*}\right) + \left(\frac{E}{E^*} + \frac{1}{E^*} - \frac{E}{E^*}\right) + \frac{I^*}{E^{*2}}\left(1 - \frac{I^*}{I}\right) + \left(\frac{I^*}{E^*} - \frac{I}{E^*}\right) \le 0.
$$

Finally, as the arithmetic mean exceeds the geometric mean. Hence, the following inequality from immediate expression of $V^*$ (for $n = 3$) results into,

$$
\left(2 - \frac{S}{S*} - \frac{S^*}{S}\right) \le 0, \text{ and } \left(3 - \frac{S^*}{S} + \frac{I}{I^*} - \frac{I}{I^*}\frac{S}{S^*}\frac{E^*}{E} - \frac{E}{E^*} + \frac{1}{E^*} - \frac{I^*}{E^* I} - \frac{I}{I^*}\right) \le 0,
$$

which shows that each of the resulting terms is non-positive. Thus, with these conditions we conclude that $V'(t) \le 0$ for all positive values of $\{S, E, I\}$ i.e., if $S = S^*$, $E = E^*$, $I = I^*$, and also for $\mathcal{R}_0 > 1$. Moreover, the strict equality $V' = 0$ holds only for $S = S^*$, $E = E^*$, and $I = I^*$.

Hence, the maximum invariance set $\{(S, E, I) \in \Omega : V'(t) = 0\}$ is the singleton $\{\mathcal{E}^*\}$, where $\{\mathcal{E}^*\}$ is the endemic equilibrium point.

Thus, as a consequence of LaSalle's Invariance principle, the EE point $(\mathcal{E}^*)$ is globally asymptotically stable in the set $\Omega$ when $\mathcal{R}_0 > 1$. In other words, every solution to the equations of the model (1) converges to the corresponding unique endemic equilibria $(\mathcal{E}^*)$, of the model as $t \to \infty$ for $\mathcal{R}_0 > 1$.

## Stability and persistence of the system

**Theorem 9**. *The DFE $\mathcal{E}^0$ of the model* (1) *is a global attractor.*

*Proof.* Omitting the proof, as a similar result can be referenced in [43].

**Theorem 10**. *If $\mathcal{R}_0 > 1$, the system described by* (1) *exhibits uniform persistence, implying the existence of a constant $\xi > 0$. This constant ensures that for any initial data $\xi$ in $\Omega$:*
$\liminf\limits_{t \to \infty} S(t) > \xi$, $\liminf\limits_{t \to \infty} E(t) > \xi$, $\liminf\limits_{t \to \infty} I(t) > \xi$, *and* $\liminf\limits_{t \to \infty} V(t) > \xi$. *Remarkably, the value of $\xi$ remains independent of the initial data in $\Omega$.*

*Proof.* When $t \to \infty$ from system (1), we have the following limiting system,

$$\begin{cases} S' = \Lambda - \dfrac{\mu}{\Lambda}(\beta_1 \xi E + \beta_2 I)S - (\mu + \phi)S. \\[2mm] V' = \phi S - \dfrac{\mu}{\Lambda}(1 - \varepsilon)(\beta_1 \xi E + \beta_2 I)V - \mu V. \\[2mm] E' = \dfrac{\mu}{\Lambda}(\beta_1 \xi E + \beta_2 I)S - (\alpha + \mu)E. \\[2mm] I' = \alpha E + \dfrac{\mu}{\Lambda}(1 - \varepsilon)(\beta_1 \xi E + \beta_2 I)V - (\mu + \delta + \gamma + \gamma_1)I. \end{cases} \quad (22)$$

For the case of notation, we still use the notation $\mathcal{E}^0$ to denote the DFE of the equation (22). Define,

$$\begin{aligned} X = &\ \{(S, E, I, V) : S \geq 0, E \geq 0, I \geq 0, V \geq 0\}, \\ X_0 = &\ \{(S, E, I, V) : S > 0, E > 0, I > 0, V > 0\}, \text{ and} \\ \partial X_0 = &\ X \setminus X_0. \end{aligned}$$

It is often suffices to show that (22) is uniformly persistent with respect to $(X_0, \partial X_0)$.

Firstly, by the form of (22), it is easy to see that both $X$ and $X_0$ are positively invariant. $\partial X_0$ is relatively closed in $X$ and system (22) is point dissipative. Consider the following set using solutions $(S(t), V(t), E(t), I(t))$ of the system (22).

$$M_\partial = \{(S(0), E(0), I(0), V(0)) : (S(t), E(t), I(t), V(t)) \in \partial X_0, \forall\, t \geq 0\}.$$

We now show that,

$$M_\partial = \{S, V, 0, 0 : S \geq 0\ ,\ V \geq 0\}. \quad (23)$$

Assume that $(S(0), V(0), E(0), I(0)) \in M_\partial$. It suffices to show that $E(t) = I(t) = 0$ for all $t \geq 0$.

Suppose not, then there exists a $t_0 \geq 0$ such that $(E(t_0) > 0, I(t_0) = 0)$ or, $(E(t_0) = 0, I(t_0) > 0)$.

For, $(E(t_0) > 0, I(t_0) = 0)$ we have,

$$I'(t_0) = \alpha \varepsilon E(t_0) > 0.$$

It follows that there is an $\epsilon_0 > 0$ such that $I(t) > 0$ for $t_0 < t < t_0 + \epsilon_0$. Clearly, we can restrict $\epsilon_0 > 0$ small enough such that $E(t_0) > 0$ for $t_0 < t < t_0 + \epsilon_0$. This means that $(S(t), V(t), E(t), I(t)) \notin \partial X_0$ for $t_0 < t < t_0 + \epsilon_0$, which contradicts the assumption that $(S(0), V(0), E(0), I(0)) \in M_\partial$. For other cases, we can show these contradict the assumption that $(S(0), V(0), E(0), I(0)) \in M_\partial$ respectively. Thus (23) holds.

Note that, $\mathcal{E}^0$ (DFE point) is globally asymptotically stable in Int $M_\partial$ (interior of $M_\partial$). Moreover, $\mathcal{E}^0$ is isolated invariant set in $X$, every orbit in $M_\partial$ converges to $\mathcal{E}^0$ and $\mathcal{E}^0$ is acyclic in $M_\partial$. We only need to show that $W^s(\mathcal{E}^0) \cap X_0 = \emptyset$ if $\mathcal{R}_0 > 1$.

In the following, we prove that $W^s(\mathcal{E}^0) \cap X_0 = \emptyset$. Suppose not, that is, $W^s(\mathcal{E}^0) \cap X_0 \neq \emptyset$. Then there exists a positive solution $(\tilde{S}(t), \tilde{V}(t), \tilde{E}(t), \tilde{I}(t))$, with $(\tilde{S}(0), \tilde{V}(0), \tilde{E}(0), \tilde{I}(0)) \in X_0$ such that $(\tilde{S}(t), \tilde{V}(t), \tilde{E}(t), \tilde{I}(t)) \to \mathcal{E}^0$ as $t \to +\infty$. Since, $\mathcal{R}_0 > 1$, we can choose a $\eta > 0$ small enough such that,

$$\mathcal{R}_0 - \eta \frac{\mu}{\Lambda}\mathcal{R}_0 > 1.$$

Thus, when $t$ is sufficiently large, we have,

$$S_0 - \eta \leq \tilde{S}(t) \leq S_0 + \eta, \ 0 \leq \tilde{V}(t) \leq \eta, \ 0 \leq \tilde{E}(t) \leq \eta, \ 0 \leq \tilde{I}(t) \leq \eta, \ \text{and}$$

$$\begin{cases} V' = \phi(S_0 - \eta) - \dfrac{\mu}{\Lambda}(1 - \varepsilon)(\beta_1 \xi E + \beta_2 I)(S_0 - \eta) - \mu V. \\[2mm] E' = \dfrac{\mu}{\Lambda}(\beta_1 \xi E + \beta_2 I)(S_0 - \eta) - (\alpha + \mu)E. \\[2mm] I' = \alpha E + \dfrac{\mu}{\Lambda}(1 - \varepsilon)(\beta_1 \xi E + \beta_2 I)(S_0 - \eta) - (\mu + \delta + \gamma + \gamma_1)I. \end{cases}$$

By the comparison principle, it is easy to see that $\tilde{E}(t) \to +\infty$, $\tilde{I}(t) \to +\infty$ as $t \to +\infty$, which contradicts $\tilde{E}(t) \to 0$, $\tilde{I}(t) \to 0$ as $t \to +\infty$. This proves that,

$$W^s(\mathcal{E}^0) \cap X_0 = \emptyset.$$

Since $W^s(\mathcal{E}^0) \cap X_0 = \emptyset$, $\cup_{x \in M_\partial} \omega(x) = \{\mathcal{E}^0\}$, $\mathcal{E}^0$ is isolated invariant set in $X$, and $\mathcal{E}^0$ is acyclic in $M_\partial$, thus we are able to conclude that the system (22) is uniformly persistent with respect to $(X_0, \partial X_0)$. Then, the system (1) is uniformly persistent.

## Phase plane analysis, contour plot analysis, and box plot analysis

In this section, we numerically analyze model (1) to support previous analytical results. Simulations using real data enhance model accuracy and help optimize parameters by comparing predictions with observed outcomes. This allows us to evaluate intervention effectiveness, such as vaccination or social distancing, and forecast epidemic progression under different scenarios. We also graphically present the relationship between parameters and $\mathcal{R}_0$ and include a phase plane analysis to illustrate various scenarios.

### Multi-compartment analysis based on threshold values

In epidemiology, the compartmental model is typically used to analyze the spread of infectious diseases in the community. We can provide an interpretation of the scenario when the basic reproduction number ($\mathcal{R}_0$) is less or greater than 1 in the context of the SVEIRT model.

Here is how the scenario from Fig 6 is interpreted for the compartments in model (1).

At first, most people in the population are susceptible to the disease. As the epidemic progresses, some individuals become infected, while others gain immunity through vaccination or natural recovery. The number of infected individuals will initially rise, indicating the epidemic's growth phase. However, since $\mathcal{R}_0 < 1$ the rate of new infections will decrease over time, as shown by the model parameters in Table 1. This decline can result from public health interventions, social distancing, or increasing immunity in the population. Meanwhile, the number of recovered individuals will gradually increase as infected individuals either recover or die. Fig 6 (a) shows that the total number of infected individuals decreases faster with treatment than without treatment.

As the epidemic slows down and the number of new infections decreases, the number of recoveries will eventually surpass the number of new infections, leading to a decline in active cases. In this case, exposed and vaccinated individuals contribute to reducing the susceptible population and can further decrease the transmission rate of the disease. We have used $\beta_1 = 0.0047$ and $\beta_2 = 0.0052$ for the simulations in Fig 6.

As $\mathcal{R}_0 < 1$, the epidemic curve will eventually reach its peak and then start to decline. From Fig 6(a), we see that from 1 to 5 weeks $S(t)$ population decreases to 50, and after that increases

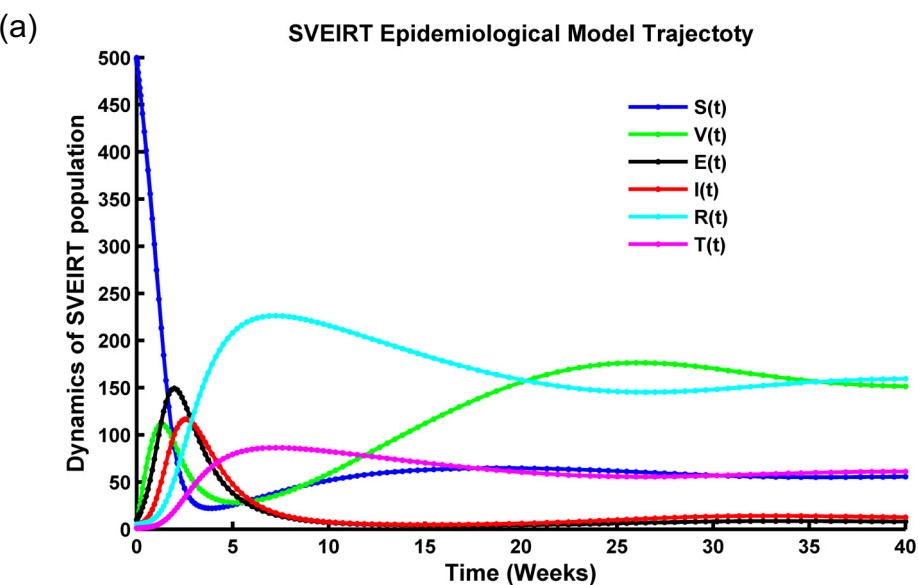

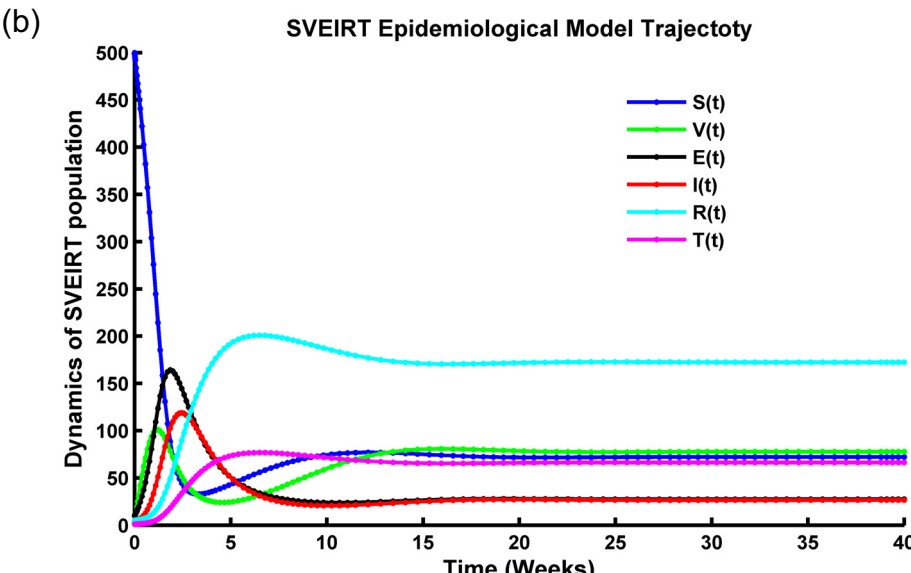

**Fig 6. Simulation of the population dynamics of $S(t)$, $V(t)$, $E(t)$, $I(t)$, $R(t)$, $T(t)$ over time, with initial conditions $S(0) = 500$, $V(0) = 1$, $E(0) = 1$, $I(0) = 1$, $R(0) = 0$, $T(0) = 1$, is conducted for two scenarios: (a) $\mathcal{R}_0 < 1$ and (b) $\mathcal{R}_0 > 1$. All parameter values are sourced from Table 1.**

up to 80. The Exposed population goes to the pick level after 5 weeks ($E(t) = 150$), and the same scenario for the infected population after 5 weeks ($I(t) = 120$).

Thus, Fig 6(a) illustrates that after a long time, the total number of exposed and infected populations converges to zero level.

Also, the recovered and treated population gradually increased after 5 weeks, and after 20 weeks $R(t)$ and $T(t)$ population goes parallelly. Hence, numerical analysis supports the analytical result at DFE when $\mathcal{R}_0 < 1$.

When the basic reproduction number $\mathcal{R}_0 > 1$, it indicates that each infected individual, on average, is infecting more than one susceptible individual.

In an $\mathcal{R}_0 > 1$ scenario, the number of infected individuals initially increases rapidly, indicating a growing outbreak (Fig 6(b)). As time progresses and interventions take place, the number of infected individuals may eventually decrease. Further, in the early stages of the epidemic, the number of recovered individuals is low. However, as the epidemic progresses, the number of recoveries gradually increases. The number of exposed individuals initially increases, reflecting the transmission from infected individuals. After the latent period, they transition to the infected compartment. The number of vaccinated individuals increases over time, reducing the susceptible population and potentially slowing down the spread of the disease.

From Fig 6(b) we see that the more $E(t)$ and $I(t)$ decreases, the $R(t)$ and $T(t)$ population grows significantly in the community. Also, we see from Fig 6(b) that, when $\mathcal{R}_0 > 1$ the total susceptible, vaccinated, recovered, and treated population remains in the community. After a long time, the infected and exposed population became parallel to the susceptible population and never approached zero level. Therefore, Fig 6(b) presented for $\mathcal{R}_0 > 1$, supports the stability of EE point numerically. We observe that EE of the model (1) exists and it is locally asymptotically stable for $\mathcal{R}_0 > 1$. This signifies an epidemic scenario where the disease is spreading rapidly within the population i.e. disease persists in the community.

## Phase plane analysis and the role of significant parameters

The phase plane is a graphical representation that allows us to analyze the dynamics of a system of differential equations, such as those used in infectious disease modeling. It helps visualize the interactions and trajectories of different compartments (e.g. susceptible, infected) over time. By examining the stability of equilibrium points and the shape of trajectories, phase plane analysis provides valuable information for understanding disease spread, identifying critical thresholds, and evaluating the impact of interventions in a concise and intuitive manner [31, 32, 44].

Fig 7(a) illustrates that when the contact rate increases in the phase plane analysis, it affects the trajectory and behavior of the susceptible compartment. Specifically, an increase in the contact rate leads to a steeper slope or gradient in the direction of the susceptible compartment axis. This steeper slope indicates that the rate at which individuals transition from the susceptible compartment to the infected compartment increases. In other words, as the contact rate rises, more susceptible individuals become infected at a faster pace. The trajectory of the susceptible compartment in the phase plane will show a more pronounced downward trend as time progresses, indicating a more rapid depletion of the susceptible population. This implies that the disease is spreading more rapidly within the population due to increased contact between infected and susceptible individuals.

Fig 7(b)–7(d) show an increase in the contact rate can lead to a higher influx of individuals into the exposed compartment. The trajectory of the exposed compartment in the phase plane may exhibit a steeper slope, indicating a faster accumulation of individuals in the exposed state. If the contact rate is significantly higher than the recovery rate, the exposed compartment grows rapidly, reaching a higher peak before declining. On the other hand, with an increased contact rate, the rate of individuals transitioning from the exposed compartment to the infected compartment will be higher. The trajectory of the infected compartment may show a steeper slope, indicating a faster rise in the number of infected individuals. If the contact rate surpasses the recovery rate, the infected compartment may continue to grow without reaching a peak, resulting in a sustained or increasing number of infected individuals.

In phase plane analysis, the trajectory of the vaccinated compartment is typically affected slightly by changes in the contact rate. It grows slowly over time, depending on the model

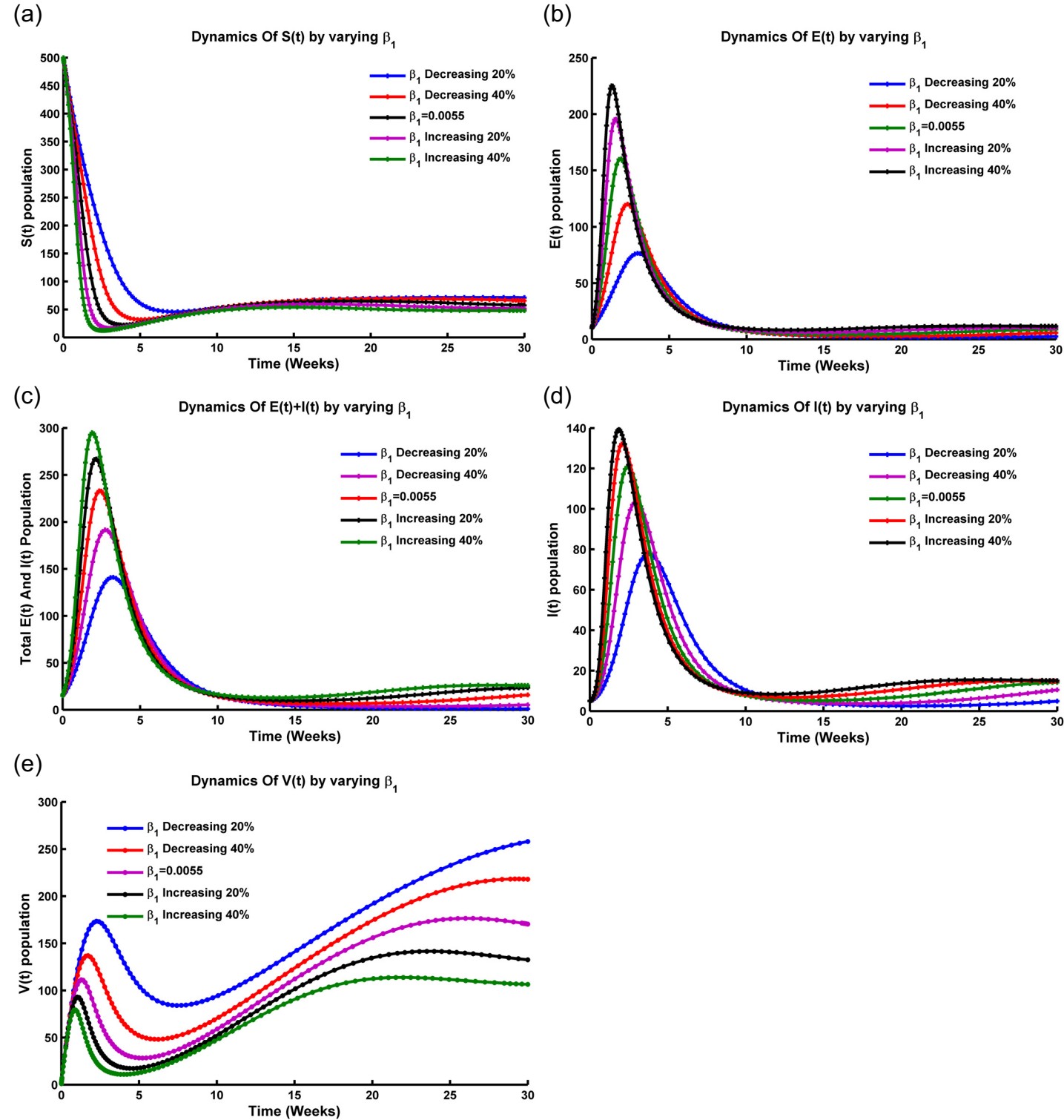

**Fig 7. Effect of $\beta_1$ in phase plane of (a) $S(t)$ compartment, (b) $E(t)$ compartment, (c) $E(t)$+ $I(t)$ compartment, (d) $I(t)$ compartment and (e) $V(t)$ compartment where $\beta_1$ varying in range [0.0035, 0.0065] and rest of parameters are taken from** Table 1.

assumptions and the rate of vaccination. Fig 7(e) illustrates that the vaccinated compartment acts as a buffer or barrier, reducing the number of susceptible individuals who can be infected due to increased contact rates. When contact rate $\beta_1$ and $\beta_2$ rises, vaccination class reduces slightly. The presence of a large and growing vaccinated compartment can lead to a decrease in the overall disease transmission within the population, as vaccinated individuals are less likely to contract and spread the infection.

When the treatment rate increases, that means infected individuals are progressing to the treated class under control from the infection at a faster rate. This has an impact on the trajectory of the infected compartment in the phase plane. Specifically, the trajectory tends to shift towards lower values along the y-axis, indicating a decrease in the number of infected individuals over time. Fig 8(a) indicates that a higher treatment rate results in a steeper slope of the trajectory, suggesting a more rapid decline in the number of infected individuals. That means the duration of the infectious period is shorter, leading to a faster resolution of the infection within the population.

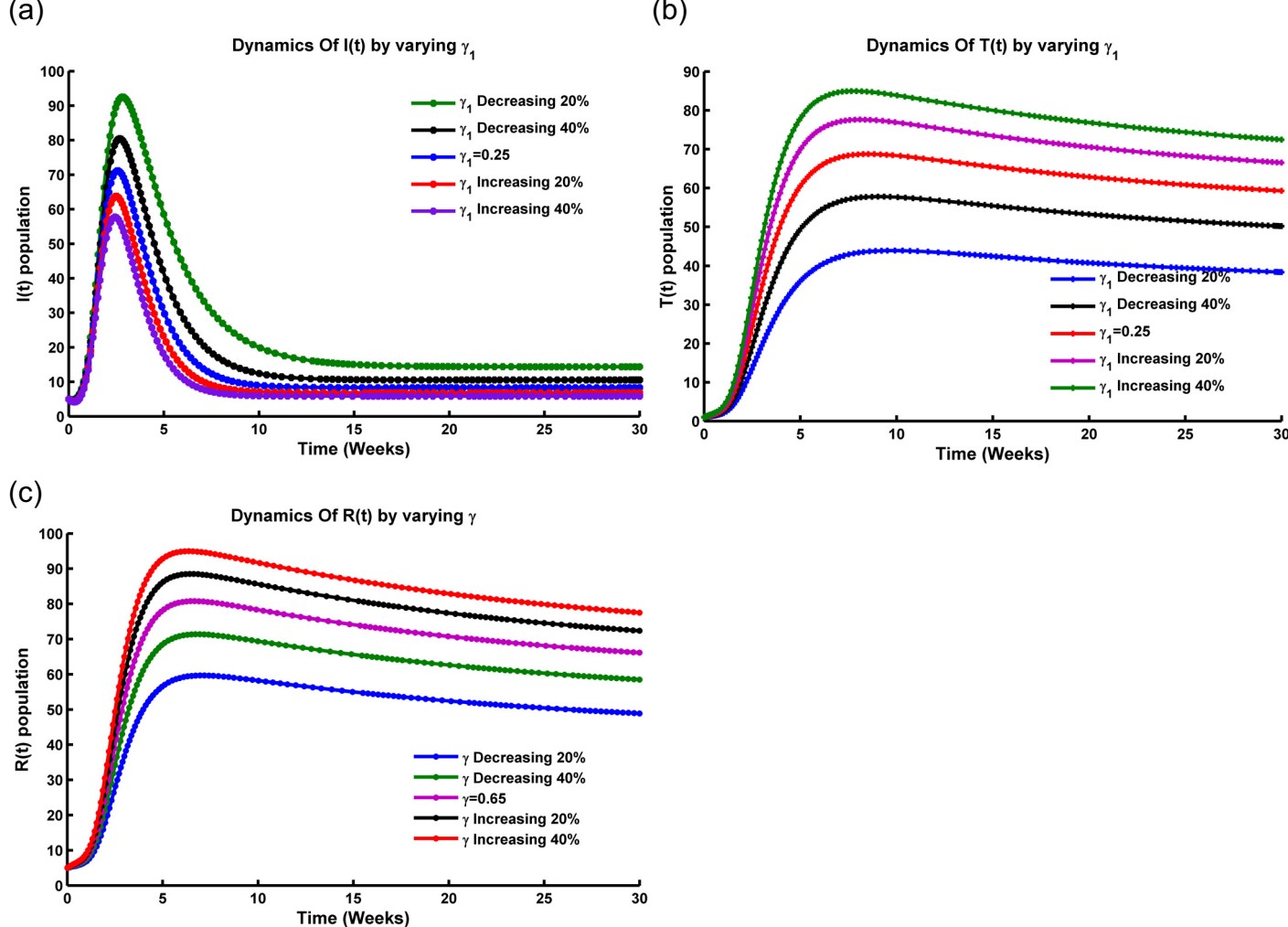

**Fig 8. Effect of $\gamma$ and $\gamma_1$ in the phase plane of (a) $I(t)$ compartment, (b) $T(t)$ compartment, (c) $R(t)$ compartment, where $\gamma_1$ varying in range [0.15, 0.45], $\gamma$ varying in range [0.35, 0.75] and rest of parameters are taken from Table 1.**

When the recovery rate increases, it conveys that infected individuals are recovering from the infection at a faster rate. This has an impact on the trajectory of the recovered compartment in the phase plane. Specifically, the trajectory tends to shift towards higher values along the y-axis, indicating an increase in the number of recovered individuals over time. Fig 8(b) demonstrates that when the treatment rate increases, individuals or entities in the treated compartment may experience faster recovery or improvement. This can be visualized in the phase plane analysis by observing trajectories that move more rapidly towards healthier states or regions associated with improved outcomes. Fig 8(c) reflects that a higher recovery rate results in a steeper slope of the trajectory, suggesting a more rapid increase in the number of recovered individuals. This implies that individuals are transitioning out of the infected compartment and into the recovered compartment more quickly. An increase in the treatment rate may result in a shift of the treated compartment towards a stable state. Therefore, the individuals or entities being treated experience positive changes and move towards a more desired condition or outcome.

## Phase plane analysis: Scenario on two compartments

The number or proportion of individuals in the exposed class influences the rate at which new infections occur. From Fig 9(a) we see that, as the exposed individuals become infectious, they transition into the infected class, contributing to the overall number of infected individuals. The interaction and flow of individuals between the exposed and infected classes determine the dynamics of the epidemic, such as the rate of transmission, the speed of disease spread, and the eventual size of the infected population. The more exposed the population grows, the faster grows of the infected population. This relationship is crucial for developing effective strategies to control and mitigate the impact of the epidemic. Fig 9(b) indicates that, as individuals recover, they transition from the infected class to the recovered class, reducing the number of active infections. The size and dynamics of the recovered class can impact the spread of the disease. A larger number of recovered individuals means a smaller pool of susceptible individuals for the disease to infect, potentially slowing down transmission rates. The rate at which individuals recover and transition to the recovered class affects the overall duration and severity of the epidemic. Faster recovery rates can lead to a quicker decline in the number of active infections. Fig 9(c) specifies that the size and effectiveness of the treatment class can impact the progression of the epidemic. Prompt and effective treatment can help reduce the duration of infectiousness and potentially lower the transmission rates. The interaction between the treatment and infected classes influences the overall burden of the disease on the healthcare system and the potential for reducing morbidity and mortality. The more the population progresses to the treated class from infected, it reduces the disease burden. Fig 9(d) indicates that vaccination reduces the susceptibility of individuals to infectious disease, thereby decreasing the likelihood of them transitioning from the susceptible class to the infected class. The higher the vaccination coverage within a population, the lower the number of individuals in the susceptible class, leading to a reduced pool of potential hosts for the disease. As the number of vaccinated individuals increases, the infected class may experience a decline in its size, resulting in a decline in the overall disease transmission rate. Thus, vaccination also plays a crucial role in reducing the transmission of the disease from infected individuals to susceptible ones, further limiting the spread of the epidemic.

Fig 10(a) reflects that initially, the infected class increases rapidly, presenting a high rate of new infections, while the cumulative infected class starts from zero. As time progresses, the infected class may reach a peak and start to decline, while the cumulative infected class continues to increase as new infections add to the total count. The shape and pattern of the trajectory

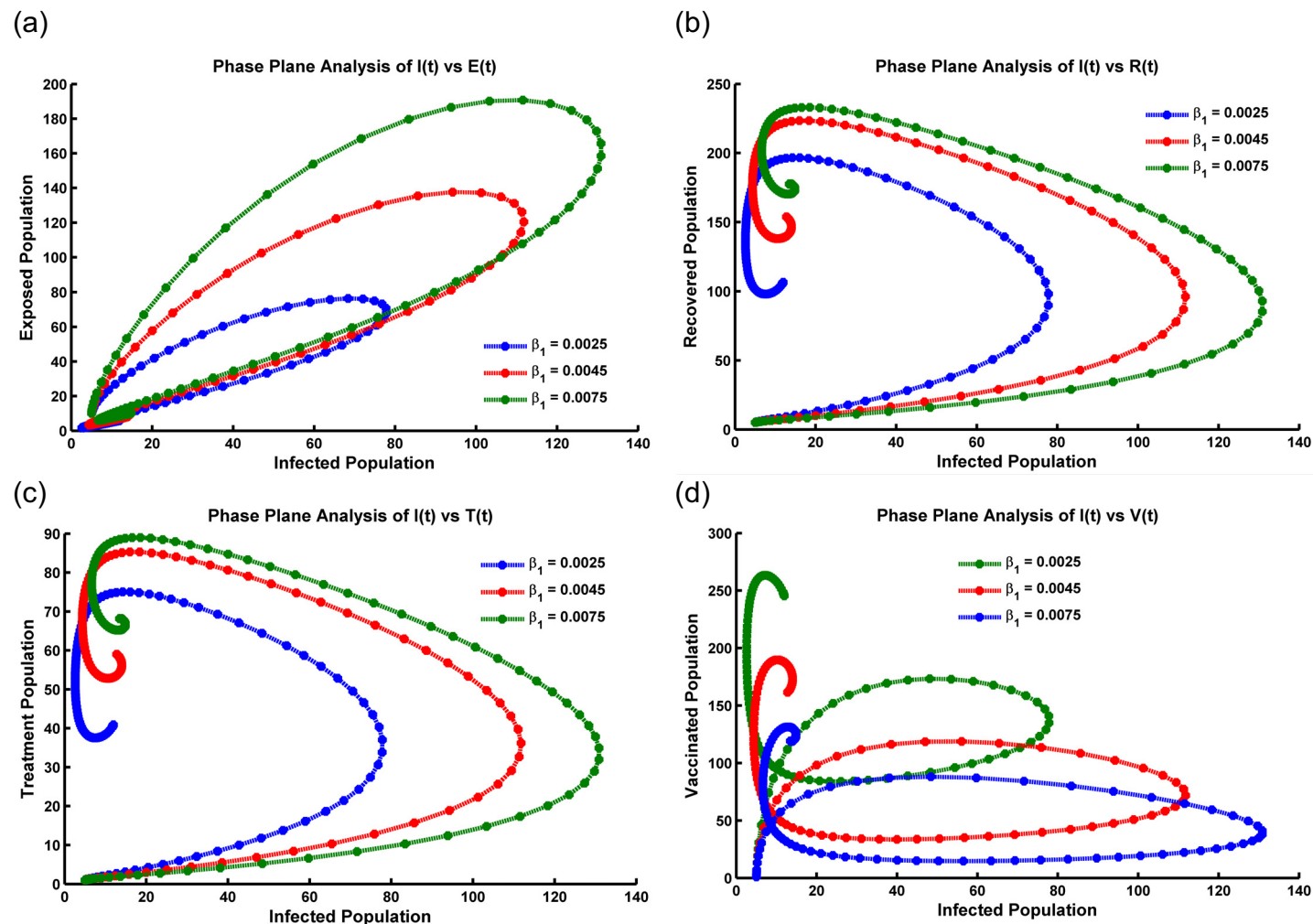

**Fig 9. Phase plane of (a) $I(t)$ vs $E(t)$ compartment, (b) $I(t)$ vs $R(t)$ compartment, (c) $I(t)$ vs $T(t)$ compartment and (d) $I(t)$ vs $V(t)$ compartment where all parameters are taken from Table 1.**

in the phase plane can reveal important information about the dynamics of the epidemic, such as the effectiveness of control measures or the presence of multiple waves. Fig 10(b) describes how the number of susceptible individuals changes over time as they become exposed to a particular infectious agent. The phase plane plot allows us to visualize the dynamics of this relationship by plotting the susceptible population on one axis and the exposed population on the other. Trajectories in the phase plane represent the flow of individuals between the susceptible and exposed states, providing insights into the progression of an infectious disease. The more susceptible individuals are reduced, it progresses to the exposed class, and the exposed population increases gradually. From Fig 10(c), these trajectories represent the movement of the system over time, considering the two variables: susceptible population and infected population. The direction and shape of the trajectories reveal the dynamics of the epidemic. Typically, when the susceptible population is high and the infected population is low, the trajectories move toward the susceptible axis. As the infected population increases, the more susceptible population reduces from the community, and the trajectories shift towards the infected axis,

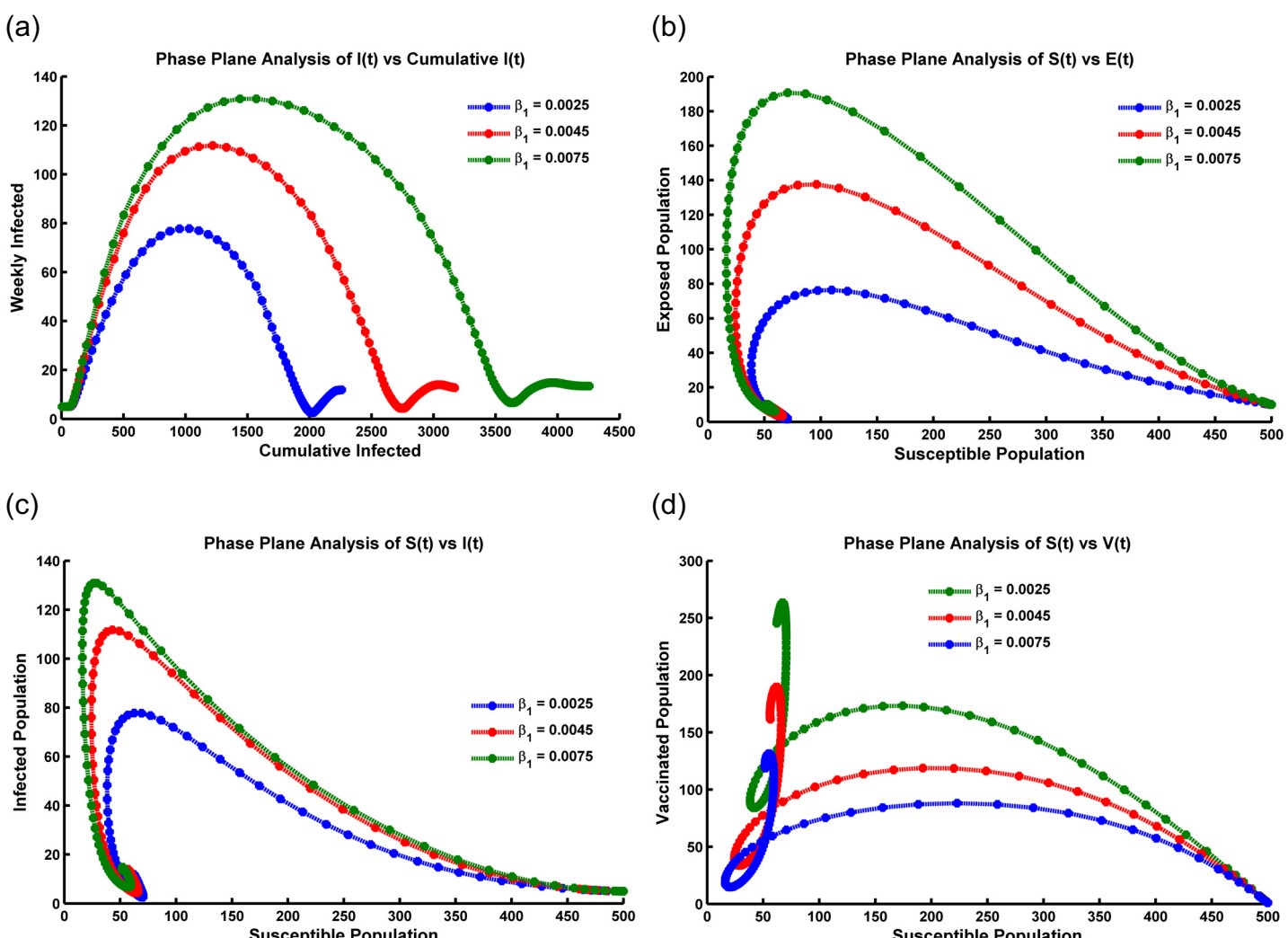

**Fig 10. Phase plane of (a)** *I(t)* vs Cumulative *I(t)* compartment **(b)** *S(t)* vs *E(t)* compartment **(c)** *S(t)* vs *I(t)* compartment and **(d)** *S(t)* vs *V(t)* compartment where all parameters are taken from Table 1.

indicating the spread of the disease. The intersection of the equilibrium point between the two axes represents the steady state, where the epidemic reaches a balance between susceptible and infected individuals. Fig 10(d) reflects that these trajectories demonstrate the dynamics of the system over time, considering the two variables: susceptible population and vaccinated population. When the susceptible population is high and the vaccinated population is low, the trajectories move toward the susceptible axis. As the vaccinated population increases with the decrease of the susceptible population, the trajectories shift towards the vaccinated axis, indicating the impact of vaccination on reducing susceptibility. That means, by vaccination, more portion of people progress to the control strategy. The intersection of the equilibrium point between the two axes represents the steady state, where a balance is achieved between susceptible and vaccinated individuals. The analysis helps assess the effectiveness of vaccination in mitigating the spread of the disease.

The vaccination parameter $\phi$ significantly impacts the basic reproduction number $\mathcal{R}_0$ and disease incidence in the model. As shown in the expression for $\mathcal{R}_0$:

$$\mathcal{R}_0 = \frac{\Lambda[\alpha\beta_2 + \beta_1(\gamma + \gamma_1 + \mu + \delta)]}{(\mu + \phi)(\alpha + \mu)(\gamma + \gamma_1 + \delta + \mu)},$$

an increase in $\phi$ raises the term $(\mu + \phi)$ in the denominator, which leads to a decrease in $\mathcal{R}_0$. This reduction indicates fewer secondary infections, resulting in lower disease incidence within the population. A higher $\phi$ reflects better vaccination coverage or efficacy, effectively diminishing the susceptible population $\tilde{S}$ and decreasing the likelihood of contact with infected individuals. Consequently, as the effective transmission potential decreases, the dynamics governing the spread of the disease are altered, making outbreaks less likely. Overall, enhancing the vaccination parameter $\phi$ is crucial for reducing $\mathcal{R}_0$ and controlling the disease's impact, highlighting the importance of vaccination strategies in public health initiatives. Thus, optimizing vaccination efforts can substantially mitigate the effects of infectious diseases on communities.

Based on phase plane analysis of the model, effective control can be achieved by manipulating key parameters. Increasing the vaccination rate ($\phi$) shifts individuals from the susceptible ($S$) to the vaccinated ($V$) class, reducing the number of potential infections. Enhancing vaccine efficacy ($\varepsilon$) ensures that vaccinated individuals are less likely to become infected, limiting the spread. Reducing transmission rates ($\beta_1, \beta_2$) through interventions like social distancing and hygiene decreases the rate at which susceptible and vaccinated individuals come into contact with exposed or infected individuals. Additionally, improving the treatment rate ($\gamma_1$) shortens the infectious period, lowering the disease burden in the population. Together, these adjustments stabilize the system, minimizing infection peaks and leading to more effective disease control.

## Contour plot analysis of $\mathcal{R}_0$

A contour plot of $\mathcal{R}_0$ (basic reproduction number) with respect to two parameters in epidemiology provides valuable insights into the spread and control of infectious diseases. By creating a contour plot, we can visualize how changes in two specific parameters affect the value of $\mathcal{R}_0$. This helps us in this thesis work to understand the dynamics of disease transmission and make informed decisions regarding interventions and control measures [31, 34]. The contour plot allows us to identify regions where $\mathcal{R}_0$ remains low or high based on the parameter values. It helps identify critical thresholds or tipping points that may lead to an outbreak or epidemic. For example, if the contour plot shows a steep increase in $\mathcal{R}_0$ as parameter values cross a certain threshold, it suggests that particular factors significantly influence disease transmission and warrant attention. Furthermore, the contour plot can guide decision-making by identifying areas where interventions or modifications to the parameters could effectively reduce $\mathcal{R}_0$. By manipulating the parameters within favorable regions, public health measures can be tailored to control or prevent the spread of infectious diseases more efficiently [32]. Moreover, contour plot serves as a powerful tool to understand the relationship between key factors, assess the potential for disease spread, and devise targeted strategies for disease control and prevention.

Additionally, The color bar in a contour plot indicates the values associated with different colors in the plot. It provides a visual representation of the magnitude or level of the variable being displayed. This typically represents the values of $\mathcal{R}_0$. Each color on the color bar corresponds to a specific range or interval of $\mathcal{R}_0$ values. The color intensity or shading within the contour plot indicates the relative magnitude of $\mathcal{R}_0$ at different points on the plot.

In this section, we have presented the relationship of the parameters by a contour plot of the basic reproduction number $\mathcal{R}_0$ as a function of parameters.

Fig 11(a) depicts the simulation of the model (1) by showing the contour plot of the threshold quantity $\mathcal{R}_0$ as a function of disease transmission rate $\beta_1$ from $E(t)$ compartment, and recovery rate $\gamma$. In general it shows that, $\mathcal{R}_0$ value decreases when $\beta_1$ increases, and $\mathcal{R}_0$ value increases when $\beta_1$ decreases. Also, $\mathcal{R}_0$ value decreases significantly, as $\gamma$ increases. But the rate of progression of $\mathcal{R}_0$ is faster than that of $\gamma$. Simulation of the model (1) is presented in Fig 11(b) showing the contour plots of $\mathcal{R}_0$ as a function of $\beta_1$ and the treatment rate $\gamma_1$. It indicates that the value of $\mathcal{R}_0$ increases (or decreases) gradually when the value of $\gamma_1$ increases (or decreases). That means if we can take a treatment strategy, we can mitigate the disease burden. Thus, one prevention strategy can be vaccination and proper isolation of the infected individuals.

The numerical simulation of the model presented in Fig 11(c) shows a contour plot of $\mathcal{R}_0$ as a function of parameter $\beta_1$ and infection rate $\alpha$. It shows that $\mathcal{R}_0$ increases (or decreases) rapidly when $\alpha$ increases (or decreases). Thus by vaccination and treatment strategies, if progression to the infected compartment can be reduced, the disease burden can be minimized. Fig 11(d) showing contour plot of $\mathcal{R}_0$ as a function of parameter $\beta_1$ and disease-induced death rate $\delta$ and natural death rate $\mu$. That indicates that $\mathcal{R}_0$ has a reverse relation with the progression rate of $\delta$ and $\mu$. Thus, the separation of the infected and exposed individuals has a significant impact on the extinction of the disease from the community. Further, Fig 11(e) presents a contour plot of $\mathcal{R}_0$ as a function of parameter $\beta_1$ and parameter $\beta_2$. In general, it shows that the threshold quantity increases (or decreases) quickly if $\beta_1$ and $\beta_2$ increases (or decreases). But the increasing (or decreasing) rate of $\mathcal{R}_0$ with respect to $\beta_2$ is faster than that of $\beta_1$. Thus, $\beta_1$ and $\beta_2$ are the crucial parameters that have a significant impact on the persistence of disease in the community. The epidemiological meaning of the simulation is that the increasing rate of $\gamma$ and $\gamma_1$ and decreasing rate of $\beta_1$, $\beta_2$ and $\alpha$ can reduce the spread of disease and mitigate the burden from the community.

## Box plot analysis

A box plot is a type of graph used to visualize the distribution of a dataset, particularly its median, quartiles, and outliers. In the context of basic reproduction number, a box plot analysis can be used to understand the variability of the basic reproduction number ($\mathcal{R}_0$) across different groups or periods. To analyze the threshold $\mathcal{R}_0$ using box plots, the median represents the typical value of $\mathcal{R}_0$, while the box indicates the interquartile range (IQR) of $\mathcal{R}_0$ values. The whiskers of the box plot represent the range of $\mathcal{R}_0$ values, while any points beyond the whiskers are considered outliers. In this section we have carried out the box plot analysis of $\mathcal{R}_0$ as a function of two parameters [31, 32, 34].

Fig 12(a) reflects valuable insights into the relationship between parameters $\beta_1$,$\beta_2$ (contact rate) and $\gamma$ (recovery rate) with the threshold quantity $\mathcal{R}_0$. The box visually displays the distribution of $\mathcal{R}_0$ at different combinations of $\beta_1$,$\beta_2$ and $\gamma$. We have observed that, the tendency of $\mathcal{R}_0$ is decreasing (or increasing) for the value of parameter $\gamma$ increasing (or decreasing). But progression of $\mathcal{R}_0$ is proportional with $\beta_1$ and $\beta_2$. From the whiskers extending from the box, we have observed the interquartile range of $\mathcal{R}_0$ is 5 to 7 when the median value is 6 (for increasing 30% of $\gamma$). Meanwhile, by increasing $\gamma$ by 60% and 90%, the median value of $\mathcal{R}_0$ decreases to 5 and 4.5 respectively; also the interquartile range. Fig 12(b) represents the influence of contact rates $\beta_1$, $\beta_2$ and treatment rate $\gamma_1$ on basic reproduction number $\mathcal{R}_0$. It reflects that increasing the rate of $\gamma_1$ reduces the median value of $\mathcal{R}_0$. When $\gamma_1$ increases 60%, the

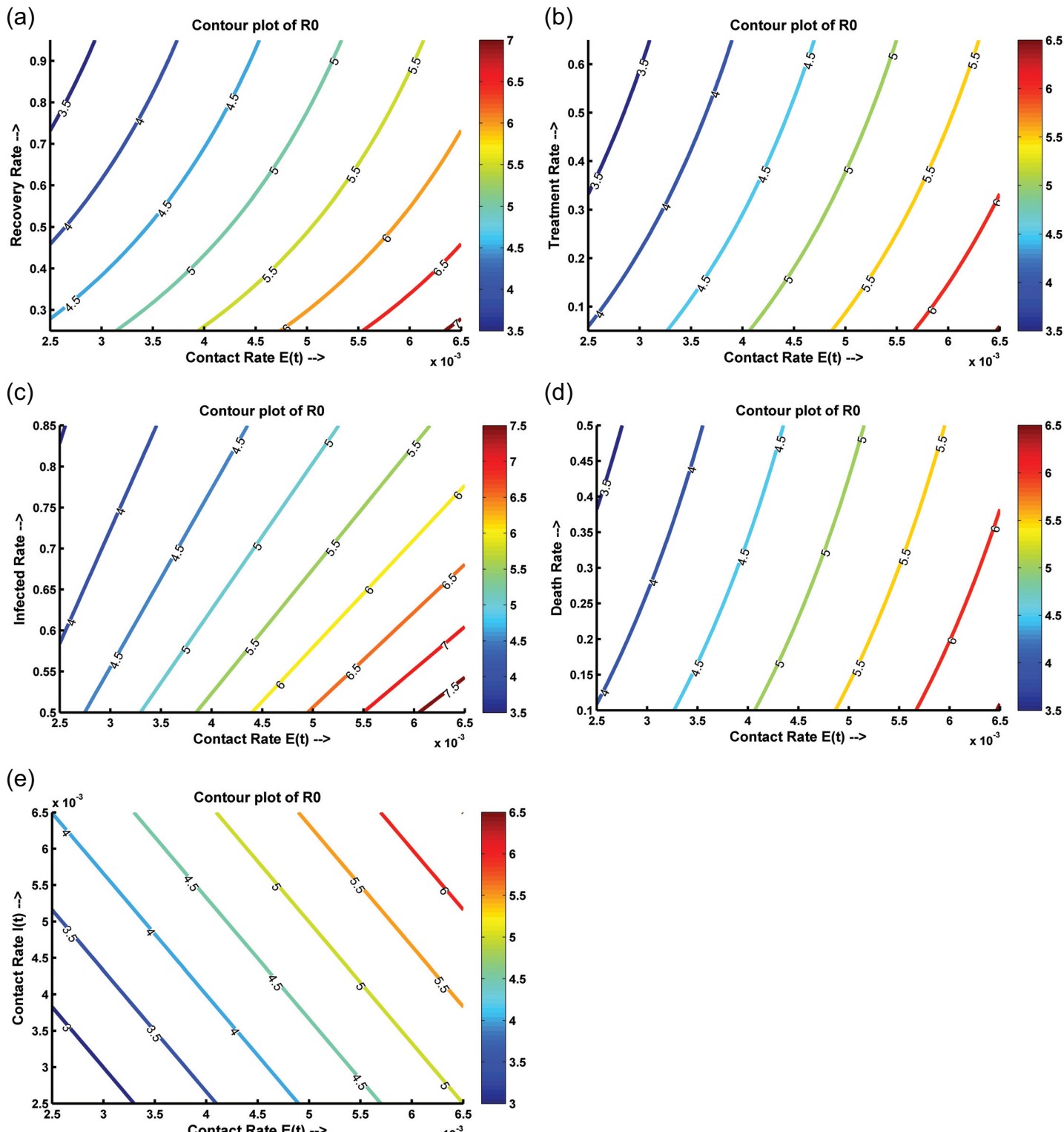

**Fig 11. Contour plots of $\mathcal{R}_0$ as a function of (a) parameter $\beta_1$ vs $\gamma$ (b) parameter $\beta_1$ vs $\gamma_1$ (c) parameter $\beta_1$ vs $\alpha$ (d) parameter $\beta_1$ vs $\delta$ and (e) parameter $\beta_1$ vs $\beta_2$, parameter values are taken from Table 1.**

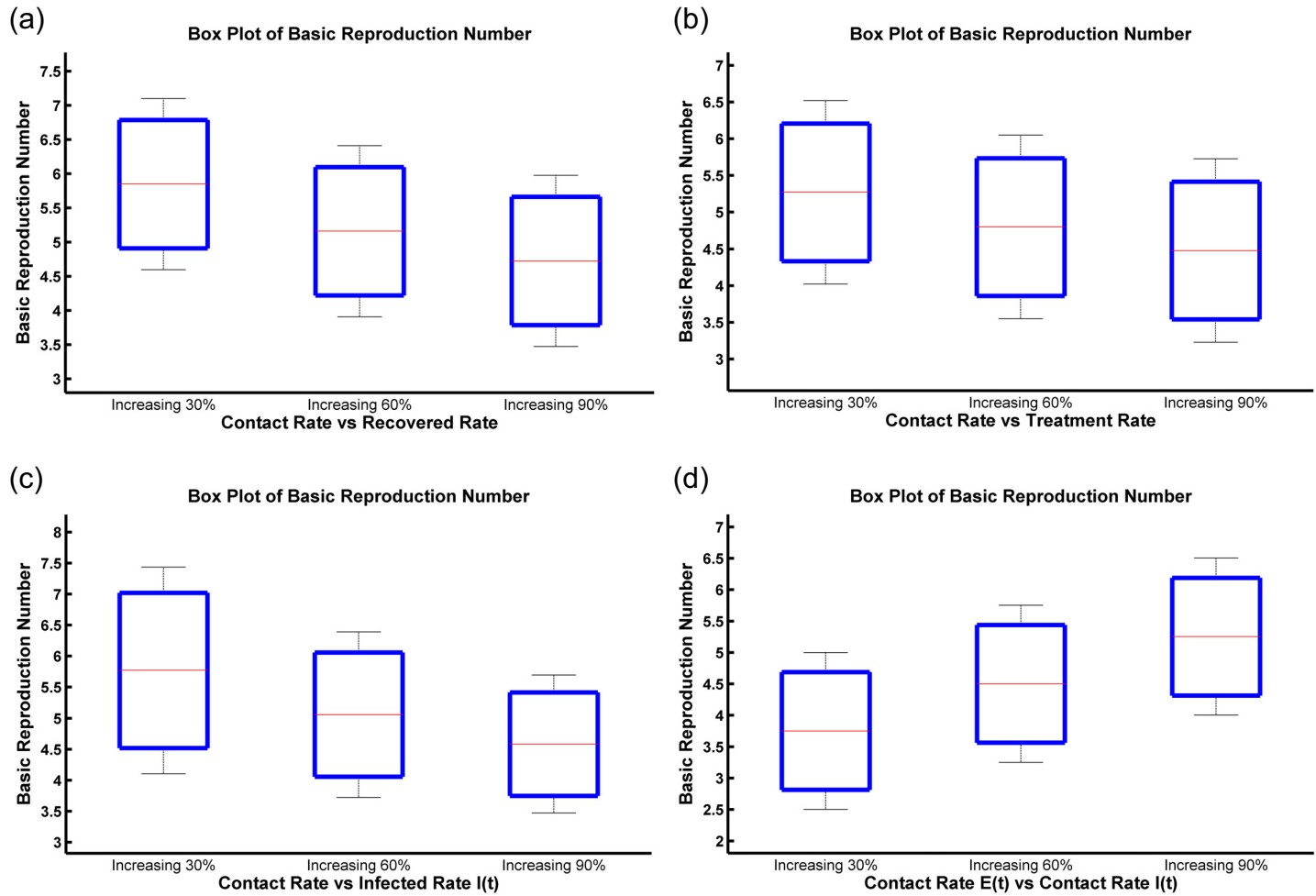

**Fig 12. Box plot analysis of $\mathcal{R}_0$ with the (a) parameter $\beta_1$ vs $\gamma$ (b) $\beta_1$ vs $\gamma_1$ (c) $\beta$ vs $\alpha$ and (d) $\beta_1$ vs $\beta_2$, where all the parameter values are taken from Table 1.**

median value of $\mathcal{R}_0$ is 4.7. From the whiskers extending from the box, we have observed inter-quartile range of $\mathcal{R}_0$ is 3.75 to 5.8.

Box plot analysis of $\mathcal{R}_0$ as a function of parameters $\beta_2$ and infection rate $\alpha$ is presented in Fig 12(c). This indicates that, with the progression of $\alpha$ and $\beta_2$, a positive impact on the inter-quartile range of $\mathcal{R}_0$. When $\alpha$ and $\beta_2$ both increase 60%, the median of $\mathcal{R}_0$ is 5, along with the interquartile range 4 to 6. But with the increasing of $\alpha$ and $\beta_2$ both 30%, the interquartile range of $\mathcal{R}_0$ increases in range [4.5, 7] and median is 5.75. Moreover, Fig 12(d) reveals box plot of $\mathcal{R}_0$ as a function of parameter $\beta_1$ and $\beta_2$. This indicates $\beta_1$ and $\beta_2$ have a strong positive influence on the growth of $\mathcal{R}_0$. When both these contact rates increase 30%, then the median value of $\mathcal{R}_0$ is 3.75, while the interquartile range is [2.75, 4.6]. After the progression of $\beta_1$ and $\beta_2$ at 60%, the median value of $\mathcal{R}_0$ increases to 4.5, and whiskers reflects the increasing of interquartile range in [3.5, 5.5]. After that, when $\beta_1$ and $\beta_2$ increases up-to 90%, the median of $\mathcal{R}_0$ is 5.2 and the interquartile range fall in [4.32, 6]. This analysis suggests that disease can more likely to spread rapidly with the increasing amount to transmission rates $\beta_1, \beta_2$, and infection rate $\alpha$.

Medicine for the Influenza virus is available in several countries. Also, experimental vaccine trials are going through in Africa's sun continent regions [1, 2]. The objective is to determine

the impact of treatment rate $\gamma_1$ on the threshold quantity $\mathcal{R}_0$. By taking the parameter's value from Table 1, box plot analysis shows that the treatment to exposed and infected individuals reduces the quantity $\mathcal{R}_0$ effectively. If individuals in the sub-acute phase are considered infectious, Fig 12 demonstrates that the treatment and vaccination control strategy significantly reduces the disease burden.

## Summary of graphical analysis

The number of infected individuals declines more quickly with treatment. When $\mathcal{R}_0 < 1$, the exposed and infected populations eventually converge to zero, as confirmed by numerical analysis at the Disease-Free Equilibrium (DFE). Conversely, when $\mathcal{R}_0 > 1$, the exposed and infected populations stabilize parallel to the susceptible population and do not approach zero, supporting the stability of the Endemic Equilibrium (EE) point numerically. The phase plane and contour plot analysis suggest that increasing treatment rates can stabilize the treated population, leading to positive outcomes for individuals. Effective treatment strategies, such as vaccination and isolating infected individuals, are crucial in reducing disease burden within communities. Box plot analyses highlight that treating exposed and infected individuals effectively lowers disease transmission rates, particularly when combined with vaccination efforts. Thus, implementing robust treatment and vaccination policies can swiftly diminish the number of infected individuals.

## A case study of influenza in Mexico

To study the current trend with our suggested model, in this section, we have examined recent data on influenza infection cases in Mexico from 1 October 2020 to 31 March 2023. The CDC and WHO websites were used to acquire all of the data [1, 2]. We have estimated illnesses for the near future and evaluated the applicability of the model using a total of 120 weekly data points. The initial population is taken as $S(0) = 500$, $V(0) = 1$, $E(0) = 1$, $I(0) = 1$, $R(0) = 1$ and $T(0) = 0$. The model outcomes predicted with real data are presented in Fig 13, where the weekly reported cases are depicted. The linear regression analysis with parameters in Table 2, implies that the model fits the weekly infected cases with excellent agreement. Thus, we demonstrate approximately 76% accuracy in tracking the original confirmed cases. Mexico is one of the important states located in North America, bordered by the United States. According to the report of CDC, in Mexico, there were over 7.5 thousand cases of influenza recorded in the 2018–2019 season. Only three instances had been reported for the 2020–2021 season as of March 13, 2021. The season of 2015–2016 had the highest number of influenza cases in the timeframe shown, with close to 10,000 cases officially reported. The worst-hit nation, Mexico, has reported 176 deaths from the novel H1N1 influenza strain. There have been cases documented all around the world, primarily among Mexican tourists, but they have been minor, and individuals have recovered with the right care and vaccinations. At the beginning of 2021 in Mexico reported cases of Influenza were 5.2% and 1.2% reported deaths. The government decided to implement a vaccine program from the outset, and it is still in place today at clinics, hospitals, community immunization centers, and pharmacies. The vaccinations worked after the prescribed dosage was given. Additional vaccines that have been authorized include Fluarix, FluLaval, Fluzone, and Afluria quadrivalent. These vaccinations are safe to provide to infants as young as six months. These vaccines have an efficacy of roughly 66%, 64%, 67%, and 62%, respectively. Fig 13 displays the correlation between the model's forecast and the real data. The results demonstrate the applicability of the proposed model.

Analyzing Fig 13(a), we see that seasonal flu cases are sporadically recorded each week in Mexico and range from 0 to 20 weeks. At week 20, there are 22 reported cases. Following that,

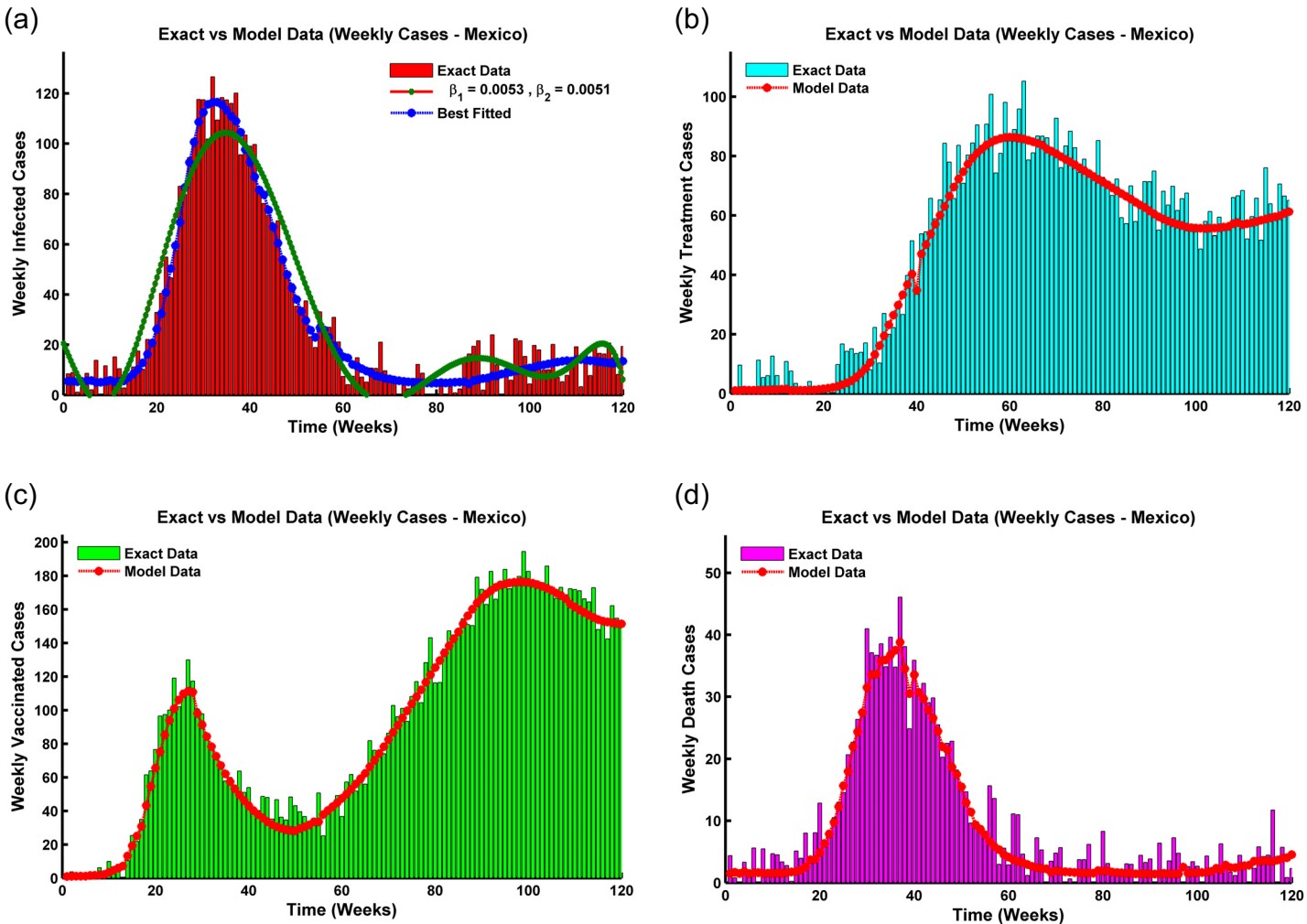

**Fig 13. Model data fitting to Mexico with (a) weekly infected data (b) weekly progression of treatment to control disease burden (c) weekly vaccination program to control disease burden (d) weekly death rates, where the fitted parameter sets are: $\beta_1 = 0.0055$, $\beta_2 = 0.0055$, $\alpha = 0.75$, $\gamma = 0.65$, $\gamma_1 = 0.25$, $\lambda = 0.55$, $\mu = 0.05$, $\delta = 0.3$.**

**Table 2. Estimated parameter values for model (1) by analyzing Mexico, and Colombia data.**

| Parameter | Mexico | Colombia | Sources |
|---|---|---|---|
| | Values | Values | |
| $\alpha$ | 0.75 | 0.77 | Estimated |
| $\beta_1$ | 0.00535 | 0.0063 | Estimated |
| $\beta_2$ | 0.00552 | 0.0065 | Estimated |
| $\gamma$ | 0.65 | 0.55 | Estimated |
| $\gamma_1$ | 0.25 | 0.33 | Estimated |
| $\lambda$ | 0.55 | 0.53 | Estimated |
| $\mu$ | 0.05 | 0.04 | Estimated |
| $\delta$ | 0.3 | 0.32 | Estimated |
| $\varepsilon$ | 0.45 | 0.42 | Estimated |

the infection grew gradually from week 21 to week 40. The highest number of infected cases (120 per 1000) were reported at weeks 33 and 37. Along weeks 30 to 50, disease conditions are not satisfactory. Due to appropriate care and isolation, the illness epidemic marginally decreased after 50 weeks. After 80 weeks, the reported instances (per thousand) vary from 20 to 40.

In Fig 13(b), the weekly treatment cases are shown. This suggests that the more contaminated cases there are, the stricter the treatment guidelines get. The reported number of treatment cases between weeks 40 and 80 ranges from 80 to 100. 80 weeks later, this rate changed with time and now varies between 40 and 75 (per thousand).

In Fig 13(c), our suggested model predicts the ongoing vaccination cases. In 20 to 35 weeks, the vaccination rate is high based on our model. A preventive policy was used on 100 populations on average (per 1000). Then, as the number of people with the infection increased, so did the number of instances of vaccination. Between weeks 60 and 120, there have been between 120 and 180 documented vaccination cases, which is important for containing the outbreak.

On the other hand, Fig 13(d) describes the recent death cases in Mexico. According to data analysis, from October 1, 2021, to weeks 30 to 50, the majority of fatality instances took place. The death rate ranges from 0 to 10 (per 1000) throughout weeks 0 to 20. After 60 weeks, the death cases became stable, fluctuating between 0 and 10 cases (per 1000).

## A case study of influenza in Colombia

In this section, we've examined current data for influenza infection cases in Colombia from 1 October 2020 to 31 March 2023, to study the recent trend using our suggested model. The websites of the CDC and WHO were used to acquire all the data [1, 2]. To assess the model's applicability and estimate disease for the near future, we took into account a total of 120 weekly data points. The starting population is regarded as $S(0) = 510$, $V(0) = 1$, $E(0) = 1$, $I(0) = 1$, $R(0) = 1$ and $T(0) = 0$. The model outcomes predicted with real data are presented in Fig 14, where the weekly reported cases are depicted. The linear regression analysis with parameters in Table 2, suggests that there is remarkable agreement between the model and the weekly infected cases. Thus, we demonstrate approximately 72% accuracy in tracking the original confirmed cases. Colombia is located in the northwest corner of South America which experiences a variety of climatic conditions due to its diverse topography and geographical location. Tropical Climate, wet and dry seasons, high rainfall, average temperature variation averages ranging from 24˚C to 28˚C and humidity. According to the report of WHO: 10,773 fatalities in Colombia were due to influenza, accounting for 5.24% of all deaths. Colombia, with an age-adjusted death rate of 19.35 per 100,000 people, is ranked 119th in the world.

In Colombia, there were 5.5% reported cases of influenza at the start of 2021, and 2.2% reported deaths. The government made the initial decision to carry out the immunization campaign, which is still being done today in clinics, hospitals, community vaccination locations, and drug stores. The immunizations were efficacious after being taken in their whole. Furthermore, the vaccinations Fluarix, FluLaval, Fluzone, and Afluria quadrivalent have received approval. You may vaccinate children as early as six months old. These vaccines have respective efficacy rates of about 66%, 64%, 67%, and 62%. The model prediction that correlates with the actual data is shown in Fig 14(a)–14(d) of the reference case study for Colombia. The findings show that the suggested paradigm is useful.

Analyzing total cases of population per 1000 for Colombia, from Fig 14(a) we see that there are a modest number of seasonal flu cases reported each week in Colombia, ranging from 0 to 27 weeks. There are 19 reported cases in week 27. From week 20 to week 38, the infection then grew progressively. The highest number of infected cases were found at weeks 33 and 340 (142

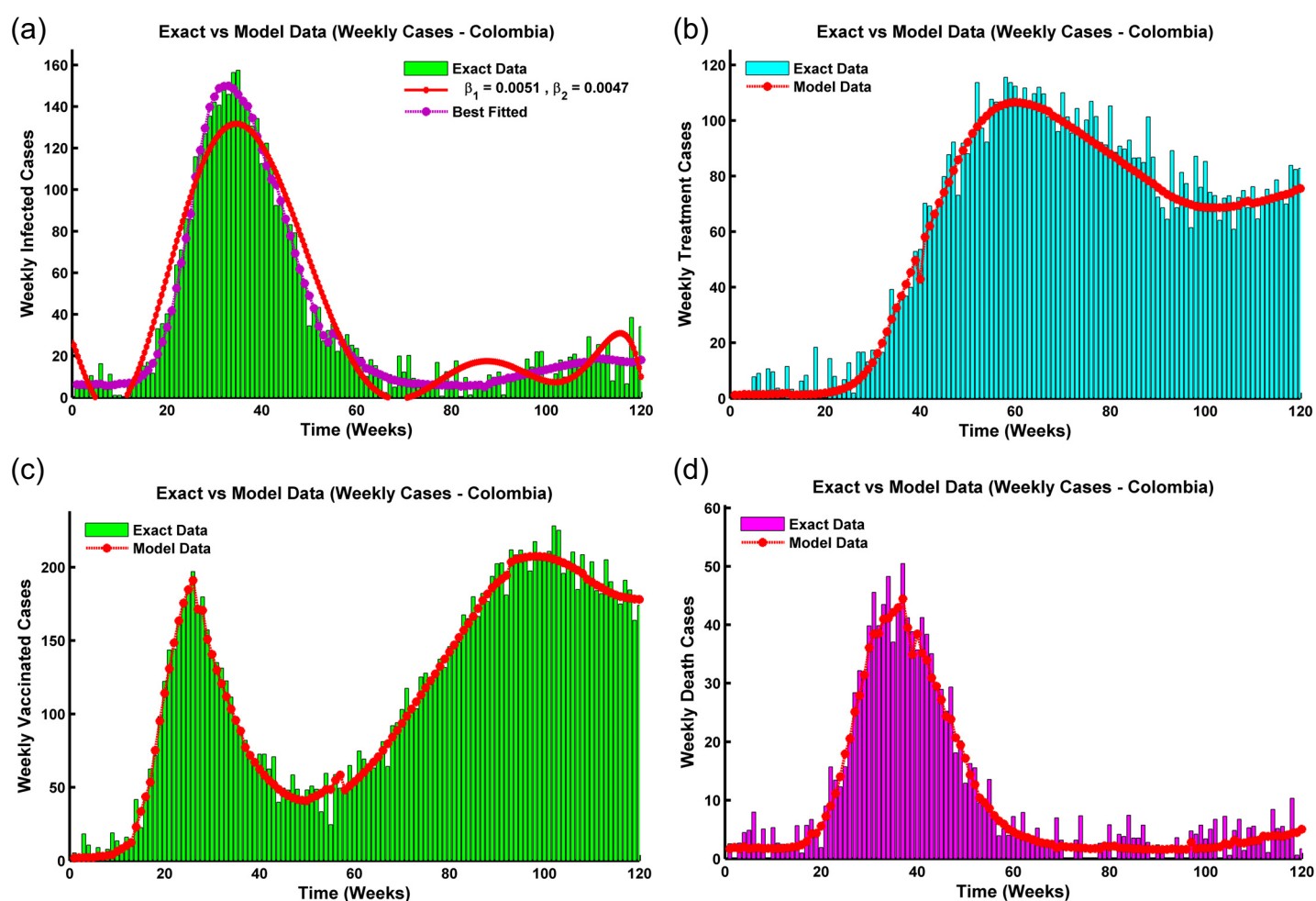

**Fig 14. Model data fitting to Colombia with (a) weekly infected data (b) weekly progression of treatment to control disease burden (c) weekly vaccination program to control disease burden (d) weekly death rates, where the fitted parameter sets are:** $\beta_1 = 0.0065$, $\beta_2 = 0.0063$, $\alpha = 0.77$, $\gamma = 0.55$, $\gamma_1 = 0.33$, $\lambda = 0.53$, $\mu = 0.04$, $\delta = 0.32$.

per 1000). Along weeks 30 to 45, the disease is in a serious state. Due to appropriate care and isolation after 45 weeks, the disease outbreak marginally lessened. The reported cases range from 18 to 35 (per thousand) after 75 weeks.

In Fig 14(b), the weekly treatment cases are shown. This presents that as more cases of infection appear, treatment guidelines are developed. From weeks 35 to 85, between 85 and 100 treatment instances are documented. Following a few weeks, this rate changes over time and oscillates between 60 and 75 (per thousand).

In Fig 14(c), with our suggested approach, the ongoing vaccination cases are projected. Our model predicts that the immunization rate will peak between 20 and 35 weeks. In Colombia, preventive measures were implemented on average on 150 populations (per 1000). Then, the number of vaccination cases increased in tandem with the number of infected patients. The range of documented vaccination cases between weeks 60 and 120 is 150–200, which is important in terms of controlling the epidemic.

On the other hand, Fig 14(d) describes the recent death cases in Colombia. According to data analysis, from October 1, 2021, through week 25–46, the majority of death cases take

place. The death rates range from 0 to 8 per 1000 people from week 0 to week 20. After 60 weeks, the death rate stabilizes and ranges from 0 to 10 instances (per 1000).

## Conclusion

This study explores a deterministic SVEIRT epidemic model for influenza, incorporating vaccination and treatment strategies. The model is proven to be well-posed, with the DFE being globally asymptotically stable when the basic reproduction number, $\mathcal{R}_0$, is less than one, as demonstrated using Lyapunov functions and the LaSalle Invariance Principle. When $\mathcal{R}_0$ exceeds one, a unique EE exists, whose local and global stability are established through the Routh-Hurwitz criterion and non-linear Lyapunov functions, respectively. Moreover, numerical simulations of the model support the existence and stability of DFE and EE. The simulations indicate that higher transmission rates ($\beta_1, \beta_2$) and infection acquisition rates ($\alpha$) lead to a rapid increase in disease burden. We have observed that if the rate of acquiring infection increases, then the disease burden will increase. Also, we have observed the growth rate in the $E(t)$ compartment raises the disease burden. Meanwhile, treatment rate and vaccination rate can mitigate the factor $\mathcal{R}_0$ effectively. Furthermore, the contact rates $\beta_1$ and $\beta_2$ resulted in the rapid increase of the disease in the community. Also, the treatment of infected individuals in the acute phase is significant. Human recruitment rate can provide a great influence on controlling the disease. According to the report of CDC [1], flu vaccination reduces the risk of flu illness by between 40% and 60% among the overall population during seasons when most circulating flu viruses are well-matched to those used to make flu vaccines. Several different brands of standard dose flu shots are available, including Afluria Quadrivalent, Fluarix Quadrivalent, FluLaval Quadrivalent, and Fluzone Quadrivalent as an active ingredient [2]. Vaccination programs can reduce the quantity $(1 - \varepsilon)$ which can reduce the rate $\mathcal{R}_0$. Quarantine policy, using masks, hand sanitizer, and droplets in a proper way is important to increase the recovery and reduction of interaction rate. Further, we have analyzed recent trends in Mexico, and Colombia data in which analytical, numerical, and statistical studies were implemented, respectively. We successfully depicted the respective disease incidence by showing numerical results with proper fittings. We have validated our framework by comparing its predictions with simulation results. Ultimately, vaccinations, treatment plans, maintaining a preventative lifestyle, and ensuring that everyone has access to a sufficiently supported medical care system may all significantly minimize the likelihood of an illness epidemic. Thus, our desired model assures the effectiveness of the strategy to reduce and gradually prevent disease outbreaks.

## Appendix

### Appendix 1

**Demonstrations of the analytical findings**. This section represents the statements and proofs for the analytical results.

**Existence of solution.** **Theorem 11**. *(Existence of solution) Let* $\{S_0, V_0, E_0, I_0, R_0, T_0\} \in \mathbb{R}^6$ *be presented. There subsists* $t_0 > 0$, *and continuously differentiable functions* $\{S, V, E, I, R, T : [0, t_0) \to \mathbb{R}\}$ *such that the ordered pairs of states* $(S, V, E, I, R, T)$ *satisfies* (1) *and* $(S, V, E, I, R, T)(0) = (S_0, V_0, E_0, I_0, R_0, T_0)$.

*Proof*. The Picard-Lindelof Theorem asserts that concerning the initial value issue, $y'(t) = f(y(t))$, $y(t_0) = y_0$, $t \in [t_0 - \epsilon, t_0 + \epsilon]$, if $f$ is continuous in $t$ and locally Lipschitz in $y$, then for some value $\epsilon > 0$, a unique solution $y(t)$ exists to the initial value problem within the range $[t_0 - \epsilon, t_0 + \epsilon]$. Because the system of ODEs is autonomous, it suffices to demonstrate that the

function $\mathbf{f} : \mathbb{R}^6 \to \mathbb{R}^6$ determined by,

$$\mathbf{f}(\mathbf{y}) = \begin{pmatrix} \Lambda - (\beta_1 E + \beta_2 I)S - (\mu + \phi)S \\ \phi S - (1 - \varepsilon)(\beta_1 E + \beta_2 I)V - \mu V \\ (\beta_1 E + \beta_2 I)S - (\alpha + \mu)E \\ \alpha E + (1 - \varepsilon)(\beta_1 E + \beta_2 I)V - (\mu + \delta + \gamma + \gamma_1)I \\ \gamma I - \mu R \\ \gamma_1 I - \mu T \end{pmatrix}$$

is locally Lipschitz in its $y$ argument. The Jacobian matrix,

$$\nabla \mathbf{f}(\mathbf{y}) = \begin{pmatrix} a_{11} & 0 & -\beta_1 S & -\beta_2 S & 0 & 0 \\ \phi & a_{22} & -\lambda\beta_1 V & -\lambda\beta_2 V & 0 & 0 \\ \beta_1 E + \beta_2 I & 0 & a_{33} & \beta_2 S & 0 & 0 \\ 0 & a_{42} & \alpha + \lambda\beta_1 V & a_{44} & 0 & 0 \\ 0 & 0 & 0 & \gamma & -\mu & 0 \\ 0 & 0 & 0 & \gamma_1 & 0 & -\mu \end{pmatrix}$$

is linear in $\mathbf{y} \in \mathbb{R}^6$. Where
$a_{11} = -(\beta_1 E + \beta_2 I) - (\mu + \phi)$, $a_{22} = -\lambda(\beta_1 E + \beta_2 I) - \mu$,
$a_{33} = \beta_1 S - (\alpha + \mu)$, $a_{42} = \lambda(\beta_1 E + \beta_2 I)$, and $a_{44} = \lambda\beta_2 V - (\mu + \delta + \gamma + \gamma_1)$.

So, on a closed interval, $\nabla\mathbf{f}(\mathbf{y})$ is continuous while differentiable on an open interval $I_1 \in \mathbb{R}^6$. According to the Mean Value Theorem,

$$\frac{|\mathbf{f}(\mathbf{y_1}) - \mathbf{f}(\mathbf{y_2})|}{|\mathbf{y_1} - \mathbf{y_2}|} \leq |\nabla\mathbf{f}(\mathbf{y}^*)|$$

for $\mathbf{y}^* \in I_1$. Let $|\nabla\mathbf{f}(\mathbf{y}^*)| = K$, we get $|\mathbf{f}(\mathbf{y_1}) - \mathbf{f}(\mathbf{y_2})| \leq K|\mathbf{y_1} - \mathbf{y_2}|$ for all $\mathbf{y_1}, \mathbf{y_2} \in I_1$, and therefore for every $\mathbf{y} \in \mathbb{R}^6$, $\mathbf{f}(\mathbf{y})$ is locally bounded. As a result, $\mathbf{f}$ is locally Lipschitz in $\mathbf{y}$ since it has a continuous, bounded derivative on any compact subset $\mathbb{R}^3$. The Pichard-Lindelof theorem states that for any time $t_0 > 0$, there is a unique solution, $y(t)$, to the ordinary differential equation $y'(t) = f(y(t))$ with starting value $y(0) = y_0$ on $[0, t_0]$.

**Positivity of solution.** The boundedness and positivity of the solutions are two main constituents of an epidemic model. To convey that any solution with positive beginning values stays positive for all times $t > 0$, it is necessary to establish that all parameters and variables are always positive for $t > 0$. Positive behavior is biologically interpreted as the long-term survival of the population [6, 7].

**Theorem 12**. *(Positivity of solution) Consider the initial conditions of the system* (1) *are* $S(0) \geq 0$, $V(0) \geq 0$, $E(0) \geq 0$, $I(0) \geq 0$, $R(0) \geq 0$, *and* $T(0) \geq 0$; *the solutions* $S(t)$, $V(t)$, $E(t)$, $I(t)$, $R(t)$, *and* $T(t)$ *are non negative* $\forall t > 0$.

*Proof.* Assume that

$$\hat{t} = \sup\{t > 0 : S(t) \geq 0, V(t) \geq 0, E(t) \geq 0, I(t) \geq 0, R(t) \geq 0, \text{and } T(t) \geq 0\} \in [0, t].$$

Since the solution is continuous and each initial condition is non-negative, there must be a period while the outcome is still positive, and we observe that $\tilde{t} > 0$. Then, each term is calculated on the interval $[0, t]$. Thus, $\hat{t} > 0$ and results from the equation of system (1) that,

$$\frac{dS}{dt} \geq \Lambda - (\lambda_1 + \mu)S. \quad [\text{where } \lambda_1 = (\beta_1 E + \beta_2 I)]$$

This inequality can be resolved by applying the integrating factor approach.

$$\frac{d}{dt}\left\{ S(t)\exp\left[\mu(t) + \int_0^t \lambda_1(s)dS\right]\right\} \geq \pi \exp\left[\mu t + \int_0^t \lambda_1(s)dS\right].$$

Integrating both sides yields,

$$S(\hat{t})\exp\left[\mu\hat{t} + \int_0^{\hat{t}} \lambda_1(s)dS\right] \geq \int_0^{\hat{t}} \pi \exp\left[\mu\hat{t} + \int_0^{\hat{t}} (\lambda_1(w))dw\right]d\hat{t} + C.$$

Where C is the integration constant depending on the upper limit of $\lambda_1$, $\mu$, and $S(0)$. Hence,

$$S(\hat{t}) \geq S(0)\exp\left[-(\mu\hat{t} + \int_0^{\hat{t}} (\lambda_1(S)dS))\right] +$$

$$\exp\left[-(\mu\hat{t} + \int_0^{\hat{t}} (\lambda_1(S)dS))\right] \cdot \left(\int_0^{\hat{t}} \pi \exp\left[(\mu\hat{t}) + \int_0^{\hat{t}} (\lambda_1(w)dw))\right]d\hat{t}\right) > 0.$$

So, $S(\hat{t}) \geq 0, \ \forall \ \hat{t} \geq 0$.

Next, from the positivity of the solutions place bounds on other compartments [6]. Here,

$$\frac{dV}{dt} \geq -((1-\varepsilon)\lambda_1 + \mu)V$$

$$\Rightarrow V(\hat{t}) \geq V(0)\exp\left[-\{\mu\hat{t} + \int_0^{\hat{t}} (1-\varepsilon)\lambda_1(s)dS\}\right] > 0, \ \forall \ \hat{t} \geq 0.$$

It is examined that,

$$E(\hat{t}) \geq E(0)e^{-(\mu+\alpha)\hat{t}} > 0.$$
$$I(\hat{t}) \geq I(0)e^{-(\mu+\delta+\gamma+\gamma_1)\hat{t}} > 0.$$
$$R(\hat{t}) \geq R(0)e^{-\mu\hat{t}} > 0.$$
$$T(\hat{t}) \geq T(0)e^{-\mu\hat{t}} > 0.$$

for $\tilde{t} \in [0, t]$. As a result, an upper limit can be set for $S(t)$, $V(t)$, $E(t)$, $I(t)$, $R(t)$, and $T(t)$. Therefore, all the solutions of the system (1) will stay non-negative for $t \geq 0$, encompassing at time $\tilde{t}$. According to continuity, there must exist $t > \tilde{t}$ such that $S(t)$, $V(t)$, $E(t)$, $I(t)$, $R(t)$, and $T(t)$ are strictly positive on the entire interval $[0, t]$. Further extending the interval of existence is possible because all functions stay bounded on this same interval [7]. The bounds on the compartments $S$, $V$, $E$, $I$, $R$, and $T$ that were derived previously hold for any brief time (compact interval). Consequently, we can extend the existence of the solution to $[0, t]$ for every $t > 0$. Based on the aforementioned reasoning, the solutions continue to be positive and confined to $[0, t]$.

**Boundedness of solution.** Boundedness can be understood as a natural growth constraint resulting from scarce resources, while positivity suggests that every member of the compartment population survives [6, 15, 45, 46].

**Theorem 13**. [7]. *(Positive invariance and boundedness of solutions) The closed region* $\Omega = \{(S, V, E, I, R, T) \in \mathbb{R}_+^6 : 0 < N \leq \frac{\Lambda}{\mu}\}$ *is positively invariant and attracting set for the system* (1).

*Proof.* We can write in vector form,

$$X = (S, V, E, I, R, T)^T \in \mathbb{R}^6.$$

We define,

$$F(X) = \begin{pmatrix} F_1(X) \\ F_2(X) \\ F_3(X) \\ F_4(X) \\ F_5(X) \\ F_6(X) \end{pmatrix} = \begin{pmatrix} \Lambda - (\beta_1 E(t) + \beta_2 I(t))S(t) - (\mu + \phi)S(t) \\ \phi S(t) - (1 - \varepsilon)(\beta_1 E(t) + \beta_2 I(t))V(t) - \mu V(t) \\ (\beta_1 E(t) + \beta_2 I(t))S(t) - (\alpha + \mu)E(t) \\ \alpha E(t) + (1 - \varepsilon)(\beta_1 E(t) + \beta_2 I(t))V(t) - (\mu + \delta + \gamma + \gamma_1)I(t) \\ \gamma I(t) - \mu R(t) \\ \gamma_1 I(t) - \mu T(t) \end{pmatrix}$$

where $F : \mathbf{C}_+ \to \mathbb{R}^6$, and $F \in \mathbf{C}^\infty(\mathbb{R}^6)$.

Now,

$$\dot{X} = F(X_t),$$

where $\cdot \equiv \frac{d}{dt}$ including $X_t(\theta) = X(t + \theta)$, $\theta \in [0, \tau]$.

It is simple to verify that whenever we desire $X(\theta) \in \mathbf{C}_+$ such that $X_i = 0$, then we acquire $F_i(X)|_{Xi}(t)=0$, $X_t \in \mathbf{C}_+ \geq 0$, $i = 1, 2, \cdots, 6$. Any result to the model's equation, alongside $X_t(\theta) \in \mathbf{C}_+$ say, $X(t) = X(t, X(0))$ is such that $X(t) \in \mathbb{R}^6_{0+}$ for all $t > 0$. The size of the population, $N = S + V + E + I + R + T$ with the initial conditions, $S(0) \geq 0$, $V(0) \geq 0$, $E(0) \geq 0$, $I(0) \geq 0$, $R(0) \geq 0$, $T(0) \geq 0$.

Now, for the boundedness of the solution we define,

$$\frac{dN}{dt} =\leq \Lambda - \mu N.$$

This indicates that $N(t)$ is bounded, and so are the $S(t)$, $V(t)$, $E(t)$, $I(t)$, $R(t)$ and $T(t)$.

$$\text{Here,} \quad N \leq N_0 e^{-\mu t} + \frac{\Lambda}{\mu}(1 - e^{-\mu t}) \tag{4}$$

from this expression when $t \to \infty$, and $N(t) \leq \frac{\Lambda}{\mu}$. The system will be examined in biologically feasible regions as follows. As a result, we can consider the feasible region $\Omega = \{(S, V, E, I, R, T) \in \mathbb{R}^6_+ : S \leq \frac{\Lambda}{\mu}, V, E, I, R, T \geq 0\}$.

**Theorem 14**. [7]. *The feasible region $\Omega$ is determined by,*

$$\Omega = \left\{ (S(t), \ V(t), \ E(t), \ I(t), \ R(t), \ T(t)) \in \mathbb{R}^6_+ \ | \ 0 \leq N \leq \ \max\left\{ N(0), \frac{\Lambda}{\mu} \right\} \right\}.$$

*with initial conditions $S(t) > 0$, $V(t) > 0$, $E(t) > 0$, $I(t) > 0$, $R(t) > 0$, $T(t) > 0$, is positively invariant and attracting with respect to system* (1) $\forall t > 0$.

*Proof.* Here,

$$\frac{dN}{dt} = \Lambda - \mu N.$$

The omission of the influenza infection ensures that,

$$\frac{dN}{dt} \leq \Lambda - \mu N.$$

Now,

$$N(t) \leq \frac{\Lambda}{\mu} + \left( N(0) - \frac{\Lambda}{\mu} \right) \exp\left( -\mu t \right). \tag{24}$$

From (24), we examine that as $t \to \infty$, $N(t) \to \frac{\Lambda}{\mu}$. So, if $N(0) \leq \frac{\Lambda}{\mu}$ then $\lim_{t\to\infty} N(t) = \frac{\Lambda}{\mu}$. On the other hand, if $N(0) > \frac{\Lambda}{\mu}$, then total population $N$ will decrease to $\frac{\Lambda}{\mu}$ as $t \to \infty$. Particularly, $N(t) < \frac{\Lambda}{\mu}$ if $N(0) < \frac{\Lambda}{\mu}$. This means that $N(t) \leq \max\left\{ N(0), \frac{\Lambda}{\mu} \right\}$. Hence, no solution path crosses any borders of $\Omega$, making the region $\Omega$ a positively invariant set of the model (1). This demonstrates that the developed model is applicable from a mathematical and epidemiological perspective [6, 8]. The model is appraised in the biologically feasible region which means the considered model is well-defined. Further, $N(t) > \frac{\Lambda}{\mu}$, then the other solution enters $\Omega$ in finite time, or $N(t)$ approaches to $\frac{\Lambda}{\mu}$, and the variables $E(t)$, $I(t)$, $R(t)$, $V(t)$ and $T(t)$ approach to zero. Hence the region $\Omega$ is attracting.

Therefore, $N(t)$ is bounded above. Subsequently, $S(t)$, $V(t)$, $E(t)$, $I(t)$, $R(t)$ and $T(t)$ are all bounded above. Thus, in $\Omega$, system (1) is a well-posed and global attractor of the system.

**Proof of Theorem 4**. Utilizing Lemma 1 on the model (1), we acknowledge that $X_1 = (S, R)$ and $X_2 = (E, I)$ when the system at the DFE. At this DFE, the state variables are given by $X_1^* = (N, 0)$. It is significant to remember that,

$$\frac{dX_1}{dt} = F(X_1, 0) = \begin{pmatrix} \mu N - (\mu + \phi)S \\ -\mu R \end{pmatrix}$$

is linear and its result can be easily identified as,

$$R(t) = R(0)e^{-\mu t}, \ \ S(t) = N - (N - S(0))e^{-\mu t}.$$

Evidently, as $t \to \infty$, both $R(t)$ tends to 0 and $S(t)$ tends to $N$, regardless of the initial values of $R(0)$ and $S(0)$. Thus, the equilibrium point $X_1^* = (N, 0)$ is globally asymptotically stable, and condition (H1) is satisfied. Next,

$$G(X_1, X_2) = \begin{pmatrix} (\beta_1 E + \beta_2 I)S - (\alpha + \mu)E \\ \alpha E - (\mu + \delta + \gamma + \gamma_1)I \end{pmatrix}.$$

We can obtain,

$$A = \begin{pmatrix} \beta_1 N - (\alpha + \mu) & \beta_2 N \\ \alpha & -(\mu + \delta + \gamma + \gamma_1) \end{pmatrix}$$

with all non-negative off-diagonal elements. Consequently,

$$\hat{G}(X_1, X_2) = \begin{pmatrix} \beta_1 E(N - S) + \beta_2 I(N - S) \\ 0 \end{pmatrix}.$$

Since, $0 \le S \le N$, it is obvious that $\hat{G} \ge 0$. That leads to the global stability of DFE for $\mathcal{R}_0 < 1$.

**Proof of Theorem 5.** To examine the global stability of $\mathcal{E}^0$, we assume a Lyapunov functional $U_1(t)$,

$$U_1 = \bar{S}F(S\bar{S}) + E + I = (S - \bar{S} - \bar{S}\ln S\bar{S}) + E + I.$$

Here, $U_1$ is continuous, well-defined, and positive definite for all $(S, V, E, I, R, T) > 0$ and $\theta \in [0, \tau]$.

It illustrates that $U_1$ is always non-negative, and $U_1$ equals zero exclusively when assessed at the non-infective equilibrium point $\mathcal{E}^0$. Moreover, the global minimum of $U_1$ is achieved at $\mathcal{E}^0$. Consequently, all outcomes converge toward the infection-free steady state $\mathcal{E}^0$. Additionally, the functions $U_1$ along the system's trajectories adhere to the following relations:

$$\frac{dU_1}{dt} = \left(1 - \frac{\bar{S}}{S}\right)(\Lambda - (\beta_1 E + \beta_2 I)S - (\mu + \phi)S) + (\beta_1 E + \beta_2 I)S - (\mu + \alpha)E + \alpha E + \lambda(\beta_1 E + \beta_2 I)V - (\mu + \delta + \gamma + \gamma_1)I.$$

Utilizing the infection-free steady state of the model (1), $\Lambda = (\mu + \phi)\bar{S}$ in above expression, then the equation becomes,

$$\begin{aligned}\frac{dU_1}{dt} &\le -\frac{(\mu + \phi)}{S}(S - \bar{S})^2 + \beta_1 E\bar{S} + \beta_2 I\bar{S} - (\mu + \phi)E + \alpha E - (\mu + \delta + \gamma + \gamma_1)I \\ &\le -\frac{(\mu + \phi)}{S}(S - \bar{S})^2 + \left(\frac{\alpha}{\alpha + \mu}\left(\frac{S_0[\alpha\beta_2 + \beta_1(\gamma + \gamma_1 + \mu + \delta)]}{(\alpha + \mu)(\gamma + \gamma_1 + \delta + \mu)}\right) - 1\right)(\mu + \delta + \gamma + \gamma_1)I - \\ &\quad (\beta_1\bar{S} - \mu)E \\ &\le -\frac{(\mu + \phi)}{S}(S - \bar{S})^2 + \left(\frac{\alpha}{\alpha + \mu}\mathcal{R}_0 - 1\right)(\mu + \delta + \gamma + \gamma_1)I - \left(\beta_1\frac{\mu N}{\mu + \phi} - \mu\right)E \\ &\le 0.\end{aligned}$$

Here, we substituted the DFE value $S_0 = \frac{\mu N}{\mu + \phi}$. If $\mathcal{R}_0 < 1$, then $\frac{dU_1}{dt}$ is negative. Additionally, $\frac{dU_1}{dt} = 0$ if and only if $S(t) = \bar{S}$ and $E(t) = I(t) = R(t) = 0$. Therefore, based on the Lasalle invariance principle, the infection-free equilibrium point $\mathcal{E}^0$ is globally asymptotically stable over $\Omega$ in the scenario where $\mathcal{R}_0 < 1$.

## Appendix 2

**The endemic equilibrium point**. This appendix presents the brief calculation for computing fixed points in Section.

The variables for the EE are substituted as $(\tilde{S}, \tilde{V}, \tilde{E}, \tilde{I}, \tilde{R}, \tilde{T}) \equiv (S^*, V^*, E^*, I^*, R^*, T^*)$, where $E^* > 0$, and $I^* > 0$ also $E^* \neq 0$, $I^* \neq 0$. And we have the following system as follows,

$$\begin{cases} \Lambda - (\beta_1 E^* + \beta_2 I^*)S^* - (\mu + \phi)S^* = 0. \\ \phi S^* - (1 - \varepsilon)(\beta_1 E^* + \beta_2 I^*)V^* - \mu V^* = 0. \\ (\beta_1 E^* + \beta_2 I^*)S^* - (\alpha + \mu)E^* = 0. \\ \alpha E^* + (1 - \varepsilon)(\beta_1 E^* + \beta_2 I^*)V^* - (\mu + \delta + \gamma + \gamma_1)I^* = 0. \\ \gamma I^* - \mu R^* = 0. \\ \gamma_1 I^* - \mu T^* = 0. \end{cases} \quad (25)$$

Now, adding the first and third equation of (25) we have,

$$\begin{cases} \Lambda - (\mu + \phi)S^* = (\alpha + \mu)E^* \\ \Rightarrow S^* = \dfrac{\Lambda - (\alpha + \mu)E^*}{(\mu + \phi)} = \dfrac{\Lambda - a_1 E^*}{a_2}. \end{cases} \quad (26)$$

Here, $a_1 = \alpha + \mu$, and $a_2 = \mu + \phi$. Now, from the second equation of (25) we have,

$$\begin{cases} \phi S^* = \{\mu + \lambda(\beta_1 E + \beta_2 I)\}V^* \quad \text{where } \lambda = (1 - \varepsilon) \\ \\ \Rightarrow V^* = \dfrac{\phi S^*}{\mu + \lambda(\beta_1 E^* + \beta_2 I^*)} = \dfrac{\phi \left(\dfrac{\Lambda - a_1 E^*}{a_2}\right)}{\mu + \lambda(\beta_1 E^* + \beta_2 I^*)} \\ \\ \Rightarrow V^* = \dfrac{\phi(\Lambda - a_1 E^*)}{a_2\{\mu + \lambda(\beta_1 E^* + \beta_2 I^*)\}}. \end{cases} \quad (27)$$

Now, from (26) substituting the value of $S^*$, we get from the third equation of (25),

$$\Rightarrow S^*(\beta_1 E^* + \beta_2 I^*) = (\alpha + \mu)E^* = a_1 E^*$$

$$\Rightarrow \left(\frac{\Lambda - a_1 E^*}{a_2}\right)(\beta_1 E^* + \beta_2 I^*) = a_1 E^*$$

$$\Rightarrow a_1 \beta_1 E^{*2} + E(a_1 a_2 + a_1 \beta_2 I^* - \Lambda \beta_1) - \Lambda \beta_2 I^* = 0.$$

Now,

$$E^* = \frac{(\Lambda \beta_1 - a_1 a_2 - a_1 \beta_2 I^*) \pm \sqrt{(\Lambda \beta_1 - a_1 a_2 - a_1 \beta_2 I^*)^2 + 4\Lambda \beta_2 I^* a_1 \beta_1}}{2a_1 \beta_1}. \quad (28)$$

Two roots of the EE point of (28) will be real if and only if,

$$(\Lambda \beta_1 - a_1 a_2 - a_1 \beta_2 I^*)^2 > -4\Lambda \beta_2 I^* a_1 \beta_1.$$

Here, from the expression of (28), one root will be always positive, other will be positive if and only if $4\Lambda \beta_2 I^* a_1 \beta_1 < 0$. Now,

$$\frac{S_0 \alpha \beta_2 + S_0 \beta_1 (\gamma + \gamma_1 + \delta + \mu) + V_0 \beta_2 \lambda(\alpha + \mu)}{(\alpha + \mu)(\gamma + \gamma_1 + \delta + \mu)} > 1$$

$$\Rightarrow S_0 \alpha \beta_2 + S_0 \beta_1 a_3 + V_0 \beta_2 \lambda a_1 > a_1 a_3.$$

Where we let $a_3 = (\gamma + \gamma_1 + \delta + \mu)$. Now, from the second equation of (25) we have,

$$V^* = \frac{\phi(\Lambda - a_1 E^*)}{a_2(\mu + \lambda\lambda_1)}.$$

Where we let $\lambda_1 = \beta_1 E^* + \beta_2 I^*$, which is the force of infection of the model (25). Similarly,

$$R^* = \frac{\lambda I^*}{\mu}, \text{ and } T^* = \frac{\lambda_1 I^*}{\mu}.$$

Thus, at the endemic equilibrium $S^*$, $V^*$, $E^*$, $I^*$, $R^*$, and $T^*$ depends on the nature of $I^*$. Now, putting all expressions from above from the third equation of (25) we have,

$$(\beta_1 E^* + \beta_2 I^*)S^* = (\alpha + \mu)E^* \Rightarrow \frac{(\alpha + \mu)E^* - \beta_1 E^* S^*}{\beta_2 S^*} = I \Rightarrow \frac{(\alpha + \mu)E^* - \beta_1 E^*\left(\dfrac{\Lambda - a_1 E^*}{a_2}\right)}{\beta_2\left(\dfrac{\Lambda - a_1 E^*}{a_2}\right)} = I^* \tag{29}$$

$$\Rightarrow a_1 E^* - \beta_1 E^*\left(\frac{\Lambda - a_1 E^*}{a_2}\right) - \beta_2 I^*\left(\frac{\Lambda - a_1 E^*}{a_2}\right) = 0.$$

Now, from the fourth equation of (25) we have,

$$\begin{cases} \alpha E^* + \lambda(\beta_1 E^* + \beta_2 I^*)V^* - (\mu + \delta + \gamma + \gamma_1)I^* = 0. \\ \alpha E^* + \lambda(\beta_1 E^* + \beta_2 I^*)\left\{\dfrac{\phi(\lambda - a_1 E^*)}{a_2\{\mu + \lambda(\beta_1 E^* + \beta_2 I^*)\}}\right\} - (\mu + \delta + \gamma + \gamma_1)I^* = 0. \end{cases} \tag{30}$$

Let, the force of infection

$$\lambda_1 = (\beta_1 E^* + \beta_2 I^*), \ a_3 = (\mu + \delta + \gamma + \gamma_1), \text{ and } a_4 = (\mu + \lambda\lambda_1).$$

Now, by solving and simplifying the above two expressions using Mathematica we have obtained,

$$I^* = \frac{G_1 + G_2}{(2a_1 a_2 a_3 a_4(-a_2 a_4(a_3\beta_1 + \alpha\beta_2) + a_1\beta_2\lambda\lambda_1\phi))}.$$

Where

$\mathbb{K} = (a_2^2 a_4(-2a_1 a_2 a_3 a_4(a_3\beta_1 + \alpha\beta_2)\Lambda + a_4(a_3\beta_1 + \alpha\beta_2)^2\Lambda^2 + a_1^2 a_3(a_2^2 a_3 a_4 + 4\beta_2\lambda\Lambda\lambda_1\phi))),$

$G_1 = -(a_1^2 a_2^2 a_3 a_4\lambda\lambda_1 + a_2 a_4\alpha(a_2 a_4(a_3\beta_1 + \alpha\beta_2)\Lambda + \sqrt{\mathbb{K}})),$ and

$G_2 = a_1(a_2^3 a_3 a_4^2\alpha + a_2 a_4(-a_3\beta_1 + \alpha\beta_2)\lambda\Lambda\lambda_1\phi + \lambda\lambda_1\phi\sqrt{\mathbb{K}}).$

Thus, for the threshold parameter $\mathcal{R}_0 > 1$ we have,

$$a_4(a_3\beta_1 + \alpha\beta_2)^2\Lambda^2 + a_1^2 a_3(a_2^2 a_3 a_4 + 4\beta_2\lambda\Lambda\lambda_1\phi) > 2a_1 a_2 a_3 a_4(a_3\beta_1 + \alpha\beta_2)\Lambda.$$

Hence, the expression of the EE point of the model (1) is obtained.

Since different virus particles and infected cells are present in varied amounts, we refer to this as vital persistence. We can also shorten the point as $\mathcal{E}^* = (S^*, V^*, E^*, I^*, R^*, T^*)$.

In mathematical biology, $\mathcal{E}^0$ represents a short-lived infection that naturally clears from the body. In contrast, $\mathcal{E}^*$ signifies a situation where the body can't eliminate the illness on its own. In this case, the influenza infection becomes more noteworthy over time [8, 42]. Consequently, more sophisticated models accounting for latent infection, the impact of macrophages, the

cytotoxic immune response (CLT), or spatial dependence become essential to explain the dynamics of influenza spread throughout the body and its evolution toward an outbreak.

If the system explained by (1) reaches an equilibrium point, it will persist throughout the remaining period. Alternatively, the system is not required to reach these equilibrium values. However, it may approach the equilibrium, deviate from it, or oscillate between definite values. Conducting a comprehensive stability study of the system is essential for precisely predicting its behavior and understanding how it will interact with the equilibrium.

## Appendix 3

**Basic Reproduction Number (BRN).** A brief illustration of BRN for both controlling and without controlling strategies is included here for Section.

**Concise computation of BRN with control.** From the model (1), in the presence of vaccination class $\mathcal{R}_{0V}$ is known as basic reproduction number with control. With vaccination the DFE of (4) is,

$$\mathcal{E}^0 \equiv \left( \frac{\Lambda}{\mu + \phi}, \frac{\phi\Lambda}{\mu(\mu + \phi)}, 0, 0, 0, 0 \right).$$

We now apply the next generation matrix method to the model (1) and modeling only the exposed and infected compartments $E(t)$ and $I(t)$ is necessary since we are only interested in cells that disseminate infection. Hence, considering subpopulation $E(t)$ and $I(t)$ containing new infection terms and disease transmission terms, we can obtain the following subsystem,

$$\begin{aligned}
\frac{dE}{dt} &= (\beta_1 E + \beta_2 I)S - (\alpha + \mu)E. \\
\frac{dI}{dt} &= \alpha E + (1 - \varepsilon)(\beta_1 E + \beta_2 I)V - (\mu + \delta + \gamma + \gamma_1)I.
\end{aligned} \tag{31}$$

From the system (9), we obtain,

$$F = \begin{pmatrix} \beta_1 S_0 & \beta_2 S_0 \\ \lambda\beta_1 V_0 & \lambda\beta_2 V_0 \end{pmatrix}, \text{ and } V = \begin{pmatrix} \mu + \alpha & 0 \\ -\alpha & \mu + \delta + \gamma + \gamma_1 \end{pmatrix}.$$

Therefore,

$$V^{-1} = \begin{pmatrix} \dfrac{1}{\alpha + \mu} & 0 \\ \dfrac{\alpha}{(\alpha + \mu)(\mu + \delta + \gamma + \gamma_1)} & \dfrac{1}{(\mu + \delta + \gamma + \gamma_1)} \end{pmatrix}.$$

Here, $F$ and $V$ stand for the new infection term and transferred terms, respectively. Thus, the next-generation matrix $FV^{-1}$ is,

$$FV^{-1} = \begin{pmatrix} \dfrac{S_0\beta_1}{(\alpha + \mu)} + \dfrac{S_0\alpha\beta_2}{(\alpha + \mu)(\mu + \delta + \gamma + \gamma_1)} & \dfrac{S_0\beta_2}{(\mu + \delta + \gamma + \gamma_1)} \\ \dfrac{V_0\beta_1\lambda}{\alpha + \mu} + \dfrac{V_0\alpha\beta_2\lambda}{(\alpha + \mu)(\mu + \delta + \gamma + \gamma_1)} & \dfrac{V_0\beta_2\lambda}{(\mu + \delta + \gamma + \gamma_1)} \end{pmatrix}.$$

Here, the eigenvalues of $FV^{-1}$ are $\left\{ 0, \dfrac{S_0\alpha\beta_2 + S_0\beta_1\gamma + S_0\beta_1\gamma_1 + S_0\beta_1\delta + V_0\alpha\beta_2\lambda + S_0\beta_1\mu + V_0\beta_2\lambda\mu}{(\alpha + \mu)(\mu + \delta + \gamma + \gamma_1)} \right\}.$

Hence, the controlled basic reproduction number $\mathcal{R}_{0V}$ which is the spectral radius of $FV^{-1}$ is given as follows,

$$\begin{aligned} \mathcal{R}_{0V} = \rho(FV^{-1}) \quad &= \frac{S_0\alpha\beta_2 + S_0\beta_1\gamma + S_0\beta_1\gamma_1 + S_0\beta_1\delta + V_0\alpha\beta_2\lambda + S_0\beta_1\mu + V_0\beta_2\lambda\mu}{(\alpha+\mu)(\mu+\delta+\gamma+\gamma_1)} \\ &= \frac{\Lambda[\alpha\beta_2 + \beta_1(\mu+\delta+\gamma+\gamma_1)]}{(\mu+\phi)(\alpha+\mu)(\mu+\delta+\gamma+\gamma_1)} + \frac{\Lambda\phi\beta_2\lambda}{\mu(\mu+\phi)(\mu+\delta+\gamma+\gamma_1)}. \end{aligned} \tag{32}$$

**Concise computation of BRN without control.** From the model (1), in absence of vaccination class i.e. when $\lambda = (1-\varepsilon) = 0$, $\varepsilon = 1$, then the threshold quantity $\mathcal{R}_{0V}$ becomes $\mathcal{R}_0$ which is known as basic reproduction number without control. Without vaccination the DFE of (4) is, $\mathcal{E}^0 \equiv \left(\frac{\Lambda}{\mu+\phi}, \frac{\phi\Lambda}{\mu(\mu+\phi)}, 0, 0, 0, 0\right)$. We can extract the two following matrices from the system (31) which are $F$ and $V$ by replacing $\varepsilon = 0$. They are presented as follows,

$$F = \begin{pmatrix} \beta_1 S_0 & \beta_2 S_0 \\ 0 & 0 \end{pmatrix}, \quad \text{and} \quad V = \begin{pmatrix} \mu+\alpha & 0 \\ -\alpha & \mu+\delta+\gamma+\gamma_1 \end{pmatrix}.$$

Thus,

$$V^{-1} = \begin{pmatrix} \dfrac{1}{\alpha+\mu} & 0 \\ \dfrac{\alpha}{(\alpha+\mu)(\mu+\delta+\gamma+\gamma_1)} & \dfrac{1}{(\mu+\delta+\gamma+\gamma_1)} \end{pmatrix}.$$

Here, $F$ and $V$ stand for the new infection term and transferred terms, respectively. Thus, the next-generation matrix $FV^{-1}$ is,

$$FV^{-1} = \begin{pmatrix} \dfrac{S_0\beta_1}{(\alpha+\mu)} + \dfrac{S_0\alpha\beta_2}{(\alpha+\mu)(\mu+\delta+\gamma+\gamma_1)} & \dfrac{S_0\beta_2}{(\mu+\delta+\gamma+\gamma_1)} \\ 0 & 0 \end{pmatrix}.$$

Here, the eigenvalues of $FV^{-1}$ are $\left\{0, \frac{S_0\alpha\beta_2 + S_0\beta_1\gamma + S_0\beta_1\gamma_1 + S_0\beta_1\delta + S_0\beta_1\mu}{(\alpha+\mu)(\mu+\delta+\gamma+\gamma_1)}\right\}$.

Hence, the basic reproduction number $\mathcal{R}_0$ which is the spectral radious of $FV^{-1}$ is given by,

$$\begin{aligned} \mathcal{R}_0 = \rho(FV^{-1}) \quad &= \frac{S_0\alpha\beta_2 + S_0\beta_1\gamma + S_0\beta_1\gamma_1 + S_0\beta_1\delta + S_0\beta_1\mu}{(\alpha+\mu)(\mu+\delta+\gamma+\gamma_1)} \\ &= \frac{\Lambda[\alpha\beta_2 + \beta_1(\gamma+\gamma_1+\mu+\delta)]}{(\mu+\phi)(\alpha+\mu)(\gamma+\gamma_1+\delta+\mu)}. \end{aligned} \tag{33}$$

The basic reproduction number ($\mathcal{R}_0$) is defined as the average number of secondary infections resulting from the introduction of a single virus cell into a host where every target cell is susceptible. In our model, $\mathcal{R}_0$ depends on two variables: the average number of target cells per unit of time (considering natural death) and the rate of disease transmission by an infective cell [29, 44].

## Supporting information

**S1 Data.**
(XLSX)

**S1 File.**
(ZIP)

**S2 File.**
(ZIP)

**S3 File.**
(ZIP)

**S4 File.**
(ZIP)

## Author Contributions

**Conceptualization:** Kazi Mehedi Mohammad, Md. Kamrujjaman.

**Data curation:** Kazi Mehedi Mohammad.

**Formal analysis:** Kazi Mehedi Mohammad, Md. Kamrujjaman.

**Funding acquisition:** Md. Kamrujjaman.

**Investigation:** Asma Akter Akhi.

**Methodology:** Kazi Mehedi Mohammad.

**Resources:** Asma Akter Akhi.

**Software:** Kazi Mehedi Mohammad, Md. Kamrujjaman.

**Supervision:** Md. Kamrujjaman.

**Validation:** Asma Akter Akhi, Md. Kamrujjaman.

**Writing – original draft:** Kazi Mehedi Mohammad.

**Writing – review & editing:** Asma Akter Akhi, Md. Kamrujjaman.

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
