## [Decision Letter · Decision Letter 0]

9 Sep 2024

PONE-D-24-32461Bifurcation Analysis of an Influenza A (H1N1) Model with Treatment and VaccinationPLOS ONE

Dear Dr. Kamrujjaman,

Thank you for submitting your manuscript to PLOS ONE. After careful consideration, we feel that it has merit but does not fully meet PLOS ONE’s publication criteria as it currently stands. Therefore, we invite you to submit a revised version of the manuscript that addresses the points raised during the review process.

We look forward to receiving your revised manuscript.

Kind regards,

Junyuan Yang

Academic Editor

PLOS ONE

Journal Requirements:

Reviewers' comments:

Reviewer's Responses to Questions

**Comments to the Author**

1. Is the manuscript technically sound, and do the data support the conclusions?

Reviewer #1: Yes

Reviewer #2: Partly

2. Has the statistical analysis been performed appropriately and rigorously? 

Reviewer #1: Yes

Reviewer #2: Yes

3. Have the authors made all data underlying the findings in their manuscript fully available?

Reviewer #1: Yes

Reviewer #2: Yes

4. Is the manuscript presented in an intelligible fashion and written in standard English?

Reviewer #1: Yes

Reviewer #2: Yes

5. Review Comments to the Author

Reviewer #1: In this article, the authors have studied a deterministic SVEIRT epidemic model for influenza, incorporating vaccination and treatment strategies. The model is proven to be well-posed, with the DiseaseFree Equilibrium (DFE) being globally asymptotically stable when the basic reproduction number, R0, is less than one, as demonstrated using Lyapunov functions and the LaSalle Invariance Principle. When R0 exceeds one, a unique Endemic Equilibrium (EE) exists, whose local and global stability are established through the Routh-Hurwitz criterion and non-linear Lyapunov functions, respectively. Moreover, numerical simulations of the model support the existence and stability of DFE and EE. The simulations indicate that higher transmission rates (β1, β2) and infection acquisition rates (α) lead to a rapid increase in disease burden. It is observed that if the rate of acquiring infection increases, then the disease burden will increase. Also, the growth rate in the E(t) compartment raises the disease burden. Meanwhile, treatment rate and vaccination rate can mitigate the factor R0 effectively. Furthermore, the contact rates β1 and β2 resulted in the rapid increase of the disease in the community. Also, the treatment of infected individuals in the acute phase is significant. Human recruitment rate can provide a great influence on controlling the disease.

This article may be publishable.

The following important references may be cited in the text:

1. Stability analysis and optimal control of avian influenza virus A with time delays, International Journal of Dynamics and Control, 6(3), pp. 1351-1366 (2018). http://dx.doi.org/10.1007/s40435-017-0379-6

2. Permanence and extinction for a nonautonomous avian-human influenza epidemic model with

distributed time delay, Mathematical and Computer Modelling, 52(9-10), pp. 1794-1811 (2010).

http://dx.doi.org/10.1016/j.mcm.2010.07.006

3. Global Dynamics of a Nonautonomous SIRC Model for Influenza A with Distributed Time Delay,

Differential Equations and Dynamical Systems, 18(4), pp. 341-362 (2010). http://dx.doi.org/10.1007/s12591-010-0066-y

4. Dynamical behavior of SIRS model incorporating government action and public response in presence of deterministic and fluctuating environments. Chaos, Solitons and Fractals, 164 (2022) 112643. https://doi.org/10.1016/j.chaos.2022.112643

5. Periodic transmission and vaccination effects in epidemic dynamics: a study using the SIVIS model.

Nonlinear Dyn 112, 2381–2409 (2024). https://doi.org/10.1007/s11071-023-09157-4

6. Modelling disease transmission through asymptomatic carriers: a societal and environmental perspective. Int. J. Dynam. Control (2024). https://doi.org/10.1007/s40435-024-01387-7

7. Assessing the influence of public behavior and governmental action on disease dynamics: a PRCC analysis and optimal control approach. Eur. Phys. J. Plus 139, 527 (2024). https://doi.org/10.1140/epjp/s13360-024-05327-4

8. Analysis of a delayed epidemic model with pulse vaccination. Chaos, Solitons & Fractals 66 (2014) 74–85.

http://dx.doi.org/10.1016/j.chaos.2014.05.008

Reviewer #2: After a thorough review, I believe the paper can be reconsidered after addressing the following points comprehensively:

1. Rewrite the first three lines of the abstract.

2. There are several formatting issues. Once the terms "disease-free equilibrium (DFE)" and "endemic equilibrium (EE)" are introduced, consistently use the abbreviations throughout the text.

3. In the *Model Formulation* section, move the journal discussion to the *Introduction* and focus solely on the formulation aspects. Provide a detailed explanation of the transmission parameters and state variables.

4. Clearly state the assumptions imposed in the model formulation in this section.

5. Figure 2 is incomplete and lacks explanation. Include a detailed flow chart.

6. Discuss the biological feasibility and reference studies where, in the case of Influenza, a vaccinated person can still acquire the infection.

7. What about the waning of vaccine immunity in the model? Consider addressing this aspect.

8. Specify the values of \\(\\beta_1\\) and \\(\\beta_2\\) in Table 1, used for simulations.

9. All parameters used in the simulation are sourced from the literature, as indicated in Table 1. Clarify the purpose of using real data and parameter fitting.

10. Conduct a simulation analysis of the model based on estimated parameters.

11. Suggest effective control parameters.

12. Estimate parameters and \\(R_0\\) based on real data.

13. In Theorem 5, how is \\(dU_1/dt < 0\\)? This result is ambiguous. Please review it carefully.

14. Analyze the impact of the vaccination parameter on the reproduction number and, ultimately, on disease incidence.

15. From Table 2, \\(\\beta_1 > \\beta_2\\), meaning the transmission probability from the exposed class is greater than the infected class. Explain the rationale behind this.

6. PLOS authors have the option to publish the peer review history of their article (what does this mean?). If published, this will include your full peer review and any attached files.

Reviewer #1: No

Reviewer #2: No

---

## [Author Response · Author response to Decision Letter 0]

10 Oct 2024

Please see the attached file, file name "Response_R1_PONE-D-24-32461 "

---

## [Decision Letter · Decision Letter 1]

25 Nov 2024

Bifurcation Analysis of an Influenza A (H1N1) Model with Treatment and Vaccination

PONE-D-24-32461R1

Dear Dr. Kamrujjaman,

We’re pleased to inform you that your manuscript has been judged scientifically suitable for publication and will be formally accepted for publication once it meets all outstanding technical requirements.

Kind regards,

Junyuan Yang

Academic Editor

PLOS ONE

Additional Editor Comments (optional):

Reviewers' comments:

Reviewer's Responses to Questions

**Comments to the Author**

1. If the authors have adequately addressed your comments raised in a previous round of review and you feel that this manuscript is now acceptable for publication, you may indicate that here to bypass the “Comments to the Author” section, enter your conflict of interest statement in the “Confidential to Editor” section, and submit your "Accept" recommendation.

Reviewer #1: (No Response)

Reviewer #2: All comments have been addressed

2. Is the manuscript technically sound, and do the data support the conclusions?

Reviewer #1: (No Response)

Reviewer #2: Yes

3. Has the statistical analysis been performed appropriately and rigorously? 

Reviewer #1: (No Response)

Reviewer #2: Yes

4. Have the authors made all data underlying the findings in their manuscript fully available?

Reviewer #1: (No Response)

Reviewer #2: Yes

5. Is the manuscript presented in an intelligible fashion and written in standard English?

Reviewer #1: (No Response)

Reviewer #2: Yes

6. Review Comments to the Author

Reviewer #1: (No Response)

Reviewer #2: Dear Editor,

I have carefully read the revised version. The paper can be accepted in the current form.

Thank you

7. PLOS authors have the option to publish the peer review history of their article (what does this mean?). If published, this will include your full peer review and any attached files.

Reviewer #1: No

Reviewer #2: No

---

## [Editor Report · Acceptance letter]

9 Dec 2024

PONE-D-24-32461R1 

PLOS ONE

Dear Dr. Kamrujjaman, 

I'm pleased to inform you that your manuscript has been deemed suitable for publication in PLOS ONE. Congratulations! Your manuscript is now being handed over to our production team.

Kind regards, 

on behalf of

Dr. Junyuan Yang 

Academic Editor

PLOS ONE